# Novel regulators of heparan sulfate proteoglycans modulate cellular uptake of α-synuclein fibrils
Benoît Vanderperre [1,2,3,4,10] ✉, Amitha Muraleedharan[1,2,3,10], Marie-France Dorion [5], Frédérique Larroquette[4], Esther Del Cid Pellitero [4], Nishani Rajakulendran[6], Carol X.-Q. Chen[5], Roxanne Larivière[4], Charlotte Michaud-Tardif[4], Thomas Goiran[4], Rony Chidiac[6], Damien Lipuma[1,2,3], Graham MacLeod[6], Rhalena Thomas [4,5], Zhangjie Wang[7], Wolfgang E. Reintsch[5], Wen Luo [5], Irina Shlaifer [5], Fuming Zhang [8], Ke Xia[8], Zachary Steinhart[6], Robert J. Linhardt [8,11], Jean-François Trempe [9], Jian Liu[7], Thomas M. Durcan[5], Stephane Angers [6] & Edward A. Fon[4] ✉

Synucleinopathies are characterized by the accumulation and propagation of α-synuclein (α-syn) aggregates throughout the brain, leading to neuronal dysfunction and death. In this study, we used an unbiased FACS-based genome-wide CRISPR/Cas9 knockout screening to identify genes that regulate the entry and accumulation of α-syn preformed fibrils (PFFs) in cells. We identified key genes and pathways specifically implicated in α-syn PFFs intracellular accumulation, including heparan sulfate proteoglycans (HSPG) biosynthesis and Golgi trafficking. All confirmed hits affected heparan sulfate (HS), a post-translational modification known to act as a receptor for proteinaceous aggregates including α-syn and tau. Intriguingly, deletion of *SLC39A9* and *C3orf58* genes, encoding respectively a Golgi-localized exporter of $Zn^{2+}$, and the Golgi-localized putative kinase DIPK2A, specifically impaired the uptake of α-syn PFFs, by preventing the binding of PFFs to the cell surface. Mass spectrometry-based analysis of HS chains in *SLC39A9*$^{-/-}$ and *C3orf58*$^{-/-}$ cells indicated major defects in HS homeostasis. Additionally, Golgi accumulation of NDST1, a prime HSPG biosynthetic enzyme, was detected in *C3orf58*$^{-/-}$ cells. Interestingly, *C3orf58*$^{-/-}$ human iPSC-derived microglia and dopaminergic neurons exhibited a strong reduction in their ability to internalize α-syn PFFs. Altogether, our data identifies new modulators of HSPGs that regulate α-syn PFFs cell surface binding and uptake.

Synucleinopathies are a class of neurodegenerative disorders that include Parkinson's disease (PD), Dementia with Lewy Bodies (DLB), and Multiple System Atrophy (MSA)[1]. The molecular hallmark of these neurodegenerative diseases is the presence in the brain of aggregates composed mainly of the protein α-synuclein (α-syn)[2]. In these diseases, α-syn misfolds and aggregates in a prion-like amyloidogenic cascade culminating, in PD, in the

formation of higher-order aggregates termed Lewy bodies (LBs) and Lewy neurites. Similar to other proteinopathies, evidence suggests that cell-to-cell transmission of α-syn aggregates underlies disease progression[3,4]. It was first exemplified by the appearance of LBs in initially healthy grafted tissue following transplantation in the brain of a PD patient[5]. Later on, in vitro experiments and the use of animal models have confirmed that α-syn

[1]Département des Sciences biologiques, Université du Québec à Montréal, Montréal, QC, Canada. [2]Centre d'Excellence de Recherche sur les Maladies Orphelines – Fondation Courtois (CERMO-FC), Montréal, QC, Canada. [3]Network for Research on Protein Function, Engineering and Applications (PROTEO), Montréal, QC, Canada. [4]McGill Parkinson Program, Department of Neurology and Neurosurgery, Montreal Neurological Institute and Hospital, McGill University, Montréal, QC, Canada. [5]The Neuro's Early Drug Discovery Unit (EDDU), Department of Neurology and Neurosurgery, Montreal Neurological Institute-Hospital, McGill University, Montréal, QC, Canada. [6]Leslie Dan Faculty of Pharmacy, University of Toronto, Toronto, ON, Canada. [7]Division of Chemical Biology & Medicinal Chemistry, Eshelman School of Pharmacy, University of North Carolina at Chapel Hill, Chapel Hill, NC, USA. [8]Department of Chemistry and Chemical Biology, Rensselaer Polytechnic Institute, Troy, NY, USA. [9]Department of Pharmacology & Therapeutics and Centre de Recherche en Biologie Structurale, McGill University, Montréal, QC, Canada. [10]These authors contributed equally: Benoît Vanderperre, Amitha Muraleedharan. [11]Deceased is Robert J. Linhardt. ✉e-mail: vanderperre.benoit@uqam.ca; ted.fon@mcgill.ca

preformed fibrils (PFFs) made of recombinant α-syn can reach the intracellular compartment in a wide variety of cell types[6–9]. This is followed by self-templated aggregation of native cytosolic α-syn, and transmission of newly formed aggregates to neighboring cells by a variety of proposed mechanisms, including the secretion of exosomes[10], tunneling nanotubes[11,12], trans-synaptic spread[13], or misfolding-associated protein secretion[14].

The molecular mechanisms underlying these events are poorly understood. This is especially the case for the entry of extracellular aggregates in recipient cells, where several endocytic pathways (clathrin-dependent[15,16] and independent[17] endocytosis, macropinocytosis[18,19]) and surface receptors (LAG3[20], heparan sulfate proteoglycans (HSPGs)[16,18] including neurexin 1β[20–22]) have been identified, likely because of the variety of cellular models or libraries used in overexpression screens. Because spreading could potentially be driven by several cell types within the brain[23–25], it is important to unravel mechanisms and molecular players that underlie this cell-to-cell transmission across cell types, but that are also sufficiently specific to α-syn aggregates uptake to prevent advert effects of possible therapies.

In this regard, HSPGs are a very promising class of receptors, having been shown to mediate α-syn PFFs uptake in several neural cell types including mouse primary neurons[16,18] and neuroblastoma, oligodendrocyte-, astrocyte- and to a lesser extent microglia-like cell lines[26]. HSPGs are key multifunctional components of the cell surface and extracellular matrix (ECM) that play crucial structural and communication roles and can act as receptors/coreceptors of a wide variety of ligands[27]. HSPGs are post-translationally modified with heparan sulfate (HS) chains composed of disaccharide units. HSPG biosynthesis starts in the endoplasmic reticulum (ER) and continues in the Golgi apparatus where the exostosin complex (composed of EXT1 and/or EXT2) catalyzes the elongation of the disaccharide chain[27]. Modification of the HS chain then occurs also in the Golgi, mainly by sulfation (which adds negative charges) at various positions of disaccharide units by several sulfotransferases, using PAPS (3′-phosphoadenosine-5′-phosphosulfate) as a sulfate donor[28]. Depending on the core protein that bears the HS moieties, plasma-membrane HSPGs fall mainly into two classes: the membrane integral syndecans (SDC) or glycosylphosphatidylinositol-anchored glypicans (GPC)[27]. Once they reach the cell surface, HSPGs expose their negatively charged HS chains to serve as membrane receptors and co-receptors for a wide range of positively charged cargoes[27,29]. Recently, it was shown that knock-out (KO) of the SLC35B2 transporter, responsible for PAPS import into the Golgi and sulfation of HS, drastically reduces the uptake of α-syn PFFs by HEK293T cells and mouse primary neurons[16]. This study uncovered a general mechanism of HSPG-dependent cell surface binding of positively charged cargoes is necessary for uptake of α-syn PFFs, even if several endocytic mechanisms may substitute for one another (e.g., clathrin-mediated endocytosis[16], macropinocytosis[18]) downstream of HSPG binding. Therefore, identifying α-syn PFFs-specific modulators of HSPG-dependent cell-surface binding could prove very useful to develop therapies aiming at reducing the intercellular spread of α-syn aggregates, while preventing adverse effects that global HSPG impairment could cause.

In the present study, we aimed to identify cellular factors affecting the uptake of α-syn PFFs using an unbiased genome-wide CRISPR/Cas9 KO screen in human cells. Our screen confirmed that HSPGs are major receptors for α-syn PFFs, with several genes identified regulating HSPG expression, and uncovered new genes, the silencing of which strongly and specifically inhibit α-syn PFFs uptake. More specifically, we found that loss of C3orf58, a putative kinase in the Golgi, or SLC39A9, a $Zn^{2+}$ exporter in the Golgi membrane, results in marked perturbations in HS homeostasis, which was associated with a large reduction in PFFs uptake secondary to decreased PFFs binding to the cell surface. We also show that C3orf58 is necessary for PFFs uptake in human iPSC-derived microglia and dopaminergic neurons. C3orf58 also regulated Golgi accumulation of the key HS biosynthetic enzyme NDST1. Overall, our study provides an unprecedented, pathway-level view of α-syn PFFs uptake mechanisms and highlights novel regulators of HS homeostasis, which will help in the design of future therapeutic strategies.

## Results

### Genome-wide CRISPR screening identifies genetic modifiers of α-syn PFFs accumulation

To identify key genes involved in α-syn PFFs accumulation, we performed a pooled genome-wide CRISPR/Cas9 knock-out screen in a transformed human Retina Pigmented Epithelial cell line (RPE-1) (Fig. 1A), monitoring cell fluorescence after a 24-hour treatment with fluorescently labeled α-syn PFFs (see Supplementary Fig. 1A–E for quality control of PFFs by transmission electron microscopy, ThT fluorescence and sedimentation assays). Cells with the 15% lowest and 15% highest PFFs fluorescence were isolated by FACS. Next-generation sequencing was used to identify the sgRNAs present in each population, and sgRNAs/target genes enrichment analysis between the two populations was performed using MAGeCK[30] (Fig. 1B, Tables S1, S2, S3). Genes enriched in the "Low PFFs" population represent putative facilitators of α-syn PFFs accumulation because disrupting them results in a lower PFFs amount per cell. Conversely, genes enriched in the "High PFFs" population are putative inhibitors of PFFs accumulation. Gene ontology analysis (Fig. 1C) indicated that genetic facilitators of PFFs accumulation were mainly associated with Golgi vesicle trafficking (COPB1, COPG1, TMED10), heparan sulfate biosynthesis (EXT1, NDST1, SLC10A7[31], TM9SF2, SLC39A9), and to a lesser extent phagosome acidification (v-ATPase subunits). Other well-ranked genes included the cell cycle regulators TP53 and CDKN1A as well as a poorly characterized gene, C3orf58. Inhibitors of PFFs accumulation were surprisingly mostly related to metabolic processes needed for cell growth and cell cycle such as nucleic acid metabolism. Exceptions were the MOSPD2 and STARD3 genes, which encode two interacting proteins that regulate cholesterol transfer from the endoplasmic reticulum to endosomes[32], and VPS35, which regulates endosomal trafficking and is mutated in familial forms of PD[33]. Explaining the identification of genes related to cell cycle/growth, several lines of evidence indicated a clear link between cell size and PFFs accumulation. This is described in detail in Supplementary Fig. 2 and prompted us to design a validation strategy that takes into account this cell size bias at play in our screen.

### Validation of hits by high-content microscopy

To identify genes that modulate α-syn PFFs accumulation independently from effects on cell size, we normalized PFFs content to cell size using high-content microscopy (Fig. 2). We used RPE-1 cells stably expressing Cas9 and EGFP and transfected them with synthetic sgRNAs against a selection of putative hits (Fig. 2A; for hits selection criteria, see Supplementary Fig. 3A). Two sgRNAs per gene were used:, one from the tKOv3 library, and one custom sgRNA. Additionally, control sgRNAs targeting the AAVS1 locus were used to induce double-strand break formation similar to those induced by sgRNAs targeting selected hits to negate the effects of randomized DNA breaks while minimizing perturbations in gene expression[34]. Three days post-transfection, a time at which more than 80% of cells were successfully gene-edited using an EGFP-targeting sgRNA (Supplementary Fig. 3B), we performed a 24 h accumulation assay with fluorescent α-syn PFFs. Using high-content microscopic analysis, we quantified the α-syn PFFs fluorescent signal normalized to the area of each cell, using an EGFP mask (Fig. 2B). Consistent with the possible enrichment of fast-cycling cells in the "Low PFFs" population from the screen, knocking-out the tumor suppressors CDKN1A, TP53, and DYRK1A significantly reduced cell size but did not decrease the normalized PFFs accumulation. Conversely, knock-out of known oncogenes or genes that mediate G2/M cell cycle transitions (CDK1, GLN3, KIF11, KIF23, PSMD12/13) increased cell size at least for one of the two sgRNAs without modifying normalized PFFs content, explaining their enrichment in the "High PFFs" population of the screen. Importantly, the normalization of α-syn PFFs content to cell size allowed us to validate 7 facilitators (C3orf58, COPB1, COPG1, EXT1, NDST1, SLC39A9, and TM9SF2) and one inhibitor (the PD-related gene VPS35) of α-syn PFFs

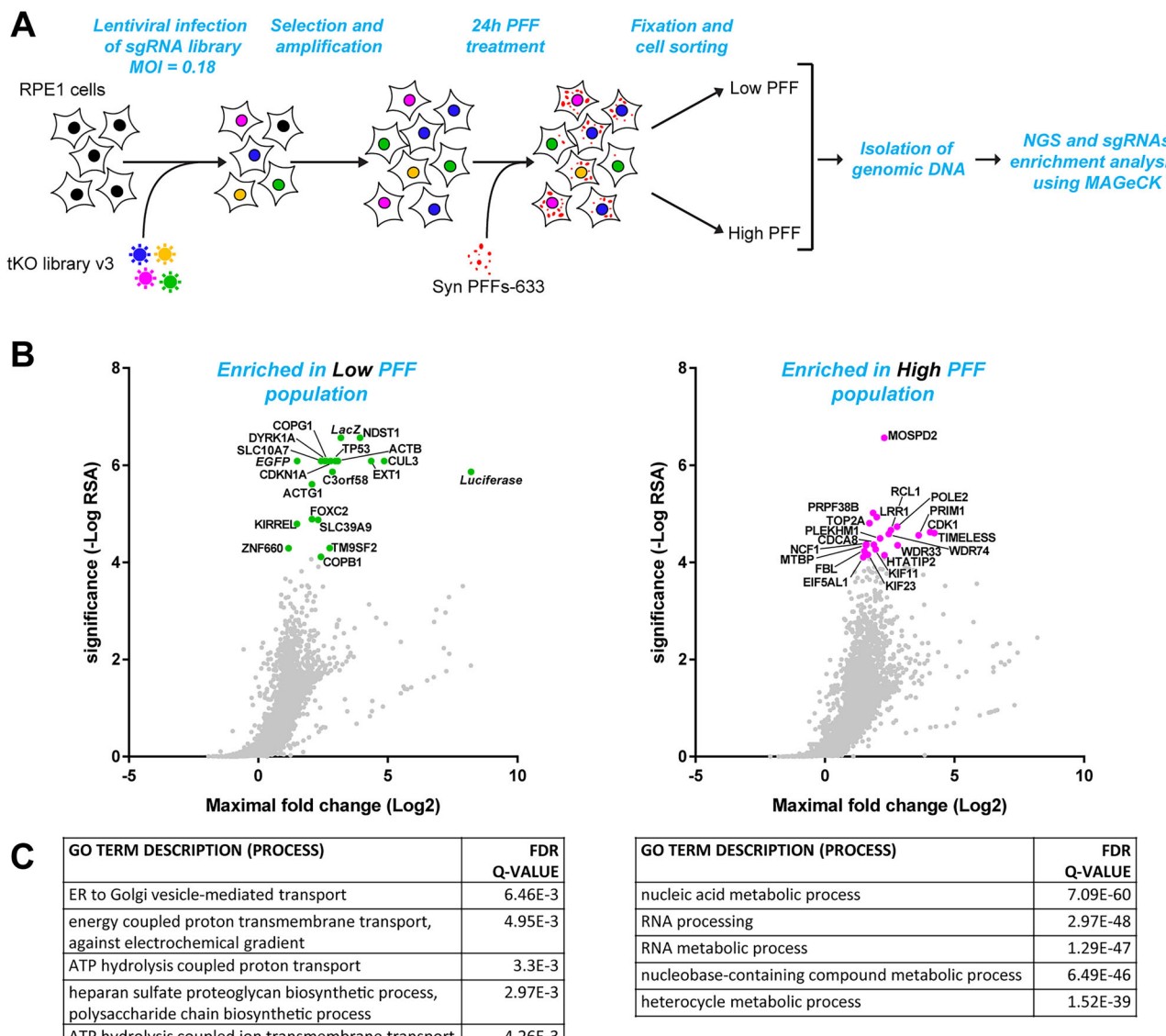

**Fig. 1 | CRISPR/Cas9 screening identifies genetic modifiers of α-syn PFFs accumulation in RPE-1 cells. A** Strategy of the genome-wide CRISPR screening used to identify genetic modifiers of α-syn-PFFs accumulation in RPE-1 cells. FACS-sorting was used to isolate the cell populations with the 15% lower and 15% higher PFFs fluorescence intensity. **B** sgRNA enrichment in the Low PFFs population (left panel) and High PFFs population (right panel) was calculated using the MAGeCK algorithm. For all genes in the tKOv3 library, the significance (reproducibility of effect across all 4 sgRNAs for a given gene) was plotted as a function of the maximal Log2 fold change (sgRNA showing the highest enrichment for that gene). The top 20 most significant genes are shown as green or magenta dots, and the associated gene symbols are indicated. **C** Gene ontology (GO) analysis was performed with the GOrilla online tool, with the ranked lists of genes from the MAGeCK analysis as inputs (Low PFFs population, left; High PFFs population, right). The enriched GO terms for the Process category and their associated false-discovery rate Q-values are reported in the tables.

accumulation (Fig. 2B, Supplementary Fig. 3C). *VPS35* deficiency was previously reported to induce an increase in the uptake of amyloid-beta (Aβ) aggregates through an unknown mechanism[35], and it thus appears that *VPS35* could play a broad role in mediating the cellular uptake of various species of protein aggregates. The *EXT1* and *NDST1* genes encode two key biosynthetic enzymes in the biogenesis of heparan sulfate (HS), a post-translational modification that is known to mediate the cellular attachment and entry of α-syn fibrils, as well as FTD-associated tau oligomers[18,36]. In addition, *COPB1* and *COPG1*, encoding two coatamer subunits that mediate retrograde vesicular trafficking in the Golgi apparatus, were previously identified in a screen for cellular entry of *Chlamydia trachomatis* by indirectly impacting HS biogenesis[37]. *TM9SF2* may function similarly since TM9SF family proteins have a consensus KxD/E motif (KVD in *TM9SF2*) at the C terminus, which interacts with the *COPI* coatamer[38]. It was previously shown that *TM9SF2* knock-out reduces HS surface expression by affecting

the proper localization and stability of NDST1[39]. Similarly, knock-out of *SLC39A9*, encoding a Zn²⁺ exporter in the Golgi membrane, results in a small decrease in HS surface expression, possibly explaining its identification as an important factor in Chikungunya virus infection[39]. The molecular and cellular function of C3orf58 (also known as DIPK2A or DIA-1) is unclear. This Golgi localized protein was reported to positively regulate autophagosome-lysosome fusion[40], to act as an insulin-like growth factor receptor 1 ligand[41], and to colocalize with β-COP proteins suggesting a role in the secretory apparatus[42]. It might also be a key player in Zika virus infection of neural stem cells through an undefined mechanism[43], which remains to be confirmed.

**Cargo specificity of validated hits**

To determine whether the 8 validated hits were specific for α-syn PFFs accumulation or also affected other cargoes, we tested their ability to also

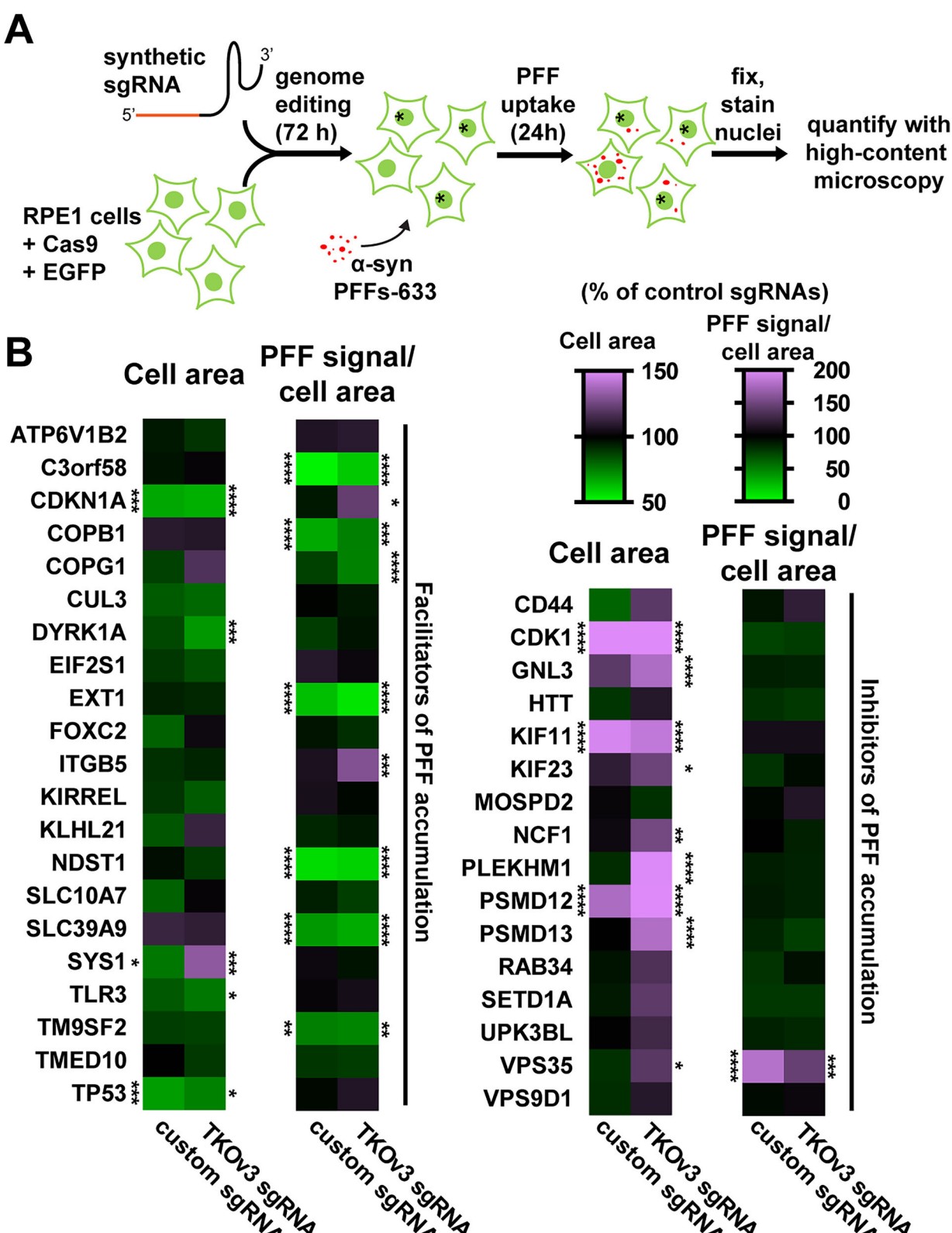

Fig. 2 | Screen validation by high-content microscopy. A Experimental pipeline used for hits validation by high-content microscopy. RPE-1 cells stably expressing Cas9 nuclease and EGFP are transfected with individual synthetic sgRNAs in 96-well plates. After 3 days, the obtained polyclonal gene-edited cells are subjected to a 24 h α-syn-PFFs-A633 uptake assay before fixation, nuclei stain with Hoechst, and quantification by high-content microscopy. An EGFP mask allows quantification of cell area, to which PFFs content is normalized. Imaging was done using a CX7 high-content microscope and quantification with the HCS Studio Cell Analysis software. B Heatmap summarizing validation data. On a per-cell basis, the EGFP area and

PFFs-A633/cell area ratio were measured, for each indicated gene, and for two sgRNAs per gene (a custom sgRNA and one from the tKOv3 library). The mean cell area and mean PFFs-A633/cell area were calculated and reported as percent of control sgRNAs targeting the AAVS1 locus (see color coding at the top right). $n = 4$-7 independent experiments. Genes symbols are sorted alphabetically, with putative hits enriched in the Low PFFs population first, followed by putative hits enriched in the High PFFs population. Statistical test: one-way ANOVA; *$p < 0.05$, **$p < 0.01$, ***$p < 0.001$, ****$p < 0.0001$.

perturb the accumulation of dextran (bulk endocytosis), EGF, and transferrin (both receptor-mediated endocytosis) as well as tau oligomers (HS-dependent macropinocytosis)[18,44] (Fig. 3, Supplementary Fig. 4A). Among the seven facilitators of α-syn PFFs accumulation, none of the gene KOs decreased dextran, EGF, or transferrin accumulation, nor did the silencing of *VPS35* increase the accumulation of these cargoes (Fig. 3A-C), at least at the 24 h time point tested. However, consistent with their dependency on HS for internalization, Tau oligomers (Fig. 3D) accumulated significantly less in *EXT1* or *NDST1* silenced cells, and almost significantly for *TM9SF2*. This confirms previous findings that α-syn PFFs and Tau oligomers accumulate intracellularly via a partially overlapping HS-dependent mechanism[18,36]. This is further illustrated in a competition assay, where co-treatment of RPE-1 cells with Tau oligomers and α-syn PFFs decreased the accumulation of both types of aggregates relative to separate treatments (Supplementary Fig. 4B). Tau oligomers outcompeted α-syn PFFs likely due to their reduced size, thus larger numbers and capacity to saturate binding sites at the cell surface. Similar to its effect on α-syn PFFs (Fig. 3E) and amyloid-β oligomers[35], silencing of *VPS35* increased the accumulation of Tau oligomers (Fig. 3D). Together, these data suggest that the genes identified through our screen specifically modulate the accumulation of proteinaceous aggregates. Two of these (*C3orf58* and *SLC39A9*) were specific for α-syn PFFs.

### HS mediates most of α-syn PFFs binding at the cell surface

To further test whether HS are major receptors for α-syn PFFs in RPE-1 cells, we first sought to determine if the binding of α-syn PFFs to the cell surface was HSPG-dependent. We incubated live cells with labeled PFFs on ice to prevent endocytosis and monitored PFFs binding at the cell surface by fluorescence microscopy. As expected, pre-treatment with either heparinases (to strip-off HS from the cell surface) or sodium chlorate (to inhibit HS sulfation) strongly reduced the binding of α-syn PFFs to the cell surface (Fig. 4A, B). We also monitored surface HS expression in these intact cells by immunofluorescence using a commonly used antibody against the 10e4 epitope (N-sulfated HS, full epitope unknown)[45]. The signal was markedly reduced by heparinases treatment, confirming the specificity of the antibody towards HS. However, despite the strong inhibition of PFFs binding to the cell surface following sodium chlorate treatment, 10e4 epitope signal was nearly unaffected which could be explained by the differential effect of sodium chlorate on *N*- and *O*-sulfations[46], showing that this antibody does not fully capture the complexity of HS chains architecture and modifications. In other words, the 10e4 signal is conserved even in conditions where HS sulfation is inhibited to a degree that functionally interferes with PFFs binding.

To the best of our knowledge, it is unknown if HS are also involved in α-syn PFFs uptake in human dopaminergic neurons. To test this, we generated iPSC-derived dopaminergic neurons (iDNs) and treated them with heparinases or sodium chlorate before PFF binding and uptake assays. Both treatments significantly decreased PFFs binding to the surface of dopaminergic neurons (Fig. 4C, D), although the low dynamic range of HS signal likely prevented observation of any change in HS intensity by HS-altering treatments (Fig. 4E). These treatments were however confirmed to work properly, as a significant reduction in HS in presence of heparinases or sodium chlorate was observed in a 3 hours PFFs uptake assay (Fig. 4F,H). PFF uptake in iDNs was significantly decreased upon exposure to sodium chlorate (app. 40% of control). This was not observed for heparinases (Fig. 4G), which could be due to gradual surface presentation of initially intracellular pools of HSPGs during the 3 h uptake assay, enzymatic inactivation with time, or the action of other PFFs receptors. Together, our data confirm that HS are major receptors for cell surface binding of α-syn PFFs, including in RPE-1 cells and dopaminergic neurons, even if we cannot exclude that other receptors might compensate for PFF uptake in iDNs when HS are altered.

### Our screen identifies lesser-studied perturbators of HS homeostasis

Membrane-bound HS can serve as a primary receptor for certain ligands before endocytosis via a secondary, generally more specific receptor, but it

has also been shown to be endocytosed together with its ligand following binding at the cell surface[47]. Of the 8 hits identified in our screen, 6 have been previously linked with defects in HS expression. As mentioned previously, the EXT1 enzyme catalyzes the elongation of the disaccharide chain (GlcN-GlcNAc), while NDST1 is crucial for N-deacetylation/N-sulfation of GlcNAc residues, thereby regulating the binding of various ligands depending on the extent of sulfation and sulfation pattern of the HS chain. *COPB1* and *COPG1* have been shown to indirectly affect HS surface presentation[37], but because these are core essential genes[48], we did not investigate them further. In addition, the *SLC35B2* gene, which encodes a PAPS importer in the Golgi that is necessary for HS sulfation, was also well scored in the screen (Table S1), consistent with a previous report identifying it as a key regulator of PFFs uptake[16].

Although, *SLC39A9* and *TM9SF2* have been suggested to indirectly affect HS surface expression[39], the role of *C3orf58* and *VPS35* in HS biology has not been investigated previously. To evaluate a possible effect on HS expression of these four additional identified hits not directly implicated in HS biosynthesis, we generated monoclonal CRISPR-edited cells for these genes (Supplementary Fig. 5). For our two hits that were specific for α-syn PFFs (*C3orf58* and *SLC39A9*), we also generated rescued lines by reintroducing active forms of the proteins. We quantified the HS signal with the 10e4 antibody in fixed cells, before or after permeabilization, to quantify cell-surface and total HS respectively (Supplementary Fig. 6A, B). In agreement with a previous report[39], *SLC39A9* and *TM9SF2* deficient cells exhibited strongly decreased N-sulfated HS signals, which was rescued for *SLC39A9*. *VPS35* heterozygous cells (homozygous VPS35 KOs were lethal) had increased levels of HS in permeabilized conditions, compatible with its effect on the uptake of α-syn, tau, and Aβ proteinaceous aggregates. *C3orf58* had surprising effects on the N-sulfated HS signal. 10e4 signal was not affected by the loss of *C3orf58*, but its over-expression in the rescue line resulted in decreased signals (Supplementary Fig. 6A, B). As stated earlier, investigation of a single epitope in the complex HS molecules is insufficient to conclude on the expression and modification of HS. Nevertheless, our results clearly establish that manipulation of expression of *SLC39A9, TM9SF2, C3orf58* and *VPS35* perturbed N-sulfated HS signals to varying degrees, suggesting that they might all act on PFFs accumulation by modifying a major receptor for α-syn PFFs, HS.

### SLC39A9-mediated $Zn^{2+}$ transport, but not C3orf58 kinase activity, is necessary for α-syn-PFFs accumulation

Because *C3orf58* and *SLC39A9* deficiency specifically reduced the accumulation of α-syn PFFs but not that of other cargoes, including Tau oligomers, we decided to study more in detail the mode of action of these two genes. SLC39A9 (also known as Zip9) is a $Zn^{2+}$ transporter that was previously proposed to export $Zn^{2+}$ from the Golgi lumen to the cytosol[49,50]. C3orf58 is a predicted transmembrane protein, and a member of the FAM69 family of secreted kinases[51] even though its kinase activity has not been confirmed to date. Using stable and constitutive lentivirus-mediated expression of HA-tagged wild-type (WT) SLC39A9 or C3orf58 constructs, we generated rescue lines. C3orf58 possesses a putative transmembrane domain in its N-terminal part (Fig. 5A), which was consistent with its insolubility in a sodium carbonate extraction assay (Supplementary Fig. 7A). Both proteins were localized to the Golgi apparatus (Supplementary Fig. 7B). To test the importance of $Zn^{2+}$ transport and putative kinase activity in mediating the effects of respectively SLC39A9 and C3orf58 on α-syn PFFs accumulation, we generated mutant constructs by altering residues predicted to disrupt these respective functions. To do so, we used AlphaFold for homology-based 3D structural modeling to identify key residues for the specific predicted activities of both proteins. The D159 and H185 residues of SLC39A9 are predicted to coordinate the $Zn^{2+}$ ion as it reaches the exit of the channel and mutating these two amino acids should thus abolish the transport activity of SLC39A9 (Fig. 5). Key residues in the putative active site of C3orf58's kinase domain were also identified: D306 and K198 are predicted to be implicated in ATP-$Mg^{2+}$ binding, while A287 is predicted as the catalytic base that removes a proton from the nucleophilic group

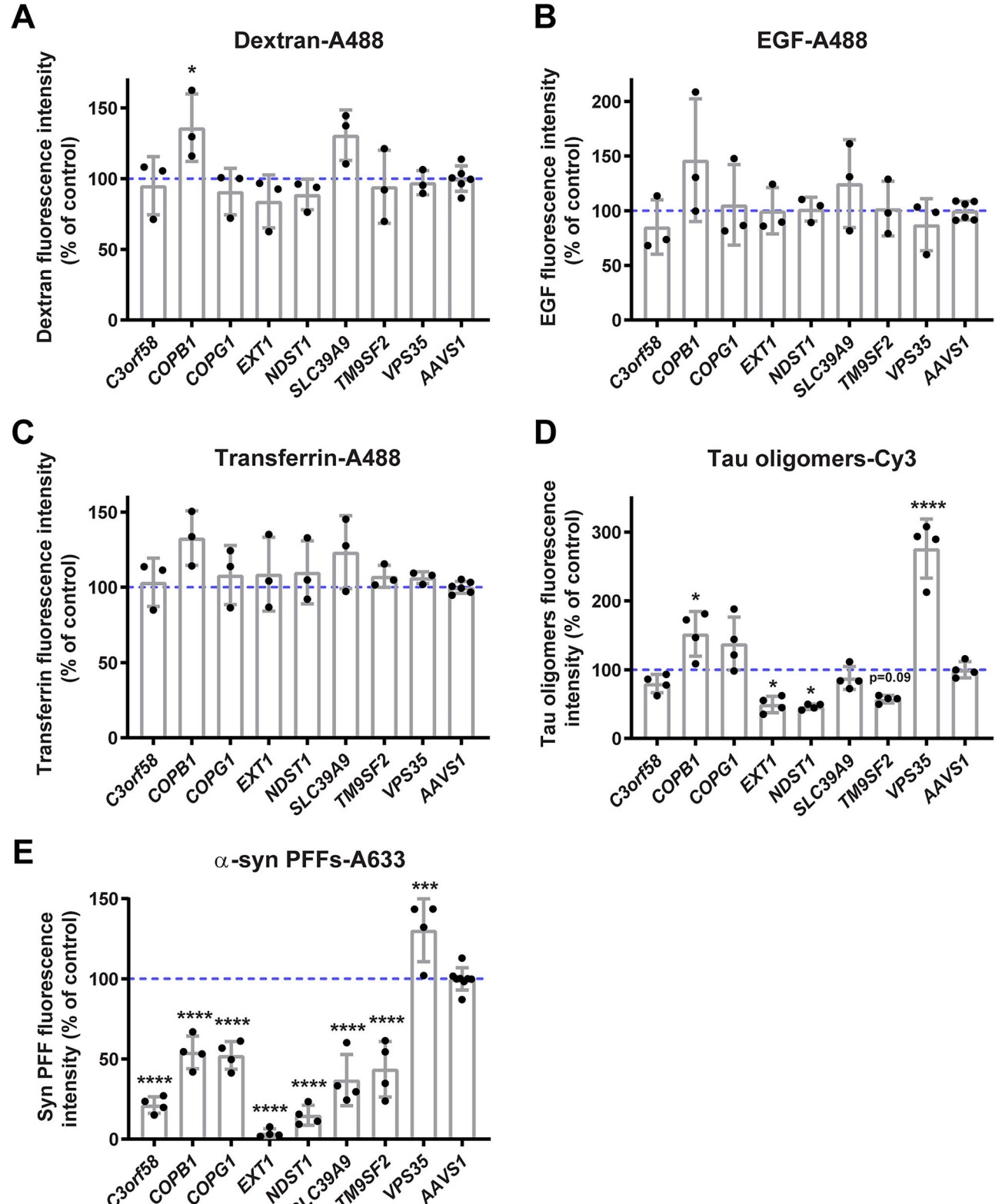

**Fig. 3 | Cargo specificity of validated hits.** **A–E** RPE-1 cells stably expressing Cas9 nuclease were transfected in 96 wells plates with individual synthetic sgRNAs, and after 3 days, were subjected to 24 h uptake of the following cargoes: **A** dextran-Oregon Green 488, **B** EGF-A488, **C** transferrin-A488, **D** Tau oligomers-Cy3, and **E** α-syn-PFFs-A633 as controls of sgRNAs' effects. Cells were then fixed, and nuclei were stained with Hoechst 33342. Imaging was done using a CX7 high-content microscope and quantification with the HCS Studio Cell Analysis software. The mean total fluorescence intensity per cell was measured for each cargo. Images are shown in Suppl. Fig. 4A. Only Tau oligomers showed important changes in uptake upon invalidation of some hits, especially *EXT1*, *NDST1*, and *TM9SF2* which are known to affect heparan sulfate expression, and *VPS35* which was previously reported to increase the uptake of proteinaceous aggregates. Intriguingly, *C3orf58* and *SLC39A9* did not significantly decrease Tau oligomers-uptake. $n = 3–4$ independent experiments. For the *AAVS1* negative control, two separate sgRNAs were used. Graph bars represent mean ±s.d. Statistical test: one-way ANOVA: *$p < 0.05$, ***$p < 0.001$, ****$p < 0.0001$.

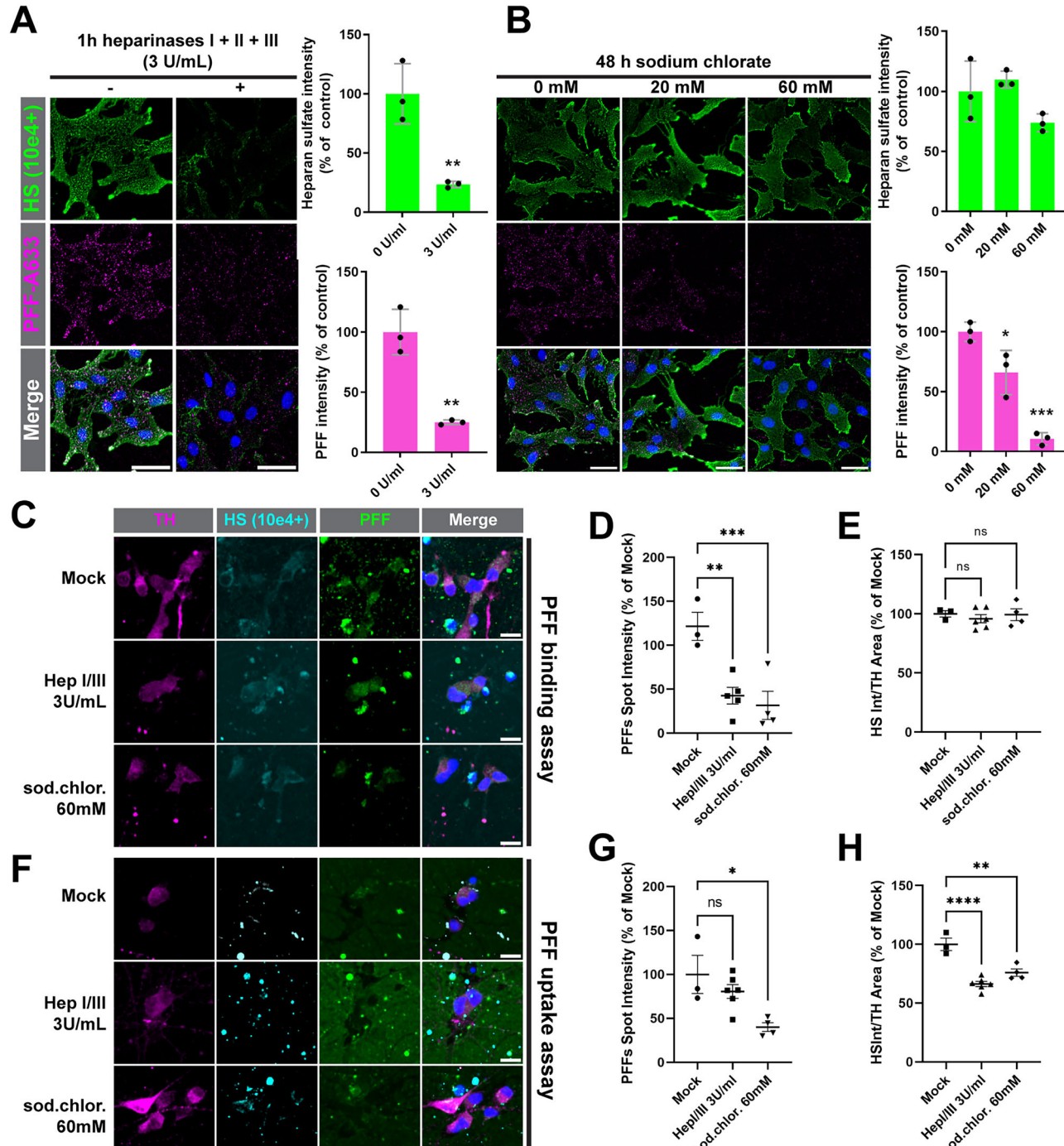

**Fig. 4 | HSPGs are major receptors for α-syn PFFs binding on the cell surface.**
**A**, **B** PFFs-A633 (magenta) binding and surface HS immunostaining (10e4 epitope, green) were performed on ice in live WT RPE-1 cells before fixation and nuclei staining with Hoechst (blue). Before the binding assay, cells were pre-treated with a combination of heparinases I, II, and III for 1 h (**A**) or with sodium chlorate for 48 h (**B**) at the indicated concentrations. Representative epifluorescence images at 20x are shown, and quantifications performed with ImageJ are shown on the right. Bar graphs are shown as mean ± SD percent of mock-treated cells. $N = 3$ independent experiments. PFFs binding (**C**) or uptake (**F**) assays in iDNs. iDNs pre-treated with

heparinases I + III (1 h) or sodium chlorate (48 h) were incubated on ice for 20 min (**C**) or at 37 °C for 3 h (**F**) in presence of PFFs-A488 (green) and anti-HS 10e4 antibody (cyan) before fixation and immunofluorescence against TH (magenta). Nuclei were stained with Hoechst (blue). PFFs (**D**, **G**) and HS (**E**, **H**) signal intensities in TH-positive areas were quantified and normalized to Mock treated iDNs. Data is presented as mean ±s.d. Statistical tests: (**A**) unpaired t-test, (**B**, **D**, **E**, **G**, **H**) one-way ANOVA; *$p < 0.05$, **$p < 0.01$, ***$p < 0.001$, ****$p < 0.0001$. Scale bars: 50 μm (**A**, **B**); 10 μm (**C**, **F**).

(Fig. 5C). We generated HA-tagged SLC39A9 D159A and H185R constructs as well as a K198A-D287N-D306N mutant (3MUT) version of C3orf58 and used these to rescue *SLC39A9^-/-* and *C3orf58^-/-* cells. The correct Golgi-localization of the wildtype and mutant proteins in the rescue lines

were confirmed using immunofluorescence (Supplementary Fig. 7B). Following 24 h of treatment with fluorescent α-syn PFFs, PFFs accumulation could be rescued to the levels of WT cells upon re-expression of the WT proteins (Fig. 5D). Putative kinase-dead (3MUT) C3orf58 also rescued PFFs

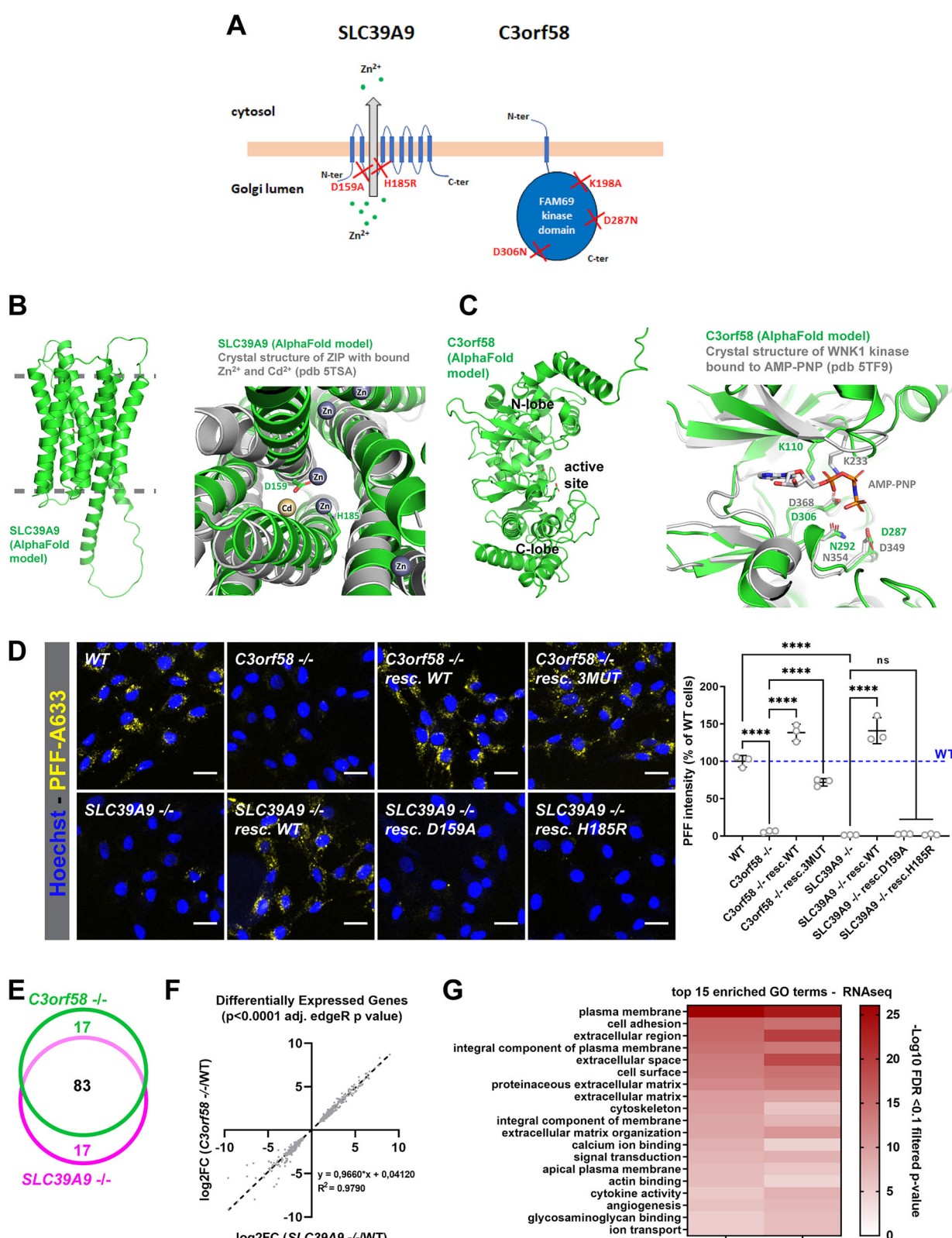

accumulation almost to WT levels, but less efficiently than its WT counterpart. This difference might rather be explained by slightly lower expression levels and/or modified glycosylation (Supplementary Fig. 7C, two of the 3 mutations add asparagines as potential N-glycosylation sites). This observation also suggests that if C3orf58 does have a kinase activity, it does not play a major role in its ability to facilitate PFFs accumulation.

$SLC39A9^{-/-}$ cells rescued with SLC39A9 D159A or H185R mutants were unable to accumulate more PFFs than $SLC39A9^{-/-}$ cells (Fig. 5D). This is unlikely to be due to the lower expression levels of D159A and H185R rescue constructs compared to the WT (Supplementary Fig. 7C, D), as no increase in PFFs signal compared to $SLC39A9^{-/-}$ cells could be detected at all. This is consistent with the inability of these two mutants to rescue the accumulation

**Fig. 5 | Structure/function study of C3orf58 and SLC39A9. A** Schematic representation of the topology of SLC39A9 and C3orf58 in the Golgi membrane. SLC39A9 is a $Zn^{2+}$ transporter that exports $Zn^{2+}$ from the Golgi lumen to the cytosol, whereas C3orf58 is a putative kinase of the FAM69 family. Designer mutations were engineered in both proteins (red crosses) to abolish the putative function of both proteins. For C3orf58, a triple mutant version of the protein was designed. **B** Left: global structure of SLC39A9, predicted by AlphaFold2 (AF-Q9NUM3-F1). The approximate position of the membrane bilayer is shown in grey. Right: structural superposition of SLC39A9 and the homologous zinc transporter ZIP bound to cadmium and zinc ions. Important residues for metal transport are shown as sticks. **C** Left: global structure of C3orf58, predicted by AlphaFold2 (AF-Q8NDZ4-F1). The protein harbors two lobes typically found in kinases, with a nucleotide-binding active site at the interface. Right: structural superposition of C3orf58 and the homologous kinase WNK1 bound to AMP-PNP. Important residues located around the active site are shown as sticks. **D** 24 h uptake of 60 nM PFFs-A633 (yellow) in monoclonal C3orf58^{-/-} or SLC39A9^{-/-} RPE-1 cells, either in the absence or presence of

stable expression of the indicated C-terminally HA-tagged rescue constructs. Nuclei were stained with Hoechst, and the mean PFFs-A633 fluorescence intensity was quantified by high-content microscopy and normalized to WT RPE-1 cells (see graph, right panel). Data is presented as mean ±s.d. Scale bars: 30 μm. $n = 3$ independent experiments. Statistical test: one-way ANOVA; ****$p < 0.0001$. RNA-seq analysis was performed to investigate transcriptomic changes in SLC39A9^{-/-} and C3orf58^{-/-} cells at steady states, compared to WT RPE-1 cells ($n = 3$ biological samples per genotype). The edgeR package was used to perform differential expression and gene ontology analysis. **E** Venn diagram showing overlap between the top 100 differentially expressed genes in C3orf58^{-/-} and SLC39A9^{-/-}, compared to WT cells. **F** The magnitude of changes in gene expression were strikingly similar in both lines ($R^2 = 0.9790$ for genes with $p < 0.0001$ common to both lines). **G** The top 15 GO terms of differentially expressed genes are shown. Of note, these 15 terms were identical between the two genotypes, indicating highly similar changes at the plasma membrane/cell surface in both KO lines.

of $Zn^{2+}$ in the Golgi apparatus observed in SLC39A9^{-/-} cells, in contrast to the WT rescue line (Supplementary Fig. 8). These results suggest that $Zn^{2+}$ homeostasis in the Golgi is important for PFFs accumulation. Note that in C3orf58^{-/-}, TM9SF2^{-/-} and VPS35 heterozygous monoclonal cells, Golgi $Zn^{2+}$ levels were unaffected, suggesting that they do not exert their role on α-syn PFFs accumulation by modulating SLC39A9's function. Finally, Tau oligomers accumulation was unaffected in monoclonal C3orf58^{-/-} and SLC39A9 -/- cells (Supplementary Fig. 9), and α-syn PFFs accumulation was decreased in TM9SF2^{-/-} and increased in VPS35 heterozygous monoclonal cells (Supplementary Fig. 6C), confirming our results on polyclonal KO populations (Fig. 2B, Fig. 3E).

### C3orf58 and SLC39A9 regulate binding of α-syn-PFFs to the cell surface

To further investigate the role of SLC39A9 and C3orf58 in mediating α-syn PFFs accumulation, we compared the transcriptome of both KO lines to that of WT cells using RNAseq (Fig. 5E–G, Supplementary Fig. 10, Tables S7, S8). Strikingly, both KO lines showed highly similar transcriptomic changes, sharing 83% of their top 100 differentially expressed genes (Fig. 5E), and 606 in total (Supplementary Fig. 10A–C) with highly comparable magnitude of changes (Fig. 5F, correlation coefficient $R^2 = 0.979$). Accordingly, PCA and clustering analysis indicated a clear distinction between KO and WT samples, with both KO genotypes indistinguishable from each other (Supplementary Fig. 10D, E). The top 15 affected GO terms based on differential gene expression were also shared (Fig. 5G). This indicated that these two genes might act in (a) similar(s) pathway(s) mediating PFFs accumulation. The GO analysis pointed towards major changes at the plasma membrane and in the extracellular matrix (Fig. 5G), suggesting that the uptake/endocytosis of α-syn PFFs might be impaired by loss of SLC39A9 and C3orf58, rather than PFFs clearance being increased. We thus monitored PFFs uptake at early time points (15 min, 2 h) in WT, C3orf58 -/-, SLC39A9 -/-, and rescued cells using fluorescence microscopy. The trypsinization step preliminary to cell fixation allows the removal of PFFs that are bound to the cell surface but not yet internalized (Supplementary Fig. 11A, PBS wash vs Trypsin wash). Image analyses revealed that both the percentage of PFFs-positive cells and the median fluorescence intensity were severely reduced in C3orf58^{-/-} and SLC39A9^{-/-} cells, which could be rescued by WT constructs, slightly less by kinase-dead C3orf58, and not at all by $Zn^{2+}$ transport-deficient SLC39A9 H185R (Supplementary Fig. 11B, C, D). These results clearly establish that C3orf58 and SLC39A9 deficiencies reduce PFFs uptake by impairing early steps of internalization. Since RNAseq indicated that the plasma membrane composition might be altered in C3orf58^{-/-} and SLC39A9^{-/-} cells, we investigated whether the binding of α-syn PFFs to the cell surface was affected (Fig. 6A, B). PFFs binding was greatly reduced by loss of C3orf58 or SLC39A9 and rescued by the expression of WT proteins but not the SLC39A9 H185R mutant. Taken together, our transcriptomic data suggest that C3orf58 or SLC39A9 deficiency may significantly alter the plasma membrane and

extracellular matrix composition, leading to decreased binding of α-syn PFFs to the cell surface.

We then tested whether this could be due to changes in specific proteins exposed at the cell surface of KO cells by surveying the surface proteome. We used a biotin-containing compound to label surface proteins (Supplementary Fig. 12A, B)[52], and identified labeled proteins by streptavidin pulldown followed by LC-MS/MS (Table S9). No significant changes were observed in the surface proteome of C3orf58^{-/-} and SLC39A9^{-/-} cells, nor in VPS35 heterozygous cells (Supplementary Fig. 12C, D, F). TM9SF2^{-/-} cells had elevated levels of NID2 (nidogen-2) and LOXL1 (lysyl oxidase homolog 1) (Supplementary Fig. 12E), two proteins the homolog of which may interact together in other species[53,54] and regulate cell-extracellular matrix interactions[55]. Since TM9SF2 deficiency affected not only α-syn PFFs but also Tau oligomers uptake (Fig. 3D, E), we did not pursue this further. Together, our data suggest that C3orf58 and SLC39A9 deficiencies do not modulate α-syn PFFs binding by perturbing the cell surface proteome, although it cannot be excluded that lower abundance proteins not detected in our proteomics analysis are affected.

### C3orf58 and SLC39A9 regulate HS homeostasis

Several lines of evidence suggested that alteration of HS homeostasis might explain the decreased PFFs binding at the cell surface of SLC39A9 and C3orf58 deficient cells. First, HS seems to mediate most of PFFs binding to the cell surface in RPE-1 cells based on genetic evidence from the screen and loss of PFFs binding upon heparinases or sodium chlorate treatment (Fig. 4). Second, RNAseq indicated changes at the cell surface including "glycosaminoglycan binding" in both KOs (Fig. 5G). Third, post-fixation staining of HS expression estimated by IF against the 10e4 epitope showed decreased HS in SLC39A9^{-/-} cells and C3orf58 overexpressing cells (resc. WT line) (Supplementary Fig. 6A, B, Supplementary Fig. 21E).

To further characterize the regulation of HS expression by SLC39A9 and C3orf58, we performed surface HS labeling in parallel to the PFFs binding assay (Fig. 6). Quantification indicated a severe loss of sulfated HS signal in SLC39A9^{-/-} cells (Fig. 6A, C), which was rescued by SLC39A9 WT but not the H185R mutant. C3orf58^{-/-} cells, despite having lower PFFs binding capacity, displayed similar HS signals to WT cells (Fig. 6A, C). Surprisingly, upon rescue with C3orf58 WT, PFFs binding was restored despite an overall decreased 10e4 antibody reactivity and regardless of the highly variable N-sulfated HS signal in individual cells. In the C3orf58 3MUT rescue line, restored PFFs binding was not accompanied by a decrease in HS staining. Interestingly, at higher magnification, puncta of PFFs bound to the cell surface of C3orf58^{-/-} cells showed reduced colocalization with sulfated HS puncta, phenocopying sodium chlorate treatment (Supplementary Fig. 13) and confirming that sulfation of HS plays a key role in HS-PFFs binding (Fig. 4C) as previously reported[26,36]. Again, the C3orf58 3MUT rescue line behaved differently, with only a partial restoration of colocalization. Interestingly, the cell surface binding of GFP+, a recombinant version of GFP exposing positively charged residues to mediate its HS-

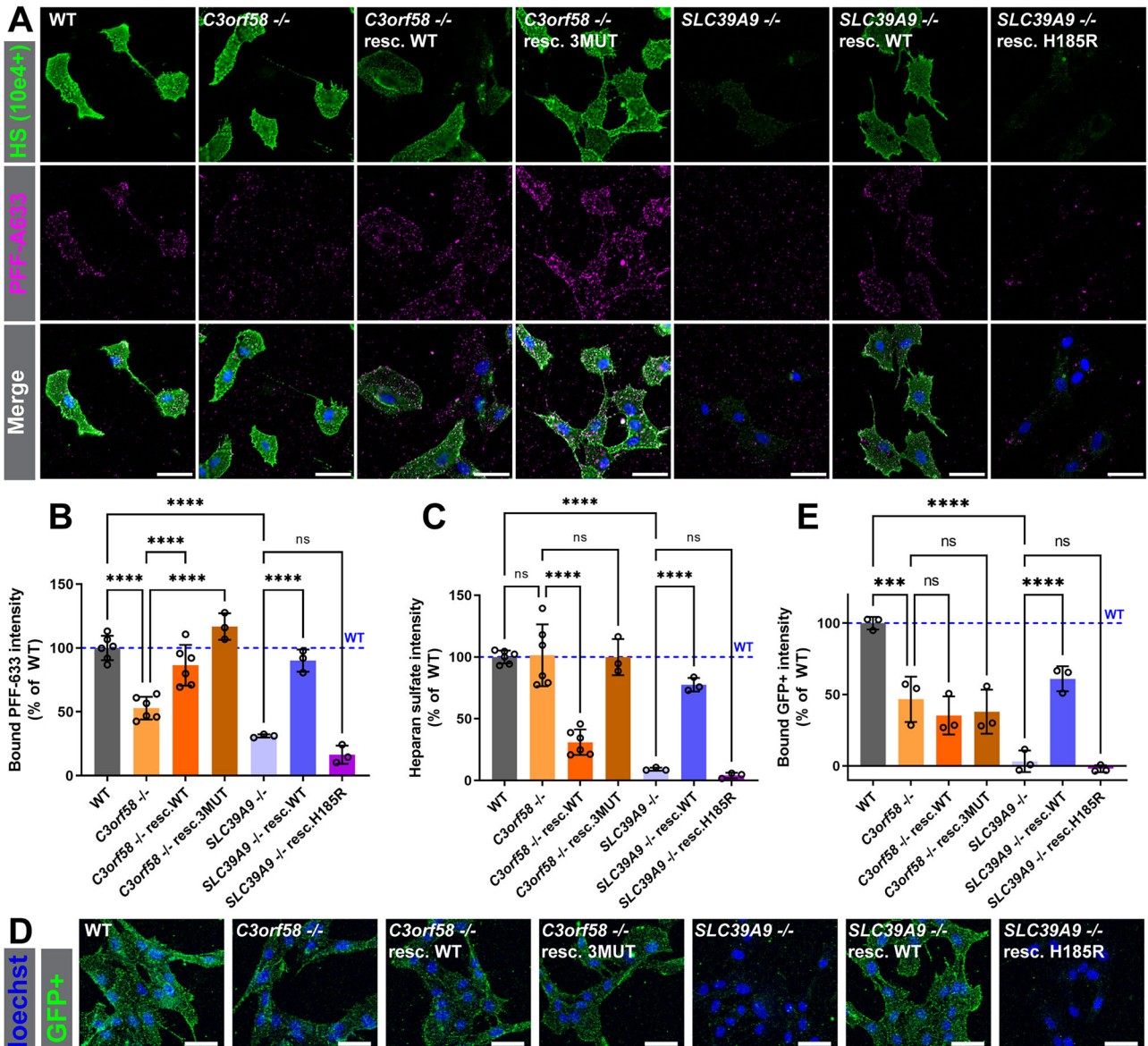

**Fig. 6 | PFFs binding to the cell surface is decreased in *C3orf58*⁻/⁻ and *SLC39A9*⁻/⁻ cells. A** PFFs-A633 (magenta) binding and surface HS immunostaining (green) was performed on ice in live WT, *C3orf58*⁻/⁻ or *SLC39A9*⁻/⁻ monoclonal RPE-1 cells (± indicated rescue constructs) before fixation and nuclei staining with Hoechst (blue). Mean HS (**B**), PFFs (**C**) signals per cell were quantified using ImageJ and are shown as percent of WT cells. n = 3-6 independent experiments. **D** GFP+ binding (green, quantified in **E**) was performed on ice before fixation and nuclei staining with Hoechst (blue). Scale bars: 50 μm. Graph bars represent mean ±s.d. Statistical test: one-way ANOVA; ***$p < 0.001$, ****$p < 0.0001$.

dependent cell surface binding and endocytosis[16], was significantly decreased as well in *C3orf58* -/- and *SLC39A9* -/- cells (Fig. 6D,E).

Our RNAseq analysis supported perturbed HS homeostasis in *C3orf58*⁻/⁻ and *SLC39A9*⁻/⁻ cells. Supplementary Fig. 14 shows significant modulations of some HS biology-relevant genes that were similar between the two lines, including changes in HS core proteins genes (*SDC3*, *GPC1*, *GPC6*) and HS biosynthetic enzymes (*EXT1*, *NDST1*, *NDST3*, *HS2ST1*, *HS3ST1*, *HS3ST3A1*, *HS3ST3B1*, *XYLT1*). An increase in the chondroitin sulfate core protein gene *CSPG4* indicated a possible more general dysregulation of glycosaminoglycan metabolism.

These data suggest that *SLC39A9* and *C3orf58* may both play a role in HS biosynthesis and/or modification, resulting in decreased PFFs and GFP + binding upon loss of either gene, but in a way that maintains tau oligomers uptake in both KO lines (Fig. 3D). Sulfated HS detection using the 10e4 antibody, a staining that is unlikely to recapitulate the complexity and diversity of HS chains, was also unchanged in *C3orf58*⁻/⁻ cells.

## Alterations in expression and composition of surface HS in C3orf58 and SLC39A9 deficient cells revealed by LC-MS/MS analysis

These results prompted us to analyze in more detail the HS expression and modification using LC-MS/MS to quantify HS disaccharides and tetrasaccharides species purified from WT, *C3orf58*⁻/⁻ and *SLC39A9*⁻/⁻ cells, and rescued counterparts. When combined, total HS species were significantly increased in *C3orf58*⁻/⁻ cells in a rescuable manner (Fig. 7A, B, E). This was true for many disaccharides and tetrasaccharides species, including the increase in the most abundant species (ΔUA-GlcNAc, ΔUA-GlcNS, Fig. 7F, G; see other species in Supplementary Fig. 15). On the other hand, total HS were slightly decreased in *SLC39A9*⁻/⁻ cells compared to WT, but not significantly ($p = 0.198$), which was also rescued (Fig. 7A, E). This was again driven by a significant decrease in the most abundant disaccharides exposed at the surface of WT RPE-1 cells (ΔUA-GlcNAc, ΔUA-GlcNS, ΔUA2S-GlcNS, Fig. 7A, B, F–H, as well as ΔUA2S-GlcNAc, Supplementary Fig. 15).

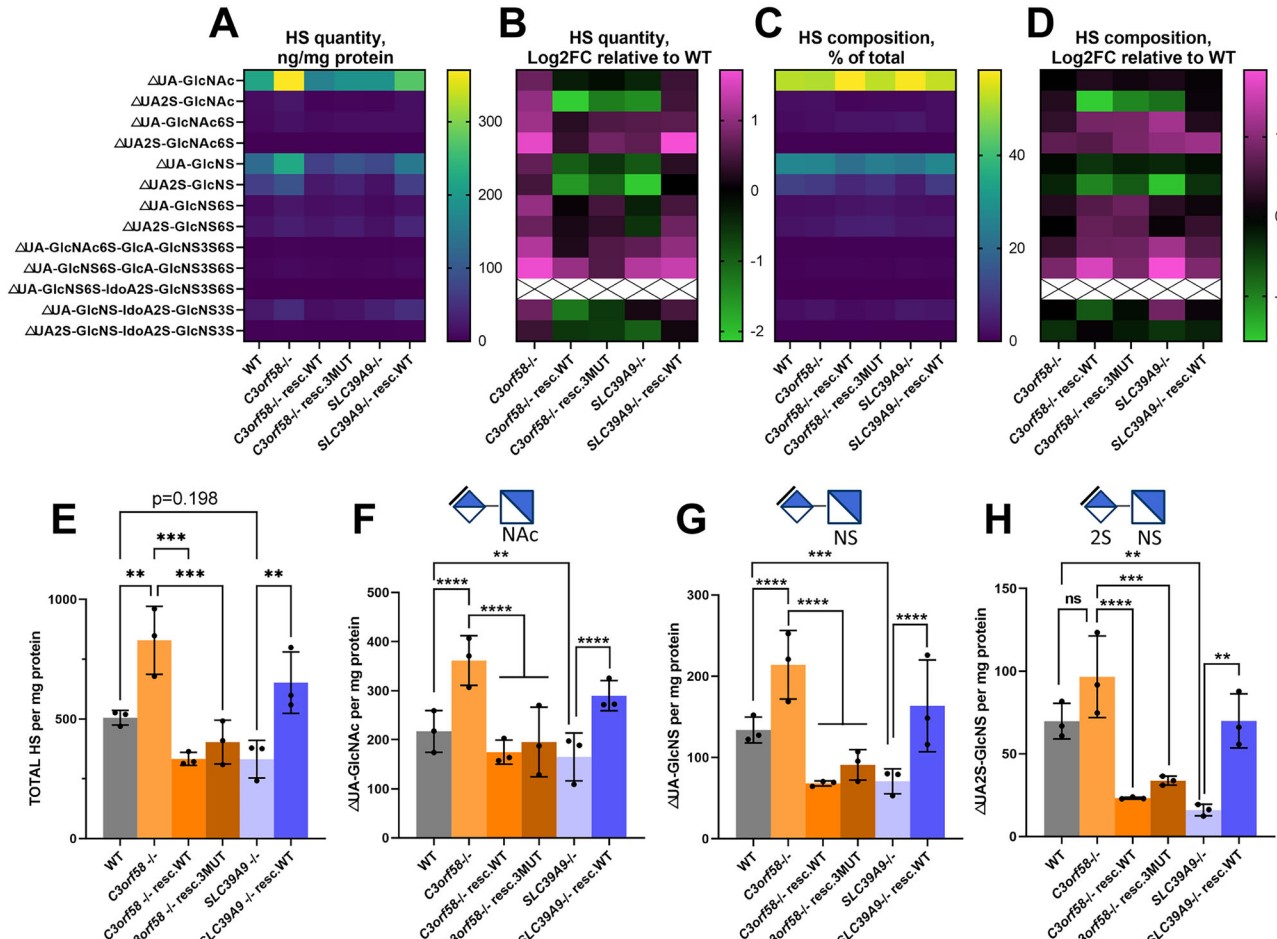

**Fig. 7 | *C3orf58* and *SLC39A9* regulate HS levels and composition in RPE-1 cells.**
**A–G** Cell surface HS from WT, *C3orf58*⁻/⁻ and *SLC39A9*⁻/⁻ monoclonal RPE-1 cells (± indicated rescue constructs) were harvested by trypsin treatment before isolation and quantification of indicated HS disaccharides and tetrasaccharides by LC-MS/MS using ¹³C-labeled internal standards. HS amounts were normalized to protein content from the cellular sample from which HS was isolated. *n* = 3 independent samples per genotype. For each disaccharide/tetrasaccharide analyzed, we show as heat maps: **A** normalized quantities (ng/mg protein), **B** Log2 fold-change of normalized quantities relative to WT cells, **C** HS composition (percentage of each species relative to total HS), **D** HS composition as Log2 fold-change relative to WT. Black crosses on white background indicate an absence of detection in our samples. Bar graphs showing normalized HS quantities of the indicated HS species: **E** total HS in samples, **F** ΔUA-GlcNAc, **G** ΔUA-GlcNS, **H** ΔUA2S-GlcNS. Graph bars represent mean ±s.d. Statistical tests: one-way ANOVA; **$p < 0.01$, ***$p < 0.001$, ****$p < 0.0001$. See supplementary Fig. 15 for all disaccharide and tetrasaccharide quantification data.

Beyond the absolute amount of total HS and individual species, their relative abundance in HS chains (i.e., HS composition) is an important parameter that determines ligand binding capacities[29]. Contrary to the significant increase in many HS species observed in *C3orf58*⁻/⁻ cells, their overall HS disaccharide and tetrasaccharide composition was unchanged except for a slight increase in ΔUA2S-GlcNAc (Fig. 7C, D; Supplementary Fig. 15). However, HS composition was strongly affected in C3orf58 rescued lines, consistent with a possible implication of this protein in HS homeostasis, as suggested by the decreased HS 10e4 signal in C3orf58 resc. WT cells and reduced PFFs binding in *C3orf58*⁻/⁻ cells (Fig. 6, Supplementary Fig. 6A, B).

*SLC39A9* -/- cell surface HS composition was severely affected. An increase, although not reaching significance, was observed for the unmodified ΔUA-GlcNAc disaccharide, while a decrease was observed for the other abundant disaccharides ΔUA-GlcNS, ΔUA2S-GlcNS (Fig. 7C, D; Supplementary Fig. 15). This pattern of HS composition with an increased ratio of unmodified disaccharides coupled to a decrease in N-sulfated species was highly similar to that observed in C3orf58 resc. WT cells and consistent with decreased 10e4 antibody signal in intact and permeabilized cells (Fig. 6, Supplementary Fig. 6A, B). Altogether, our analysis of cell surface HS abundance and composition reveals changes that can explain the binding of the 10e4 antibody, but not the PFFs binding phenotype. However, they also confirm that both C3orf58 and SLC39A9 are important

players in HS expression and composition, given that changes observed in the KO lines were rescued, and that overexpression of C3orf58 impacts HS composition.

ER-Golgi dynamics have been shown to regulate the localization of HS-modifying enzymes[56]. Thus we monitored whether Golgi structure was affected in *C3orf58*⁻/⁻ and *SLC39A9*⁻/⁻ cells by using a modified Golgi morphology analysis pipeline[57] (Supplementary Fig. 16). Golgi area was increased in both lines in a rescuable manner. Both cisGolgi and transGolgi were larger in *SLC39A9*⁻/⁻ cells, while only the cisGolgi increase in area reached significance in *C3orf58*⁻/⁻ cells (Supplementary Fig. 16B). Other parameters (compactness, cisGolgi and transGolgi markers signal intensities, and number of fragments) were unchanged (Supplementary Fig. 16C–F). Whether perturbed Golgi morphology is sufficient to explain changes in HS homeostasis and the PFFs uptake phenotype remains to be determined.

**A tight balance in NDST1 expression is needed for PFFs uptake, and *C3orf58* regulates NDST1 expression and localization**
To get a sense of the underlying causes of surface HS homeostasis perturbations in *C3orf58*⁻/⁻ and *SLC39A9*⁻/⁻ cells, we first assessed whether expression levels of the key biosynthetic enzyme NDST1 (which activity is thought to be a pre-requisite for further HS modification[58]) was affected in

these cells. Western blot analysis showed a 1.5-fold increase in total NDST1 levels, an observation in line with the RNASeq data (Supplementary Fig. 14) suggesting an accumulation of the enzyme and faster migration in lysates from *SLC39A9* KO cells, probably due to lesser N-glycosylation[59] (Fig. 8A). Beyond expression levels, regulation of HS enzymes trafficking is believed to modulate HS modification[56]. C3orf58 was previously shown to colocalize with COPI vesicles responsible for intra-Golgi trafficking[42] and its over-expression modulates secretory protein transit from the ER to the Golgi[60]. N- and O-glycosylation defects in *SLC39A9* cells[59] might affect the trafficking of glycosylated enzymes such as NDST1. We thus checked whether the localization of NDST1 was affected in *C3orf58*[-/-] or *SLC39A9*[-/-] cells. We did not manage to detect endogenous NDST1 by immunofluorescence, so we transduced cells with a doxycycline-inducible lentiviral cassette to express C-terminally HA-tagged NDST1 (shown to be functional previously[61]). Leaky expression of NDST1-HA was observed (Fig. 8B), which allowed us to monitor NDST1 localization while preventing saturation of trafficking machinery. Signal intensity from leaky expressed NDST1-HA in the Golgi of *C3orf58*[-/-] and *SLC39A9*[-/-] cells was similar to WT cells (Fig. 8B,C). However, the proportion of leaky expressed NDST1-HA localized in the Golgi was significantly higher in *C3orf58*[-/-] cells (Fig. 8B,D). An opposite, yet non-significant tendency was observed in *SLC39A9*[-/-] cells. Upon doxycycline induction, NDST1-HA levels increased similarly independent from genotype (Fig. 8B,C), yet showing low levels of expression relative to endogenous NDST1 (see absence of increase in total NDST1 in Supplementary Fig. 20). The proportion of NDST1-HA localized in the Golgi was still significantly higher in *C3orf58*[-/-] cells compared to WT (although to a lesser extent), and this was also the case for *SLC39A9*[-/-] cells (Fig. 8D). This suggested increased sorting and/or retention of NDST1 in the Golgi, at more physiological NDST1 expression levels upon C3orf58 deficiency and at higher expression levels in both genotypes. This is compatible with increased levels of surface HS in *C3orf58*[-/-] cells, including of N-sulfated disaccharides (Fig. 7G, H). Together, our analysis of NDST1 localization suggests that C3orf58 deficiency, and SLC39 A9 deficiency at higher NDST1 expression levels, perturb NDST1 trafficking through an unknown mechanism.

It is worth noting that NDST1-HA expression, either at low or high levels, did not rescue PFFs uptake in *C3orf58* -/- or *SLC39A9*[-/-] cells (Fig. 8B, E). Strikingly, we noticed that WT cells expressing high levels of NDST1-HA upon doxycycline treatment were unable to uptake PFFs (Fig. 8B, arrowheads). Because knock-out of NDST1 has a similar effect (Fig. 2B, Fig. 3E), we conclude that a balanced expression of NDST1 is crucial for proper PFFs uptake. It indicates that HS chains carrying too little or too much N-sulfation might both be incompatible with maintaining PFFs binding capacity.

### C3orf58 is an important player for PFFs uptake in human iPSC-derived microglia and dopaminergic neurons

Finally, we tested whether *C3orf58* and *SLC39A9* are important for α-syn PFFs uptake in a PD-relevant cell type. We generated *C3orf58*[-/-] and *SLC39A9*[-/-] human induced pluripotent stem cells (iPSC) using CRISPR/Cas9 (Fig. 9A, Supplementary Fig. 17A–C). Pluripotency, genome stability, and karyotype of the lines were verified (Fig. 9A, Supplementary Fig. 17D–F). These cells differentiated adequately into microglia (iMGL) as indicated by their morphology, qRT-PCR, and FACS-based quantification of microglial markers (Fig. 9A, Supplementary Fig. 18). WT, *C3orf58* -/- or *SLC39 A9* -/- iMGL were then treated with α-syn PFFs, and a trypan blue washing step allowed to visualize only internalized fibrils. Interestingly, *C3orf58* -/- reduced PFFs uptake by approximately 60% in iMGL, while *SLC39A9* -/- did not have a significant effect (Fig. 9B, C). Similarly, α-syn PFFs uptake experiment was conducted on iDNs derived from the *C3orf58* -/- and *SLC39A9* -/- iPSC lines (Fig. 9A). Similar to iMGL, *C3orf58* -/- iDNs also showed ~50-60% decrease in PFFs uptake in TH-positive iDNs with no effect on *SLC39A9* -/- (Fig. 9 D, E). These data demonstrate the importance of *C3orf58* for α-syn PFFs uptake in a PD-relevant cell type.

## Discussion

In this study, we sought to identify key molecular mechanisms and genes involved in the intracellular uptake of α-syn PFFs. We used a genome-wide unbiased approach in a neuroepithelial cell line, RPE-1 cells, and identified genes affecting HSPGs as major factors in α-syn PFFs binding to the cell surface and subsequent uptake in recipient cells. All the validated hits were specific for proteinaceous aggregates (α-syn PFFs or tau oligomers), with *C3orf58* and *SLC39A9* being highly specific to α-syn PFFs. These two genes could thus be of particular interest for the treatment of synucleinopathies, because targeting their activity might have less secondary effects than HSPG in general, given the many biological functions HSPG exert. In the initial screen, FACS sorting for enrichment of cells with higher or lower PFFs accumulation was performed solely based on labeled PFFs fluorescence, without accounting for cell size. In retrospect, we clearly showed that PFFs content was correlated to cell size and that silencing of genes that modify the growth or division of cells can be a major confounding factor in such FACS-based screens (Figs. 1, 2 and Supplementary Fig. 2). These data will help other researchers in the design of studies with similar approaches in the future, especially given the effect of CRISPR/Cas9 genome editing on cell cycle[62]. Another limitation in our study pertains to the shorter time allowed for obtaining a polyclonal perturbed cell population in our validation phase (to prevent over-confluency that would affect image analysis) compared to the screen itself, before PFFs treatment (3 days vs 9 days, respectively). Although applying Tracking of Indels by Decomposition (TIDE) to evaluate editing efficiency would help verify adequate gene disruption, it would not guarantee that a sufficient decrease at the protein level is achieved within the time limits. Thus, it is possible that our validation phase may contain false negatives that could have shown an effect at later time points. This includes genes that have previously been linked with our key identified pathways: *SLC10A7* deficiency alters HS expression[31], while *TMED10* has been attributed a role in unconventional targeting of proteins in the ER-Golgi intermediate compartment[63] and to act as a cargo receptor for COPI-mediated Golgi to ER retrograde transport[64]. *TMED10* has also been proposed to regulate macropinocytosis through an unclear mechanism[65], plasma membrane v-ATPase (several subunits of which were targeted in the low PFFs population of our screen) is also able to control macropinocytic events[66] and VPS35 can localize to maturing macropinosomes[67]. Macropinocytosis is involved in PFFs uptake in RPE-1 cells, since treatment with the macropinocytosis inhibitor EIPA strongly reduced PFFs accumulation (Supplementary Fig. 19), but also in other cell types including Hela cells, iPSC-derived astrocytes and dopaminergic neurons[19]. Validation in monoclonal KO cells of additional hits may reveal new players and better delineate their molecular interplay in key pathways necessary for PFFs uptake in the future.

For feasibility purposes, we used a 24 h PFFs treatment paradigm in the screen, which did not allow to directly conclude whether uptake or removal of PFFs was affected, and we thus use the term "accumulation" when appropriate. Quantification of early uptake events, before significant degradation by the cellular machinery can occur, and PFFs binding assays performed in the cold allowed to confirm the involvement of *C3orf58* and *SLC39A9* in the adhesion of PFFs to the cell surface. Given i) the magnitude of uptake inhibition in cells KO for either gene (Fig. 5D) and ii) the extremely similar changes in transcriptomic profiles in KO cells (Fig. 5E-G and Supplementary Fig. 10), this strongly suggests that both genes affect (a) similar pathway(s) the deregulation of which leads to largely decreased PFFs binding to the cell surface (Fig. 6). We provide several lines of evidence suggesting that HSPG biosynthesis and maturation are commonly regulated by both genes. First, the HSPG biosynthetic enzymes EXT1 and NDST1 were extremely significant hits in our screen (Fig. 1B) and their effect on PFFs uptake was very strong in the validation phase (Fig. 2B). Second, in RPE-1 cells, PFFs binding to the cell surface was very severely affected by pre-treatments that stripped off or decreased sulfation of HSPG (Fig. 4). Third, the Golgi PAPS transporters *SLC35B2*, shown previously to alter HS-dependent binding and uptake of PFF[16], was well scored in our screen (Table S1). Fourth, KO of *C3orf58* phenocopied sodium chlorate treatment

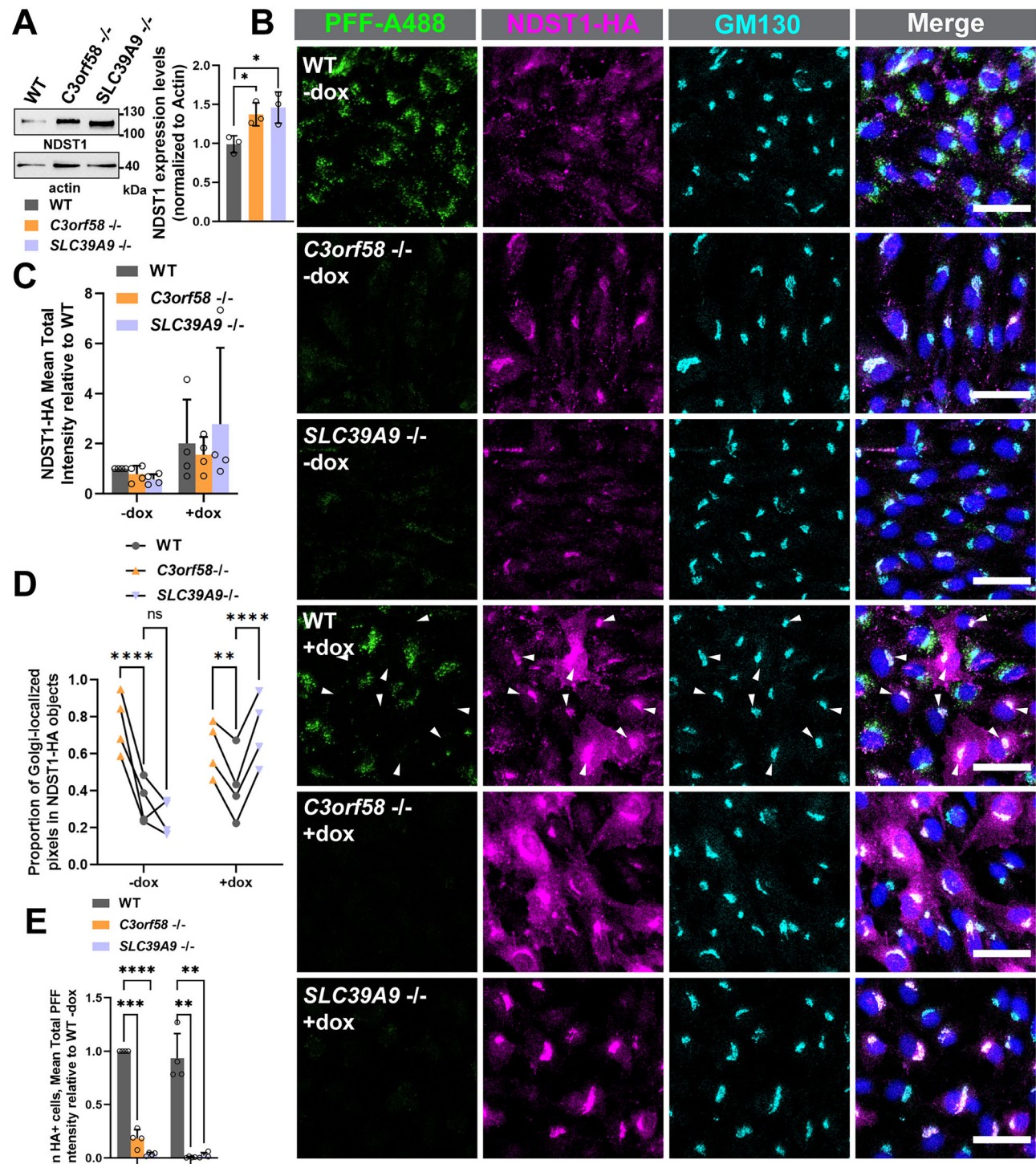

**Fig. 8 | NDST1 is mistrafficked in C3orf58[-/-] cells, and NDST1 overexpression inhibits PFFs uptake. A** Western blot analysis of endogenous NDST1 expression in WT, *C3orf58[-/-]*, and *SLC39A9[-/-]* monoclonal RPE-1 cells. *n* = 3 independent experiments. **B** PFFs-A488 (green) uptake in doxycycline-treated (25 ng/ml) WT, *C3orf58* or *SLC39A9* monoclonal KO RPE-1 NDST1-HA-inducible cells stained with anti-HA (magenta) for NDST1, anti-GM130 (cyan) for Golgi and Hoechst (blue) for nucleus. Scale bars: 20 μm. Correspond to indicated quantifications for HA-positive cells from **B**. **C** Mean total intensity of NDST1-HA in Golgi objects. **D** Proportion of Golgi-localized NDST1-HA pixels. **E** Total PFFs intensity in HA-positive cells is represented as fold change relative to WT, for both dox-treated and untreated conditions. *n* = 4 independent experiments Statistical test: unpaired student T-test (**A**), one-way ANOVA (**C,D,E**); ns, non-significant; *, *p* < 0.05; **, *p* < 0.01; ***, *p* < 0.001; **** *p* < 0.0001. Bar graphs represent mean ± SD.

on HSPG signals and the colocalization between HSPG and PFFs bound to the cell surface (Figs. 4, 6; Supplementary Fig. 13A, B). Fifth, both *C3orf58* and *SLC39A9* KO lines displayed changes in HSPG-related genes (Supplementary Fig. 14, that may arise in response to HS dyshomeostasis) and perturbations in HS quantity or composition at the cell surface (Fig. 7,

Supplementary Fig. 15). Finally, *C3orf58* and *SLC39A9* have both been identified in genome-wide CRISPR screens as cellular factors regulating infection by viruses that depend on HS for cellular entry (*C3orf58*, Zika virus[43]; *SLC39A9*, Chikungunya virus[39]), and C3orf58 was recently shown as

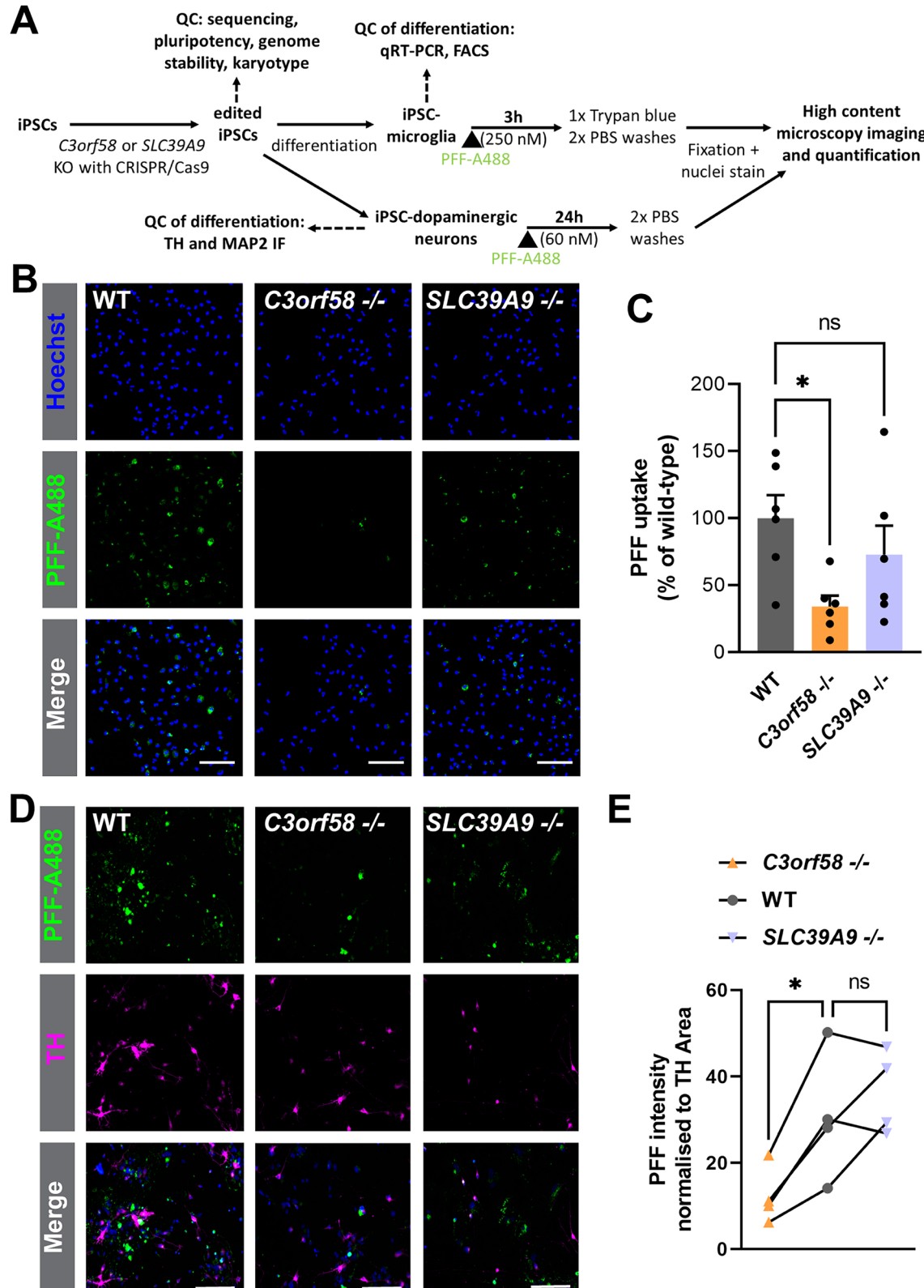

**Fig. 9 | *C3orf58* is important for PFFs uptake in human microglia and dopaminergic neurons. A** Pipeline for generation of WT, *C3orf58* -/- and *SLC39A9* -/- iPSC-derived microglia (iMGL) and dopaminergic neurons (iDNs) used for performing α-syn PFFs uptake assay. Quality control of editing, genome stability, and pluripotency is described in Supplementary Fig. 17, and quality control of differentiation of the various lines in iMGL is shown is Supplementary Fig. 18. **B** Representative images of α-syn PFFs uptake assay in iMGL of the indicated genotypes (scale bars: 250 μm), and **C** corresponding quantification showing a 60% reduction

of α-syn PFFs uptake in *C3orf58* -/- iMGL compared to WT. Mean ± SEM of n = 6 independent experiments. Statistical test: Kruskal-Wallis; * $p < 0.05$.
**D** Representative images of α-syn PFFs uptake assay in iDNs of the indicated genotypes (scale bars: 150 μm), and **E** corresponding quantification showing a 60% reduction of α-syn PFFs uptake in *C3orf58* -/- iDNs compared to WT. n = 4 independent experiments. Statistical test: repeated measures one-way ANOVA; * $p < 0.05$.

a key player in binding of cell-surface RNAs to the membrane, an HS-dependent process[68].

We observed that Tau oligomers and α-syn PFFs competed with one another for uptake in co-treatment experiments (Supplementary Fig. 4B). Our analysis only allows limited conclusions, since the similar concentrations (monomer-equivalent) of aggregates (80 nM for Tau, 60 nM for α-syn) used result in much more aggregates units of the smaller Tau oligomers compared to the larger α-syn PFFs. Nevertheless, these results are consistent with both aggregate types depending on HS for cell surface binding and uptake. In contrast to the binding of tau aggregate to HS which relies on specific N- and 6-O sulfation, it appears that no specific sulfate moiety, but rather global sulfation levels, are necessary for α-syn PFFs binding and uptake[26,36]. We confirm these findings since we did not find specific changes in the di-/tetra-saccharide unit at the cell surface of *C3orf58* -/- and *SLC39A9* -/- cells that correlated with changes in α-syn PFFs binding/uptake (Fig. 7, Supplementary Fig. 15). The observation that KO of *NDST1* greatly decreases PFFs uptake (Fig. 2B, Fig. 3E) is insufficient to conclude that N-sulfation is the sole necessary modification for HS/PFFs interaction. Indeed, NDST1 activity adds an N-sulfate group that favors higher overall sulfation by facilitating the action of most other HS modification enzymes[69]. Despite KO of *C3orf58* and *SLC39A9* having opposite effects on global HS levels in our LC-MS/MS analysis, they had similar effects on binding of HS ligands (PFFs and GFP+) to the cell surface, strongly indicating that both genes regulate HS homeostasis, although the specific perturbations of HS chains require additional analyses especially for *C3orf58*. HS di-/tetra-saccharide composition is likely not providing the adequate resolution of HS determinants for α-syn binding, as changes in total amount but not composition were observed at the surface of *C3orf58* -/- cells. HS chains are arranged as alternating domains of varying numbers of disaccharides in length, either enriched in N-sulfated (NS, 6 to 16 units) or N-Acetylated (NAc, up to 18 units), or containing a mix of NAc/NS. HS-binding proteins can bind one or two domains simultaneously[29]. The contribution of more complex parameters (length or numbers of HS chains, or length, numbers and distribution of high-/low-sulfation domains along the chains) to PFFs binding remains to be determined. Future analyses using synthetic, structurally defined oligosaccharides, or investigating the domains organization are likely to uncover the specific arrangements that are necessary to allow α-syn PFFs binding to HS chains. Even though we cannot exclude that one or several proteins might be needed as secondary receptors, our data suggests that at least in RPE-1, HS are the major receptor for α-syn PFFs binding to the cell surface.

*C3orf58* and *SLC39A9* may act as general regulators of glycans/glycosaminoglycans biosynthesis and/or maturation in the Golgi, which could have implications in several HS-related diseases or infections. This would be consistent with the previously reported decrease in complex-type N-glycans and O-glycans (due to decreased C1GalT1 expression) in *SLC39A9* -/- cells[59]. B4GalT4 and C1GalT1 are both $Mn^{2+}$-dependent galactosyl transferases, a class of enzymes whose activity is decreased upon increased $Zn^{2+}$/$Mn^{2+}$ ratio[70]. $Zn^{2+}$ levels are increased in the Golgi of *SLC39A9* -/- cells (Supplementary Fig. 8 and ref. 49), possibly explaining the effects on N- and O-glycosylations[59]. More generally, SLC39A9 may act as a general regulator of the activity of Golgi-resident, $Mn^{2+}$-dependent enzymes, by decreasing the local $Zn^{2+}$ concentration. Thus, C3orf58 and SLC39A9 could both play a role in regulating glycans biosynthesis, including glycosaminoglycan-like HS and chondroitin sulfate (CS), thereby shaping a cell surface environment that is competent for α-syn PFFs binding and uptake. This is compatible with the observed changes in the expression of HS and CS-related genes in our transcriptomic analysis (Supplementary Fig. 14). We showed that *C3orf58* -/- cells appear to have altered NDST1 trafficking (Fig. 8), and it is possible that other HS biosynthetic enzymes may also be affected. HS enzymes are subjected to both anterograde and retrograde trafficking via COPII and COPI vesicles respectively. Brefeldin-A treatment, which inhibits COPI-mediated retrograde transport, causes mislocalization of HS-modifying enzymes (including NDST1) from the cis- to the transGolgi[56]. Given the previously reported colocalization of C3orf58 with the β-COP

subunit of the COPI, it can be hypothesized that retrograde trafficking is compromised in *C3orf58* -/- cells. Altered trafficking of glycosyltransferases including HS biosynthetic enzymes may also occur in *TM9SF2* -/- cells, since TM9SF2 was previously shown to affect HS surface expression by affecting proper localization and stability of NDST1[39], possibly explaining its effect as a Zika virus host factor[71]. Whether trafficking defects of HS biosynthetic enzymes in *C3orf58* -/- and/or *SLC39A9* -/- cells are at the basis of their role in HS homeostasis remains to be determined.

A limit of our study is the use of RPE-1 cells as the primary model, which are likely not recapitulating the HS-related biology of more PD-relevant cell types (e.g., expression levels, modifications, trafficking). This is illustrated by unperturbed PFFs uptake in SLC39A9-deficient human iPSC-derived microglia and dopaminergic neurons (Fig. 9), which could be due to expression of redundant Golgi-localized $Zn^{2+}$ transporters (e.g., SLC39A7, SLC39A13). Nevertheless, our initial focus on RPE-1 led us to identify C3orf58 as a key factor for PFFs uptake in human dopaminergic neurons and microglia, suggesting that the RPE-1 model remains useful for studying how C3orf58 regulates HS homeostasis in the future, in the context of α-syn spreading but also regarding fundamental aspects of HS biology. Interestingly, despite a report that a murine microglial cell line (BV-2) only slightly relies on HSPG for α-syn PFFs uptake, with heparinase treatment only decreasing uptake by 25%[26], we have found that loss of the HSPG-regulating *C3orf58* gene in human iPSC-derived microglia decreased PFFs uptake by 60% (Fig. 9). This difference might be due to the various origins of microglial cells used, or to the possibility that *C3orf58* might regulate additional processes in microglia besides HSPG homeostasis. For instance, the function or expression of MerTK, a surface receptor tyrosine kinase necessary for non-inflammatory homeostatic phagocytosis in microglia that is necessary for efficient α-syn PFFs uptake in human microglia[72], might be altered upon *C3orf58* deficiency. Alternatively, an interplay between MerTK and HSPGs can be envisioned, since HSPGs have been reported previously to act as accessory molecules to influence receptor tyrosine kinase activation[73]. Validation of *C3orf58* as a gene regulating the spread of α-syn aggregation in vivo or using human organotypic in vitro models will be an important milestone for further demonstrating its therapeutic potential. More generally, the use of such complex systems to study if and how targeting HS is a promising avenue for stopping the progression of synucleinopathies. Because of the broad importance of HS in many biological processes, such studies will likely require precise spatio-temporal control over experimenter-induced HS perturbations, and a deeper understanding of mechanisms governing HS-PFFs interactions and organization of HS chains in general.

## Material and Methods
### Production of lentiviral particles and transduction
For lentiviral production, 293 T cells were transfected with constructs of interest (in the pLJM1 plasmids, lentiCas9-Blast plasmid, or pCW57.1 NDST1-HA plasmid) together with packaging plasmids (pMD2.g, pMDLg/pRRE, pRSV-Rev) using the Calcium Phosphate method. A medium change with fresh medium supplemented with non-essential amino acids mix (Wisent) was performed after 8 hours of transfection. The lentivirus-containing supernatant was harvested 48 hours post-transfection, filtered with a 0.45 μm filter, and recipient RPE-1 cells were transduced with a range of virus quantity before selection with 15 μg/ml puromycin (or 2 μg/mL blasticidin for Cas9). Wells where complete selection resulted in 70-90% cell death compared to untransduced control were chosen to ensure that most surviving cells statistically received only one lentiviral copy. These were kept as stable cell lines.

### Cell culture
Retina Pigmented Epithelial (RPE-1) and HEK 293 T cells were maintained in DMEM containing 4.5 g/l glucose, sodium pyruvate, 10% fetal bovine serum, and penicillin/streptomycin. Cells were passaged by rinsing in PBS followed by incubation in trypsin/EDTA (Wisent, 325-542-EL) at 37 °C for approximately 5 minutes before resuspension in complete medium to

inactivate trypsin. Cells were counted with a Luna cell counter (Logos Biosystems). The cell suspension was then seeded in recipient culture vessels at the desired cell density. All transgenes expressed in RPE-1 cells were delivered by lentiviral transduction. The macropinocytosis inhibitor EIPA (A3085, 5-(N-Ethyl-N-isopropyl)amiloride, Sigma) was dissolved in methanol, which was used as a vehicle control. Alexa488 EGF complex (E13345, ThermoFisher) was used at 120 ng/mL, Alexa488-conjugated transferrin (T13342, ThermoFisher) was used at 20 µg/mL, and Oregon Green 488-conjugated dextran (70,000 MW, D7172, ThermoFisher) was used at 100 µg/mL.

### iPSC culture and generation of *C3orf58* -/- or *SLC39A9* -/- iPSC

iPSC cultures were grown at 37 °C, 5% $CO_2$ for 2 weeks in supplemented mTeSR medium (STEMCELL Technologies #85850). For KO of *C3orf58* or *SLC39A9* in iPSC, iPSC cultures were grown at 37 °C, 5% CO2 for 2 weeks in supplemented mTeSR medium. Cells were then nucleofected with two synthetic guide RNAs per targeted gene (Synthego) as indicated in Supplementary Fig. 18A and Table S4, as described previously[74]. Polyclonal KO cell populations were recovered and genes KO were confirmed by genomic DNA extraction and PCR. To obtain monoclonal lines, polyclonal cell populations were dissociated and individual cells were sorted by flow cytometry and plated as single cells. Single clones were screened for gene KO by PCR and sequencing (Supplementary Fig. 18B, C), and pure monoclonal cultures were selected, amplified, and stored for further use. Staining for pluripotency markers, qPCR for genome stability assessment, and karyotyping were performed as described previously[75]. Approval for working with iPSCs for this project was obtained through the McGill University Health Centre Research Ethics Board (DURCAN_IPSC / 2019-5374).

### Differentiation of iPSCs into microglia (iMGL), quality control, and PFFs uptake

Differentiation of iPSCs into iMGL was carried out following a previously established protocol[76]. Briefly, hematopoietic progenitor cells were generated from iPSCs using STEMdiff Hematopoietic kit (STEMCELL Technologies) and cultured in microglia growth medium supplemented with 100 ng/mL interleukin-34, 50 ng/mL tumor growth factor-beta and 25 ng/mL macrophage colony-stimulating factor (Peprotech) for 25 days, following which 100 ng/mL cluster of differentiation 200 (Abcam) and C-X3-C motif chemokine ligand 1 (Peprotech) were also added to the culture. iMGL were considered mature after 28 days of differentiation. Cells were maintained at 37 °C under a 5% $CO_2$ atmosphere throughout the protocol. For qRT-PCR control of differentiation status, TRIzol (ThermoFisher Scientific) was used to extract RNA, followed by cleaning using an RNeasy mini kit (Qiagen). Reverse transcription was performed using Moloney murine leukemia virus reverse transcriptase (ThermoFisher Scientific). Real-time PCR was performed using TaqMan assays (ThermoFisher Scientific) on a QuantStudio™ 5 real-time PCR system (ThermoFisher Scientific). The $2^{-\Delta\Delta Ct}$ method was used to analyze the data using glyceraldehyde 3-phosphate dehydrogenase (GAPDH) and tyrosine 3-monooxygenase/ tryptophan 5-monooxygenase activation protein zeta (YWHAZ) as controls.

For quality control of differentiation by FACS, cells were blocked with Human TrueStain FcX and TrueStain Monocyte Blocker (Biolegend) and stained with the following antibodies: anti-CSF1R (clone #61708, R&D Systems), anti-CX3CR1 (clone #2A9-1, Biolegend), anti-MERTK (clone #125518, R&D Systems). Appropriate forward and side scatter profiles were used to exclude debris and doublets from the analysis. Dead cells were excluded based on LIVE/DEAD™ Fixable Aqua (ThermoFisher Scientific) staining. The gating strategy is shown in Supplementary Fig. 22D. Readings were done on an Attune™ Nxt Flow Cytometer and analyzed/visualized using FlowJo™ software. For PFFs uptake assay, cells were challenged with 250 nM Alexa Fluor 488-labeled PFFs for three hours and then washed once with 4% trypan blue solution to quench extracellular fluorescence, and twice with PBS. Cells were fixed in 4% paraformaldehyde solution and counterstained with Hoechst 33342 (1 µg/mL). Total green fluorescence intensity

per cell was quantified using a CellInsight CX5 High Content Screening Platform (ThermoFisher Scientific). All conditions were assessed in triplicate.

### Differentiation of iPSCs into dopaminergic neurons

iPSCs were then grown in suspension for a week in uncoated flasks in induction medium: DMEM F12 (Gibco #10565-018) containing 1x N-2 supplement (Gibco #17502048), 1x B-27 supplement (Gibco #17504044), 1x MEM NEAA solution, 1 mg/mL BSA, 200 ng/mL Noggin, 200 ng/mL SHH (C24II), 3 µM CHIR-99021, 10 µM SB431542 and 100 ng/mL FGF-8. After a week, iPSC formed embryoid bodies (EBs) which were plated in induction medium onto dishes coated with Poly-L ornithine and laminin (PO/L). Attached EBs were kept for another week in culture, then dissociated into small aggregates and plated again onto new PO/L coated flasks in induction medium. At this stage, cells differentiated into early dopaminergic neural progenitor cells (DA NPCs) and were kept in induction medium for another week. DA NPCs were then dissociated into a single cell suspension and plated in differentiation medium: Neurobasal™ Medium (Gibco #21103049) containing 1x N-2 supplement and B-27 supplement as well as 20 ng/mL BDNF, 20 ng/mL GDNF, 1 µg/mL laminin, 200 µM ascorbic acid, 0.1 µM Compound E and 0.5 mM Dibutyryl-cAMP. After 2 weeks in culture in final differentiation medium, TH positive dopaminergic neurons were observed.

### In silico structural modeling

The superposition and cartoons were generated using PyMOL v2.4. The AlphaFold2 model[77] coordinates were retrieved from Uniprot (accession numbers Q8NDZ4 and Q9NUM3 for C3orf58 and SLC39A9 respectively). Structural homologs of both proteins were identified using FoldSeek[78]. For C3orf58, the most similar PDB100 structure with a nucleotide in the active site was WNK1 (pdb 5TF9)[79], with a probability score of 0.98. For SLC39A9, the most similar PDB100 structure was the Zrt-/Irt-like ZIP protein from *Bordetella bronchiseptica* with bound $Zn^{2+}$ (pdb 5TSA)[80], with a probability score of 1.00.

### Genome-wide CRISPR screen

$250 \times 10^6$ RPE-1 cells were infected with the lentiviral Toronto Knock-Out Library (v3)[81] at low multiplicity of infection (MOI) of <0.3, targeting >18,000 genes with >71,000 individual sgRNAs (app. 4 sgRNAs per gene), including non-targeting control (LacZ, EGFP, Luciferase) and polybrene (final concentration 8 µg/mL). Cells were then plated in 15 cm dishes at a density of $5 \times 10^6$ cells per plate. The medium was changed 24 h after plating. 48 h after plating, cells were trypsinized, and replated in medium containing 15 µg/mL puromycin. A fifth of a plate of library transduced cells was also seeded in the absence of puromycin to compare its growth to selected cells and extrapolate the actual MOI at the end of the selection. One plate of untransduced cells was also treated with puromycin to determine the time necessary for the completion of selection. After 3 days of selection, all library-treated cells (with and without puromycin) were trypsinized and counted, and the MOI was determined to be 0.18, indicating that a vast majority of transduced cells has received only one lentiviral particle. The library-treated, puromycin-selected cells were plated again in 15 cm dishes at $5 \times 10^6$ cells per dish. After 3 days, cells were plated in new 15 cm dishes ($5 \times 10^6$ cells/dish) and separated into two replicates of $150 \times 10^6$ cells each for future PFFs-A633 treatment. For each replicate, 3 additional plates (3x $5 \times 10^6$ cells) were also seeded to serve as untreated (no PFF) controls, i.e., baseline representation of sgRNAs in the total population. 24 h after seeding, plates from both replicates were treated independently with 15 nM PFFs-A633 in complete medium. After 24 h, cells were harvested by trypsinization, pooled by repetitive centrifugation, counted, and split in $5 \times 15$ mL tubes containing $30 \times 10^6$ cells each (>$150 \times 10^6$ cells per replicate). These were spun down and gently resuspended in 4% PFA/PBS (pH 7.4, 10 mL/ $30 \times 10^6$ cells) by slowly vortexing. After a 15 min incubation on a rotator, cells were spun down and rinsed 3 times in PBS by repetitive resuspension/ centrifugation. The cell suspensions were kept in 5 mL PBS at 4 °C in the

dark before proceeding to triage by FACS. A population of RPE-1 cells transduced with a lentivirus encoding Cas9 and a LacZ-targeting sgRNA was subjected to the same treatments as the lentiviral library-transduced population (transduction, selection, and treatment with or without PFFs-A633), to use as gating controls in FACS sorting. The gating strategy is shown in Supplementary Fig. 22A-C.

The cells with the 15% lowest and 15% highest PFFs-A633 fluorescence were sorted by FACS. Briefly, cell suspensions were adjusted to $30 \times 10^6$ per 2 mL and dispensed in 5 mL FACS tubes with a cell strainer snap cap (Corning, 352235). Replicate 1 was sorted on FACS Aria II and replicate 2 on a BD Influx System. In total, $10 \times 10^6$ to $13 \times 10^6$ cells per sample were obtained after sorting, providing 138-fold library coverage. The sorted cells for each population (replicate 1 – Low PFF/High PFFs; replicate 2 – Low PFF/High PFF) were pooled, centrifuged, and the cell pellet was kept at −80 °C before gDNA extraction.

## Genomic DNA (gDNA) extraction from fixed cells

A custom protocol was used for gDNA extraction from fixed cells, as follows, and scaled proportionally for each sample depending on actual cell counts. In a 15 mL conical tube, 6 mL of NK lysis Buffer (50 mM Tris, 50 mM EDTA, 1% SDS, pH 8.0, sterile water) and 30 µL of 20 mg/mL Proteinase K (Qiagen, 19131) were added to the $30 \times 10^6$ cells, and incubated at 55 °C 30 min, inverted and gently vortexed, then incubated again at 55 °C overnight. The next day, 30 µL of 10 mg/mL RNase A (Qiagen, 19101, diluted in NK lysis buffer to 10 mg/mL and then stored at 4 °C) was added to the lysed sample, which was then inverted 25 times and incubated at 37 °C for 30 min. Samples were cooled on ice before the addition of 2 mL pre-chilled 7.5 M ammonium acetate (Sigma A1542) to precipitate proteins. Stock solutions of 7.5 M ammonium acetate were made in sterile dH$_2$O and kept at 4 °C until use. After adding ammonium acetate, the samples were vortexed at high speed for 20 seconds followed by centrifugation at > 4000 g for 10 min. After the spin, a tight pellet was visible in each tube and the supernatant was carefully decanted into a new 15 mL canonical tube. 6 mL of 100% isopropanol was then added to the tube, inverted 50 times, and centrifuged at >4000 g for 10 min. gDNA was visible as a small white pellet in each tube. The supernatant was discarded, 6 mL of freshly prepared 70% ethanol was added, the tube was inverted 10 times, and then centrifuged at >4000 g for 1 min. The supernatant was discarded by pouring, the tube was briefly spun, and the remaining ethanol was removed using a P200 pipette tip. After air drying for 10-30 min, the DNA changed appearance from a milky white pellet to slightly translucent. At this stage, 500 µL of 1x TE buffer (Sigma, T9285) was added, and the tube was incubated at 65 °C for 1 h and at room temperature overnight to fully resuspend the DNA. The next day, the gDNA samples were vortexed briefly. The gDNA concentration was measured using a Nanodrop (Thermo Scientific), and gDNA concentration was adjusted to 400 ng/µL in 1X Tris-EDTA buffer.

## sgRNA sequences libraries preparation, Illumina sequencing, and analysis with MAGeCK

For the first (outer) PCR, an approximately 107-fold library coverage was obtained by performing 21 reactions (50 µL each) per sample of the following PCR recipe: 2.5 µg gDNA, 10 µM Outer primers (see Table S5), 2X KAPA HiFi HotStart ReadyMixPCR Kit, ultrapure water q.s.p. 50 µL. All reactions for a given sample were pooled and mixed by vortexing, and 50 µL per sample were run on 2% agarose gel to check for the presence of a faint ~600 bp amplicon. The second (inner) barcoding PCR was performed as shown in Table S5 for unique barcoding of each sample. Each PCR product was then run on a 2% agarose gel until good separation between the barcoding PCR amplicon (~200 bp) was well separated from the ~600 bp outer PCR amplicon. The barcoding PCR amplicon was gel extracted for each sample, DNA was quantified using a Nanodrop, the A260/A280 ratio checked for sample quality, and samples were stored at −20 °C before pooling and Illumina sequencing. The number of reads obtained for each sample is indicated in Table S5. Raw sequencing data will be made available publicly.

## Validation of screen hits by high-content microscopy

Lentiviral particles encoding Cas9 (blasticidin selection) and EGFP (puromycin selection) were used sequentially to generate RPE-1 cells stably expressing Cas9 and EGFP. These cells were then used for gene editing by reverse transfection in 96-well plates for imaging. Briefly, each well received 5.6 µL of a 3 µM synthetic sgRNA solution, to which was added 25 µL of a HiPerFect (Qiagen, 301705) transfection mix (0.75 µL HiPerFect + 24.25 µL serum-free DMEM). After 10 min incubation at room temperature, 2000 cells were seeded per well (100 µL of a 20,000 cells/mL suspension in complete DMEM) before incubation at 37 °C. This brought the sgRNA concentration to app. 130 nM final. A full medium change was performed after 24 h of transfection. After 72 h of transfection, cells were treated with 15 nM PFFs-A633 for 24 h before rinsing 3x with PBS, followed by 20 min fixation in 4% paraformaldehyde (PFA) at room temperature. Cells were rinsed again in PBS, nuclei were stained with Hoechst 33342 (Invitrogen, H3570) for 10 min (0.2 µg/ml in PBS), and after 2x rinsing in PBS, cells were imaged using a CellInsight CX7 high-content microscope (ThermoFisher). Image analysis was performed using the HCS Studio Cell Analysis software. Briefly, cells were defined as objects using the thresholded EGFP channel, and the cell area was quantified as the EGFP-positive area (in pixels) for each cell. The integrated PFFs-A633 signal per cell was also quantified using the thresholded PFFs-A633 channel. The ratio of PFFs-A633 signal over cell area was then calculated for each cell. The mean value of PFFs signal/cell area was calculated for each well, normalized as a percentage of the mean of AAVS1 sgRNAs-treated wells, and used to create the final heatmap in GraphPad Prism. sgRNAs sequences used are shown in Table S4.

## Generation of monoclonal CRISPR-edited RPE-1 cells

RPE-1 cells stably expressing Cas9 were transfected using HiPerFect in 24 well plates with synthetic sgRNAs against the target genes (*C3orf58*, *SLC39A9*, *TM9SF2*, *VPS35*). The method used was as described in the "validation of screen hits by high-content microscopy" section, except that all volumes and cell numbers were scaled up 6 times, and a combination of two sgRNAs was used, respecting the final sgRNA concentration. sgRNAs sequences used are shown in bold in Table S4. 72 h post-transfection, cells were trypsinized and resuspended. App. 25% of the suspension was used for gDNA extraction (QuickExtract, Lucigen, QE09050) and validation of gene editing by PCR, assuming that efficient cleavage by both sgRNAs results in loss of a portion of the amplicon (*C3orf58*, *TM9SF2*, *VPS35*), or of one of the two reverse primers binding sites (*SLC39A9*). The rest of the suspension was used to isolate single cells in individual wells of 96 wells plates by FACS, and monoclonal lines were then amplified, screened by PCR, and gene editing validated by allelic sequencing. See Supplementary Fig. 5 for PCR screening and allelic sequencing strategies.

## Molecular cloning and plasmids

The lentiCas9-Blast plasmid (Addgene #52962, kind gift from Feng Zhang) was used to generate RPE-1 cells constitutively expressing Cas9. pLJM1-EGFP (puromycin selectable, Addgene #19319, gift from David Sabatini) was used to drive constitutive expression of EGFP in RPE-1-Cas9 cells. The pLJM1 backbone (derived from pLJM1-EGFP) was used to drive constitutive expression of all other constructs. The coding sequence of wild-type (WT) C3orf58 (codon optimized) and WT SLC39A9, both tagged with a C-terminal HA epitope, were ordered as gBlock gene fragments (Integrated DNA Technologies) and cloned into the NheI/EcoRI restriction sites of pLJM1 using Gibson Assembly (New England Biolabs (NEB)). The HA-tagged C3orf58-3MUT was generated by first introducing the D287N mutation using a mutated gBlock and the same cloning strategy as WT C3orf58. The additional K198A and D306N were then introduced using the QuikChange II site-directed mutagenesis kit (Agilent). The same kit was used to generate the HA-tagged D159A and H185R SLC39A9 mutants. The WT and D620N VPS35-HA constructs were PCR amplified from WT and D620N VPS35-6G-GFP constructs (in pAAV-GFP-CAG vector, kind gifts from Austen Milnerwood) and cloned into the NheI/EcoRI restriction sites of pLJM1 using Gibson Assembly. Doxycycline inducible NDST1-HA was

PCR amplified from a home-made plasmid containing the human NDST1 cDNA fused to a C-terminal HA tag (pLJM1 NDST1-HA), before cloning into the NheI and BamHI sites of the pCW57.1 plasmid (Addgene #41393, gift from David Root) using T4 DNA ligase (NEB). All PCRs were made using Q5 DNA polymerase (NEB). All constructs were sequence verified. The sequence of oligonucleotides and dsDNA gene fragments used for cloning are indicated in Table S5. The plasmids generated here will be deposited shortly after publication in the Addgene repository.

### Antibodies
Antibodies used in this study are listed in Table S6.

### PFFs uptake and binding assays for PFFs and GFP + , and live staining of surface heparan sulfate
For PFFs binding assays, cells grown to 50–75% confluency on 96 well plates (Falcon, #353219) were placed on ice for 3 min, and rinsed once with cold unsupplemented DMEM. Cells were then incubated with 180 nM Alexa fluor-labeled PFFs and 10e4 anti-heparan sulfate antibody (1/500) in unsupplemented DMEM for 20 min on ice. After rinsing 3 times in cold DMEM, cells were fixed in cold 4% PFA/PBS for 15 min, rinsed 3 times in PBS at room temperature (RT), and incubated for 25 min at RT in blocking solution (5% Normal Goat Serum in PBS). Finally, cells were incubated with Alexa fluor-coupled anti-mouse secondary antibody (ThermoFisher, 1/1000) and Hoechst 33342 (0.2 µg/ml) in blocking solution for 20 min and rinsed 3 times in PBS before imaging on a Zeiss fluorescence microscope and analysis with a custom ImageJ pipeline. Cells were plated at least 24 hours before starting the assays. Where indicated, cells were pre-treated at 37 °C with sodium chlorate (Sigma, 403016-100 G) in complete medium for 48 h, or for 1 h in serum-free DMEM with a cocktail of heparinase I (NEB, P0735S), heparinase II (NEB, P0736S) and heparinase III (NEB, P0737S) at 3 U/mL each. GFP+ binding assay was performed as for PFFs binding assay, except cells were treated with 50 nM GFP+ for 20 min without co-treatment with anti-HS antibody.

For PFFs uptake assays in RPE-1 cells, adherent cells with treatments mentioned above were subjected to a medium change with culture medium containing Alexa-dyes coupled PFFs (15 nM in the FACS-based screen, or 60 nM for subsequent microscopy experiments) before overnight incubation at 37 °C. Cells were then rinsed three times with PBS, before further processing. Imaging was performed with a CellInsight CX7 high-content microscope and PFFs uptake quantification using the HCS Studio Cell Analysis software. Briefly, cell nuclei were identified as objects, and the corresponding ROIs were extended. The integrated PFFs signal per cell was then quantified using the thresholded PFFs channel, and the mean integrated PFFs intensity per cell was calculated for each condition, before normalization to the control indicated in each experiment.

For PFFs binding assay in iDNs following HS-altering treatments, iDNs were differentiated for 2 weeks in 96 well plates, and treated with either Heparinase I + III (3U/ml each, 1 hour) or 60 mM sodium chlorate (48 hours) before transfer on ice and incubation with 180 nM PFFs-A488 and 1/500 anti-HS antibody (10e4 epitope) for 20 min. iDNs were then fixed, and processed for TH immunostaining, 10e4 signal detection with a fluorescence secondary antibody, and nuclear staining with Hoechst 33342. Images were acquired using an Opera Phoenix high-content imaging platform before image analysis using Harmony Columbus software. PFFs spot intensity was quantified in TH-positive area and normalized to the PFFs-only condition. HS intensity in TH-positive area was also measured. PFFs uptake in WT iDNs following heparinases or sodium chlorate treatment was quantified in a similar manner, except that iDNs were incubated with PFFs-A488 and anti-HS antibody for 3 hours at 37 °C before fixation.

For PFFs uptake in *C3orf58* -/- and *SLC39A9* -/- iDNs, neurons were differentiated for 2 weeks in 96 well plates, and treated with 60 nM PFFs-A633 for 24 hours before fixation, permeabilization, TH immunostaining and nuclear staining with Hoechst 33342. Images were acquired using a Zeiss fluorescence microscope. PFFs intensity was measured in TH-positive cells using a custom ImageJ macro, normalized to TH-positive area, and corrected for background fluorescence using images from cells not treated with PFFs.

### PFFs pulse-chase assay for early uptake
RPE-1 cells of the indicated genotypes were incubated on ice with 180 nM α-syn PFFs-A633 for 20 min in serum-free DMEM. Cells were then incubated at 37 °C for the indicated times, before a 30 sec wash with either PBS or pre-warmed trypsin to remove surface-bound PFFs and retain only the signal of internalized PFFs. Cells were then fixed, and nuclei stained with Hoechst 33342 before imaging on a Zeiss epifluorescence microscope at 20x. Image quantification was performed using ImageJ. Briefly, for each cell, an ROI was drawn manually using the Hoechst and WGA stainings as guides. The PFFs signal was thresholded (pixel intensities min = 1087, max = 1580) and the data relative to PFFs fluorescence intensity per cell was measured. PFFs positive and negative cells were identified manually on thresholded images.

### Cell surface proteome labeling
The optimal concentration of EZ-Link™ Sulfo-NHS-Biotin reagent (21331, ThermoFisher) to be used for staining a maximum of cell surface proteins without labeling the interior of the cells was determined by microscopy (Supplementary Fig. 12A, B). 8000/well WT RPE-1 cells were plated in 96-well plates for imaging and incubated at 37 °C for 24 hours. All subsequent steps until fixation were performed at 4 °C. Cells were then rinsed twice with ice-cold PBS containing $Ca^{2+}$ and $Mg^{2+}$ (311-011-CL, Wisent), then incubated for 30 min in 70 µL biotinylation solution consisting of EZ-Link™ Sulfo-NHS-Biotin reagent diluted at the indicated concentrations in PBS. After rinsing twice with PBS, the remaining biotinylation reagent was inactivated by the addition of 100 µL 100 mM glycine in PBS for 15 min before fixation in 4% PFA. Then, cells were incubated for 1 h at room temperature in a PBS solution containing 0.2 µg/mL Hoechst 33342 (H3570, ThermoFisher) and 1 µg/mL Streptavidin-Alexa488 (016-540-084, Jackson ImmunoResearch). After 3 rinses with PBS, cells were imaged and analyzed using a CX7 high-content microscope.

For MS, one 10 cm plate per sample of ~90% confluent RPE-1 cells were used. For each line (WT, C3orf58-/-, SLC39A9-/-, TM9SF2-/-, VPS35+/-), biological triplicates were analyzed. All steps were performed in the cold until freezing cell pellets. Cells were washed twice with 7 mL PBS $Ca^{2+}/Mg^{2+}$ before incubation with 7 mL 0.25 mg/mL EZ-Link™ Sulfo-NHS-Biotin in PBS for 30 min, then washed twice with 7 mL PBS. The remaining biotinylation reagent was inactivated by the addition of 7 mL 100 mM glycine in PBS for 15 min before 2 additional washes in 7 mL and 4 mL PBS. Cells were then scraped in 1 mL PBS, and pelleted by centrifugation at 5000 g for 5 min at 4 °C. The supernatant was removed, and pellets snap frozen for 3 min in a slurry of dry ice/70% ethanol. Pellets were frozen at −80 °C until processed.

### Mass-spectrometry analysis of cell surface proteome
Lysis buffer containing 10 mM Tris-HCl, 150 mM NaCl, 1% Triton, 0.1% SDS, and 1 mM protease inhibitor cocktail was added to the frozen pellet for 30 min at 4 °C. Lysates were sonicated for 5 seconds and centrifuged at 4 °C for 30 min at 4000 rpm. Cleared lysates were incubated with 70 µl of streptavidin beads (5 ml; 17-5113-01, GE Healthcare) at 4 °C for 3 h, washed with lysis buffer followed by six washes using 50 mM ammonium bicarbonate (AB0032, 500 G, Biobasic). For protein digestion, 0.2 µg of Trypsin/Lys C (V5071, Promega) was added into each 100 µl sample in 50 mM ammonium bicarbonate overnight at 37 °C. Supernatants were transferred into new microtubes, beads were washed two times with 100 µl of water, and supernatants from all washes were pooled with the initial supernatant. Trifluoroacetic acid (1%; 302031 Sigma) was added to each tube to acidify samples. Samples were then desalted using desalting spin columns (89852; Pierce, ThermoFisher) per the manufacturer's protocol and subsequently dried in a SpeedVac. Peptides were resuspended in 26 µl of 1% formic acid and kept at −80 °C. Peptides were loaded onto a 75-µm internal diameter × 150 mm Self-Pack C18 column installed in an Easy-nLC 1000 system (Proxeon Biosystems). LC-MS/MS was conducted using a 120 min

reversed-phase buffer gradient running at 250 nl/min (column heated to 40 °C) on a Proxeon EASY-nLC pump in-line with a hybrid LTQ-Orbitrap velos mass spectrometer (ThermoFisher Scientific). A parent ion scan was performed in the Orbitrap, using a resolving power of 60000. Simultaneously, up to the twenty most intense peaks were selected for MS/MS (minimum ion count of 1000 for activation) using standard CID fragmentation. Fragment ions were detected in the LTQ. Dynamic exclusion was activated such that MS/MS of the same m/z (within a 10-ppm window, exclusion list size 500) detected three times within 45 s were excluded from analysis for 30 s.

For protein identification, Raw MS files were analyzed using the Mascot search engine through the iProphet pipeline integrated into Prohits and the Human RefSeq database v.57. Search parameters specified a parent MS tolerance of 15 ppm and an MS/MS fragment ion tolerance of 0.4 Da, with up to two missed cleavages allowed for trypsin. Oxidation of methionine was allowed as a variable modification. Data were analyzed using the trans-proteomic pipeline[82] via the ProHits software suite[83]. Proteins identified with a ProteinProphet cut-off of 0.85 (corresponding to ≤1% FDR) were analyzed.

## RNAseq analysis

RNAseq analysis was performed using the GenPipes RNAseq analysis pipeline version 3.0 run on Compute Canada's high-performance computing cluster. The raw sequencing (FASTQ) files were aligned to the human genome version GRCh38 using STAR, Picard, and Cufflinks were run for corrections and calculating raw and normalized RNA read counts. Differential gene expression was calculated using EdgeR[84]. Biological triplicates for control cultures were separately compared with triplicate samples of $C3orf58^{-/-}$ and $SLC39A9^{-/-}$ lines. P values were adjusted using a false rate discovery (FDR) correction for multiple comparisons and threshold 0.1 was applied to select differentially expressed genes. The selected genes were the inputs into gene ontology analysis using GOseq[85].

## Microscopy

Except for high-content microscopy analyses, microscopic analysis was performed as described previously[86].

## Immunocytochemistry

Unless otherwise stated, cells previously plated in 96 wells plates for imaging were rinsed in PBS, fixed for 15 min in 4% PFA/PBS, rinsed 3 times in PBS, permeabilized for 5 min with 0.15% Triton X-100/PBS (or 0.05% NP-40 where indicated), rinsed again 3 times in PBS, and incubated for 30 min in blocking solution (5% normal goat serum in PBS). After blocking, cells were incubated with the antibodies of interest for 1 h in blocking solution, rinsed 3 times in blocking solution, and incubated for 1 h with the adequate fluorescently-labeled secondary antibodies diluted in blocking buffer. Cells were rinsed 3 times in PBS, incubated for 10 min with Hoechst 33342 (0.2 µg/mL) in PBS, and rinsed again 2 times before microscopy.

## Western blot

Unless otherwise stated, samples were prepared for SDS-PAGE in 1X Laemmli buffer supplemented with 0.1 M DTT, or RIPA buffer containing 0.2% SDS supplemented with protease and phosphatase inhibitors to which 5X Laemmli buffer supplemented with 0.5 M DTT was added to reach a 1X final concentration. Samples were then boiled for 10 min at 95 °C. After separation by SDS-PAGE, proteins were transferred to Nitrocellulose membranes using the semi-dry Trans-Blot Turbo system (Bio-Rad). Membranes were then blocked in 5% milk/PBS-T (PBS containing 0.1% Tween 20), and primary antibodies were incubated in 1% milk/PBS-T by shaking at 4 °C overnight. After 3 rinses in PBS-T, membranes were incubated with appropriate secondary antibodies coupled to horseradish peroxidase (Jackson ImmunoResearch) for 1 hour at room temperature, and rinsed again 3 times in PBS-T. HRP signal was revealed using ECL substrate (Pierce, or Bio-Rad's Clarity Max), and chemiluminescent signal was captured using a ChemiDoc imaging system (Bio-Rad).

## Sodium carbonate extraction

RPE-1 cells grown to 80-90% confluency in a 35 mm dish were rinsed twice with ice-cold PBS, scraped in 1 mL PBS and spun down at 5,000 g for 5 min at 4 °C. The cell pellet was resuspended in 100 µL PBS, and proteins were quantified by BCA assay (Pierce). An amount of cell suspension representing 80 µg proteins was spun down again and resuspended in 80 µL (1 µg/µL protein) freshly prepared, cold 0.1 M $Na_2CO_3$ (pH 11.5). After 20 min incubation on ice, the suspension was ultracentrifuged at 100,000 rpm for 20 min at 4 °C. The supernatant containing soluble and membrane-associated proteins was supplemented with Laemmli buffer (1X final, 100 mM DTT). The pellet containing integral membrane proteins was resuspended in an equal volume of 1X Laemmli buffer containing 100 mM DTT. Samples were then boiled for 10 min at 95 °C and processed for Western blot.

## Measurement of $Zn^{2+}$ content with Zinpyr-1

2500 RPE-1 cells per well were seeded in 96-well plates for imaging. The day after, cells were incubated at 37 °C for 30 min with complete DMEM containing 5 µM Zinpyr-1 (Abcam, ab145349), Hoechst 33342 (0.2 µg/mL), and 1 mM EDTA. Cells were then washed twice in pre-warmed Live Cell Imaging Solution (ThermoFisher, A14291DJ) containing 1 mM EDTA, live cells were imaged with a CX7 high-content microscope connected to a live module (37 °C, 5% $CO_2$, 60% humidity), and the mean average intensity per cell was quantified with the HCS Studio Cell Analysis software.

## Purification of recombinant α-syn

The full-length human wild-type α-syn coding sequence was cloned into pGEX-6P-1 plasmid and then transformed into *E.coli* BL-21(DE3) strain. Bacteria were then grown in 500 mL of LB broth supplemented with ampicillin (100 µg/mL), expression of GST-tagged α-syn was induced by the addition of 0.3 mM IPTG. Bacteria were harvested by centrifugation at 5000 g, 4 °C for 15 min, resuspended in 30 mL of 25 mM Tris-HCl (pH 8.0)-buffered solution containing 400 mM NaCl, 5% (v/v) glycerol, 0.5% (v/v) Triton-X100, 1 mM DTT and protease inhibitor cocktails (0.5 µg/mL aprotinin, 0.5 µg/mL leupeptin, 130 µg/mL phenylmethylsulfonyl fluoride, 500 µg/mL benzamidine), and sonicated on ice-bath. The cell lysate was clarified by centrifugation at 20,000 g, 4 °C for 30 min and added to a 50-mL Falcon tube holding 6 mL of Glutathione-Sepharose® 4B resin (GE Healthcare) that was pre-equilibrated in the same buffer. This mixture was incubated on a nutator at 4 °C overnight before packing onto a 10 mL disposable polypropylene column. GST-α-syn bound to the resin was then eluted from the column with 20 mM glutathione freshly prepared in 25 mM Tris, 400 mM NaCl solution (pH 8.0). After combining all GST-α-syn eluates, protein concentration was determined using a Bradford assay kit with BSA as a standard (Pierce). Based on the quantification result, GST-HRV 3 C protease was added to GST-α-syn at a ratio of 1:50 (w/w) and the cleavage reaction was incubated at 4 °C overnight to remove the GST tag from GST-α-syn. The reaction mixture was then concentrated to about 4 mL using Amicon Ultra-15 Centrifugal Filter Unit 3000 NMWL (Millipore) with multiple buffer exchanges to PBS (pH 7.4). The sample was subsequently loaded onto a PBS-equilibrated Superdex 200 16/600 column (GE Healthcare) on the ÄKTA Pure system. Fractions containing α-syn were collected and further purified by passing through a GSTrap 4B column (GE Healthcare). The purified α-syn was determined to be homogeneous by SDS-PAGE followed by Coomassie blue staining and the concentration was adjusted to 5 mg/mL in PBS (pH 7.4). Finally, the purified α-syn was sterile-filtered through a 0.22 µm PVDF syringe filter (Millipore) before being aliquoted and stored at −80 °C.

## Preparation of α-syn Pre-formed Fibrils (PFFs) and oligomers

PFFs preparation and quality control were described previously[87,88]. PFFs were prepared by shaking 0.5 mL aliquots of purified α-syn (5 mg/mL, so 346 µM equivalent monomers of 14.46 kDa α-syn) held in a 1.5 mL

microtube on a ThermoMixer (Eppendorf) set at 1000 rpm and 37 °C for 5 days. The obtained fibrils were sonicated to produce smaller species of fibrillated α-syn (i.e., PFFs) on a Bioruptor® Plus sonication unit (Diogenode) with cooling water circulation at 10 °C for 30 sec x 10 cycles. Samples (~20 μL) were harvested before and after sonication for quality control by electron microscopy imaging and dynamic light scattering analysis. For electron microscopy, PFFs were characterized using a negative staining protocol. PFFs were added to 200 mesh cupper carbon grid (SPI Supplies, 3520C-FA), fixed for 1 min with 4% paraformaldehyde, and stained with 2% uranyl acetate (EMS, 22400-2) for 1 min. PFFs were visualized using a transmission electron microscope (FEI Tecnai 12 Bio Twin 120 kV TEM) coupled to an AMT XR80C CCD Camera (Supplementary Fig. 1A). The length of individual fibrils was measured with ImageJ and the size distribution was analyzed with MATLAB. PFFs were of 54.24 nm average length after sonication (Supplementary Fig. 1B) The sonicated PFFs were either aliquoted (20 μL) for storage at −80 °C or further labeled with fluorescent dyes as indicated below.

α-syn oligomers were generated by incubating monomeric α-synuclein (1 mg/mL) with 4-hydroxynonenal (HNE; Cayman Chemical) at a 1:30 molar ratio (protein:HNE) in PBS (pH 7.4) for 18–24 hours at 37 °C, as previously described (Roberts et al., Brain, 2015; Almandoz-Gil et al., Mol Biol Rep, 2018). Unreacted aldehyde was removed, and buffer exchange was performed using a 7 kDa MWCO Zeba™ Spin Desalting Column (Thermo Fisher Scientific), followed by concentration with a 3 kDa MWCO Amicon® Ultra centrifugal filter unit (Millipore). Samples were stored at –80 °C until use. Oligomer formation was confirmed by SDS-PAGE, Thioflavin T assay and sedimentation analysis.

## Thioflavin T (ThT) fluorescence and sedimentation assays

For ThT fluorescence assays, ThT was dissolved in molecular-grade water (Wisent) at a concentration of 1 mM and filtered through a 0.22 μm syringe filter. A working solution of 50 μM ThT was freshly prepared in PBS (Wisent) immediately before use to achieve a final concentration of 20 μM in the assay. For each condition, 50 μL of α-synuclein samples —monomer, pre-formed fibrils, or oligomer— at final concentrations of 1 μM, 2.5 μM, or 5 μM were mixed with 50 μL of the 50 μM ThT solution in black, flat-bottom 96-well plates (Corning), resulting in a final ThT concentration of 20 μM. Plates were incubated for 1 hour at room temperature in the dark. Fluorescence was recorded using a FLUOstar Omega plate reader (BMG Labtech) with excitation at 440 nm and emission at 480 nm. Fluorescence intensity was expressed in arbitrary units (a.u.). PBS and molecular-grade water (H₂O) were included as negative controls. For the time-course fibrillization assay, aliquots were collected daily from the aggregation reaction, and stored at –80 °C. All time points were later analyzed in parallel under identical ThT conditions to minimize variability. Prior to ThT incubation, each aliquot was diluted to a final α-synuclein concentration of 1 μM and processed as described above.

To assess the solubility and aggregation state of the different α-synuclein species (monomers, PFFs, oligomers), samples (5 μL at 5 mg/mL) were diluted 1:10 in dPBS and ultracentrifuged at $100,000 \times g$ for 30 min at 4 °C (TLA-120.1 rotor). Supernatants and resuspended pellets were mixed with Laemmli buffer, boiled, and resolved by SDS-PAGE. Gels were stained with Coomassie Blue (0.1% Coomassie, 20% methanol, 10% acetic acid), destained in 50% methanol/10% acetic acid, washed, and imaged using a Bio-Rad ChemiDoc system.

## Preparation of fluorescent dye-labeled α-syn PFFs

Sonicated PFFs (150 μL) were gently mixed with 5 μL of Alexa Fluor® 633 or 488 succinimidyl ester dyes (5 mg/mL in DMSO) and incubated in the dark for 20 min at room temperature. The labeled product was then transferred to a Slide-A-Lyzer device (Thermo Scientific) and dialyzed against PBS (pH 7.4) at 4 °C with three changes of dialysis buffer to remove the free residual fluorescent dye. 20 μL aliquots were made before storage at −80 °C. The concentration of the labeled PFFs was extrapolated from the initial concentration (5 mg/ml) and final volume post-dialysis.

## Preparation of Cy3-labeled tau oligomers

Tau oligomers were a gift from Merck, and were characterized previously[89]. Briefly, recombinant tau monomers (human 4R2N T40/Evotec, Germany) were labeled with Cy3 (Life Technologies). To prepare tau oligomers, 5 μM monomeric tau in 100 mM MES buffer (pH = 6.5; 4-Morpholineethanesulfonic acid hydrate) was mixed with 10 μM DTT (BioShop) and incubated for 10 min at 55 °C. Subsequently, 5 μM heparin (Fisher, H19) was added to the solution to induce aggregation and incubated with shaking (1000 rpm) for 4 h at 37 °C. Tau oligomers were stored at −80 °C.

## Analysis of cell-surface Heparan Sulfate by LC-MS/MS

a) *cell surface HS harvesting*: 90% confluent RPE-1 cells in 15 cm dishes were rinsed twice in 10 mL PBS before detachment using 2 mL trypsin/EDTA for 10 min at 37 °C. 3 mL serum-free DMEM was added and the cell suspension was triturated before harvesting in a 15 mL conical tube, ensuring minimal left-over cell suspension in the plate. The suspension was centrifuged for 3 min at 2,000 rpm. The supernatant containing HS (detached from the cell surface after trypsin digestion of HSPG) was carefully removed from the pellet, frozen at −80 °C and shipped on dry ice before analysis. The cell pellet was also frozen at −80 °C before protein quantification using DC protein assay (Bio-Rad).

b) *sample preparation for LC-MS/MS:* A DEAE column was applied to purify the HS from post-trypsinization cell supernatants. Before loading to DEAE column, 2 μL ¹³C-labeled recovery calibrant NSK5P (45 ng/μL) was added to the cell supernatant. DEAE column mobile phase A contained 20 mM Tris, pH 7.5, and 50 mM NaCl, and mobile phase B contained 20 mM Tris, pH 7.5, and 1 M NaCl. After loading the sample into DEAE column, the column was washed with 1.5 mL mobile phase A, followed by 1.5 mL mobile phase B to elute the HS. The YM-3KDa spin column was applied to desalt the elute, and the retentate was subjected to heparin lyases digestion. Before the digestion, ¹³C-labeled 3-O-sulfated calibrants (each 500 ng) were added to the retentate. 100 μL of enzymatic buffer (100 mM sodium acetate/2 mM calcium acetate buffer (pH 7.0) containing 0.1 g/L BSA), and 20 μL of enzyme cocktails containing 5 mg/mL each of heparin lyase I and II was added to degrade the retentate on the filter unit of the YM-3KDa column. The reaction solution was incubated at 37 °C for 6 h. Before recovering the disaccharides from the digest solution, a known amount ¹³C-labeled non-3-O-sulfated disaccharide calibrants (ΔIS = 80 ng, ΔIIS = 80 ng, ΔIIIS = 40 ng, ΔIVS = 80 ng, ΔIA = 40 ng, ΔIIA = 80 ng, ΔIIIA = 40 ng, ΔIVA = 250 ng) were added to the digestion solution. The HS disaccharides and tetrasaccharides were recovered by centrifugation, and the filter unit was washed twice with 200 μL of deionized water. The collected filtrates were freeze-dried before the AMAC derivatization. The AMAC label and LC-MS/MS analysis of the collected disaccharides and tetrasaccharides was performed as described below. The amount of HS was determined by comparing the peak area of native di/tetra-saccharide to each corresponding ¹³C-labeled internal standard, and the recovery yield was calculated based on a comparison of the amount of ¹³C-labeled disaccharide (Δ[¹³C]UA-[¹³C]GlcNS from heparin lyases degraded ¹³C-labeled NSK5P) in the tissue samples and control, respectively.

c) *Chemical derivatization of HS disaccharides:* The 2-Aminoacridone (AMAC) derivatization of lyophilized samples was performed by adding 10 μL of 0.1 M AMAC solution in DMSO/glacial acetic acid (17:3, v/v) and incubating at room temperature for 15 min. Then 10 μL of 1 M aqueous sodium cyanoborohydride (freshly prepared) was added to this solution. The reaction mixture was incubated at 45 °C for 2 h. After incubation, the reaction solution was centrifuged to obtain the supernatant that was subjected to the LC-MS/MS analysis.

d) *LC-MS/MS analysis:* The analysis of AMAC-labeled di/tetra-saccharides was implemented on a Vanquish Flex UHPLC System (Thermo-Fisher Scientific) coupled with TSQ Fortis triple-quadrupole mass

spectrometry as the detector. The ACQUITY Glycan BEH Amide column (1.7 μm, 2.1 × 150 mm; Waters, Ireland, UK) was used to separate di/tetra-saccharides at 60 °C. Mobile phase A was 50 mM ammonium formate in water, pH 4.4. Mobile phase B is acetonitrile. The elution gradient as follows: 0-20 min 83–60% B, 20–25 min 5% B, 25–30 min 83% B. The flow rate was 0.3 mL/min. On-line triple-quadrupole mass spectrometry operating in the multiple reaction monitoring (MRM) mode was used as the detector. The ESI-MS analysis was operated in the negative-ion mode using the following parameters: Neg ion spray voltage at 3.0 kV, sheath gas at 55 Arb, aux gas at 25 arb, ion transfer tube temp at 250 °C and vaporizer temp at 400 °C. TraceFinder software was applied for data processing.

## Golgi structure analysis

Immunocytochemistry was performed on RPE-1 cells of the indicated genotypes, using GM130 as a cisGolgi marker and Golgin97 as a transGolgi marker. Z-stacks (60 stacks per field, 0.185 μm steps) were acquired using a Nikon A1 confocal microscope mounted with a Plan Apo λ 60 × 1.40 objective. Then, a customized CellProfiler pipeline largely inspired by Mejia et al.[57] was used to analyze Golgi morphology. Briefly, changes were made to the Mejia et al. pipeline so that cell outlines corresponded to nuclei extended by 17 pixels and those that touched the image border were filtered out from analyses. Diameters of objects and thresholding factors were adjusted to ensure correct feature detection. Fluorescence intensity measurement modules for GM130 and Golgin97 channels were added.

## NDST1-HA induction, Golgi localization and PFFs uptake

The dox-inducible NDST1-HA expressing WT, *C3orf58* -/- and *SLC39A9* -/- RPE1 cells were counted, and 1500 cells/well were seeded in 96-well optical plates. After 24 hr, the cells were treated with 25 ng/ml of doxycycline. 96 h post-induction, the cells were incubated with PFFs-A488 (60 nM) for 24 h to track uptake. Cells were fixed with 4% PFA and stained with Rb anti-HA for NDST1 and Ms anti-GM130 for Golgi followed by secondary antibodies staining with anti-Rb-A550 and anti-Ms-A647, and Hoechst 33342 for the nucleus. The cells were imaged at 10X using an EVOS M5000 fluorescence microscope (Invitrogen) and at 40X using a Nikon A1 laser scanning confocal microscope. Image analysis and quantification were performed using a custom Cell profiler pipeline. Briefly, cells were identified by their nuclei. Each nucleus object was expanded by 15px and objects overlapping the image edges were removed from analysis. For each cell, objects corresponding to NDST1-HA signal, Golgi apparatus, and PFFs-A488 were identified based on an adaptive thresholding method. Cells were then classified as HA positive or negative depending on the presence or absence of an NDST1-HA object. Total NDST1-HA object intensity per cell was measured, as well as NDST1-HA signal intensity comprised within Golgi objects. Total PFFs intensity per cell was also measured.

## Statistics and reproducibility

Measurements were taken from distinct samples. Statistical analyses were performed using GraphPad Prism unless otherwise stated. Biological replicates were defined as independent experiments (independent cells seeding, independent treatments). For each experiment, the number of replicates analyzed and statistical tests used are specified in the figures' legends.

## Reporting Summary

Further information on research design is available in the Nature Portfolio Reporting Summary linked to this article.

## Data availability

Tables S1-S9 are provided in separate tabs in an Excel format file (Supplementary Data 1). All source data underlying the graphs and charts presented in the main and Supplementary Figs. are available in Supplementary Data 2. The mass spectrometry proteomics data have been deposited to the ProteomeXchange Consortium via the PRIDE[90] partner repository with the dataset identifier PXD065224. RNAseq data have been deposited to the Gene Expression Omnibus (accession number: GSE299483). All other data types are available on request (including but not limited to microscopy images, Sanger sequencing ab1 files, FACS data, and scripts).

## Code availability

This manuscript does not report any original code. If additional information is needed to reanalyze the data, it can be provided by the corresponding authors upon request. Softwares used for microscopy images collection include ZEN 3.0 blue edition (Zeiss epifluorescence microscope), HCS Studio Cell Analysis v6.6.1 (CX7 high-content microscope), Harmony v.5.2 (Opera Phoenix microscope), and NIS-Elements AR v5.42.03 (Nikon A1 confocal microscope). For acquisition of ThT fluorescence data with the FLUOstar Omega, the Reader Control software was used. For FACS data acquisition, BD FACSDiva v8.0 was used for BD Biosciences FACS Aria II cell sorter, BD FACS™ Sortware v1.2.0.142 for BD Biosciences Influx Cell sorter, and Attune Cytometric Software v6 for Attune Nxt Flow Cytometer. For analysis of NGS files from the CRISPR screen, Mageck v0.5.6 was used. For high-content microscopy images analysis, the HCS Studio Cell Analysis (CX7 microscope) or Harmony v5.2 (Opera Phoenix) softwares were used. For analysis of epifluorescence microscopy images, ImageJ v1.53t was used. For analysis of Nikon A1-acquired images, CellProfiler v4.2.5 was used. FLowJo v10.4 was used for FACS analysis. For protein identification by MS, Raw MS files were analyzed using the Mascot search engine through the iProphet pipeline integrated into Prohits. For downstream proteomics data analysis, ProHits was used. For data processing of LC-MS/MS analysis of HS, TraceFinder was used. RNAseq analysis was performed using the GenPipes RNAseq analysis pipeline version 3.0 run on Compute Canada's high-performance computing cluster. For RT-qPCR analysis, the Quant-Studio Design & Analysis Software was used. To generate graphs and perform statistical tests not included in pipelines described above, GraphPad Prism 9.0 was used.

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

## Acknowledgements

This paper is dedicated to the memory of Prof. Robert J. Linhardt who has made foundational contributions in proteoglycan chemistry and biology. We thank all members (present and past) from the Fon lab, the Vanderperre lab, the Early Drug Discovery Unit led by Thomas M. Durcan, and members of the Angers lab for helpful discussions and support throughout the study. We also thank the FACS facility of the Faculty of Medicine from the University of Toronto for their help in processing CRISPR screen samples. Special thanks to Peter S. McPherson and his lab members for sharing thoughts, material and reagents, and to Grégoire Bonnamour at the CERMO-FC microscopy platform for technical help. We also thank Qi Zhang and Yihong Ye for gifting GFP+ protein, and Merck for gifting Tau-Cy3 oligomers. B.V was supported by postdoctoral fellowships from the Canadian Institutes of Health Research and from Canada First Research Excellence Fund, Healthy Brain, Healthy Lives, McGill University. D.L. was supported in part by a doctoral scholarship from PROTEO-UQAM. This work was supported by grants from the Michael J. Fox Foundation for Parkinson's Research (#004884 awarded to E.A.F., #021129 awarded to B.V., E.A.F. and T.M.D.). E.A.F. is supported by a Canada Research Chair (Tier 1) in Parkinson's disease. PROTEO (Regroupement Québécois de Recherche sur la Fonction, l'Ingénierie et les Applications des Protéines) is funded by the Fonds de recherche du Québec (Award Number: 341121).

## Author contributions

B.V. planned and conducted the experiments and data analyses unless otherwise stated, participated to study design with E.A.F., wrote the manuscript, managed the project with E.A.F. and acquired funding. A.M.

planned and conducted experiments relative to PFF binding and uptake in iDNs, NDST1 expression and localization, Golgi structure analysis, prepared samples for LC-MS/MS analysis of HS, and wrote the manuscript with B.V. and E.A.F. M.-F.D. planned and performed experiments on iMGLs. F.L. generated gene-edited iPSCs and and participated to PFF uptake assays in iDNs. E.D.C.P. performed electron microscopy characterization of PFFs and performed PFF uptake assays in iDNs. N.R. troubleshooted the screen and performed the final screen with B.V. C.X.-Q.C. performed characterization of gene-edited iPSCs. R.L. performed the early PFF uptake assay in RPE-1 cells. C.M-T. performed cargo specificity assays with help from B.V. T.G. performed PFF characterization by ThT and sedimentation assays. R.C. performed proteomics samples processing and analyses. D.L. generated NDST1-HA inducible lines and troubleshooted GFP+ binding experiments. G.M. performed screen analysis with Mageck and provided guidance with screen interpretation. R.T. performed data analysis for the RNAseq experiment. Z.W. prepared and processed samples for LC-MS/MS analysis of HS, under the guidance of J.L., who also edited the manuscript. W.E.R. provided support for high-content imaging and associated image analysis. W.L. and I.S. performed PFF production and quality control by dynamic light scattering and SDS-PAGE. Z.F. and K.X. performed initial LC-MS/MS analyses on HS in RPE-1 cells, under the guidance of R.J.L. Z.S. analyzed representation of classes of sgRNAs in FACS-sorted samples for the screen. J.-F.T. performed structural modeling of C3orf58 and SLC39A9. T.M.D. performed supervision, acquired funding, provided critical guidance and resources for iPSC-related experiments, and edited the manuscript. S.A. critically contributed to study design and provided key resources and reagents for performing the screen and interpreting associated data, and edited the manuscript. E.A.F. designed the study, provided guidance for project direction and experimental design, acquired funding, and wrote the manuscript with B.V. and A.M.

## Competing interests

J.L. is a founder and chief scientific officer of Glycan Therapeutics Corp. Z.W. is currently an employee of Glycan Therapeutics Corp. Other authors have no competing interest to declare.
