## [Transparent Peer Review file · Communications Biology]

Novel regulators of heparan sulfate proteoglycans modulate cellular uptake of α -synuclein fibrils

Corresponding Author: Professor Benoît Vanderperre

Version 0:

Reviewer comments:

Reviewer #2

(Remarks to the Author)

Summary: Vanderperre et al. used a CRISPR/Cas9 KO screen and identified a number of genes that increase or decrease the accumulation of a-syn PFFs in RPE1 cells. They focus their study on a sub-selection of genes that regulate HSPGs, and two unexpected hits SLC39A9 (Golgi Zn²⁺ exporter) and C3orf58 (Golgi localized kinase). Enzymatic cleavage of cell surface HS and chlorate-treatment (intended to reduce HS sulfation) reduced cell associated a-syn PFFs. Further, in the abstract the authors claim that mass spec. analysis of SLC39A9 ^{-/-} and C3orf58 ^{-/-} cells indicated defective HS maturation, and was attributed to the impaired cell surface binding of a-syn PFFs. In C3orf58 KO RPE1 cells, an expressed HA-tagged NDST1 accumulated in the Golgi, which is suggested as a potential explanation for the defective HS maturation. Finally, C3orf58 KO in iPSC-derived microglia is found to impede a-syn PFF uptake. The data provided supports a role for cell surface HS as a point of cell surface contact for a-syn PFFs, and indicates that SLC39A9 and C3orf58 are required for normal Golgi function, which if perturbed impacts HS (and HSPG) homeostasis. The study adds additional support for a general role of cell surface HS in interactions with amyloid proteins. There are a number of interesting findings in the study relevant to those working in the HS, amyloid and PD fields. The authors have also taken what seems to be a careful and comprehensive approach to validate the genes identified in their screen, and address cell size as an assay confounder, which was interesting to read.

However, the study is lacking a comprehensive characterization of the a-syn PFFs, and given their importance in the screen for identifying the genes of interest the authors need to provide a more detailed description of what the a-syn PFF preparation actually contains. Relatedly, the fact that heparin is used in the protocol for forming tau oligomers is highly problematic (detailed below). The relevance of the RPE1 cell line for studying a-syn PFF uptake is also not clearly justified. HSPG expression patterns and HS composition is typically cell-type specific, and to what extent the observations would prove relevant for e.g. neurons in which a-syn inclusions (derived endogenously or transferred) are typically observed, is not clear. This potential for such cell-type specific effects is observed in Fig. 9, where the iPSC derived microglial reveal impaired a-syn PFF uptake in C3orf58^{-/-}, but not SLC39A9^{-/-} cells. The study is already of considerable size with the vast majority of the mechanistic findings derived from RPE1 cells, but the authors should provide evidence that the C3orf58^{-/-} impairment of a-syn PFFs in the microglia is in fact due to altered cell surface HS.

A major finding in the study is that C3orf58^{-/-} and SLC39A9^{-/-} cells accumulate/bind less a-syn PFFs than controls. Digestion of cell surface HS (with heparinase), or reducing the degree of HS sulfation (chlorate treatment), also reduced a-syn PFF binding. However, mass spec revealed that C3orf58^{-/-} cells had higher total levels of HS (cell surface, based on the digestion protocol, correct?), more HSPG transcripts, and more HS biosynthetic enzymes. While at the same time there was no apparent difference in 10E4 binding, indicating similar availability of cell surface HS (albeit for this 10E4-positive subpopulation of HS). SLC39A9^{-/-} cells also had significantly less a-syn PFF signal intensity, and significantly less 10E4-positivity, but mass spec reveals no significant differences in total HS (relative to WT). The authors discuss how the di/tetrasaccharide HS compositional analysis may not provide sufficient resolution to determine differences in HS chain domains that might be important for a-syn PFF binding. It is particularly difficult to reconcile how suppressing these two genes, proposed as potential "general regulators of glycans/glycosaminoglycans biosynthesis and/or HS maturation in the Golgi", differently effect total HS levels, while both similarly impact a-syn PFF binding. The discussion around these differences needs greater clarity. Additional experimental proof that the cell surface HS in C3orf58^{-/-} and SLC39A9^{-/-} cells has altered binding potential could perhaps be explored using other established HS-binding proteins (e.g. fluorescently labelled FGF), and performing a similar approach as taken with the 10E4 labelling of live cells. As mentioned above, the

abstract claims that the mass spec data indicates defective HS maturation in C3orf58 and SLC39A9. However, if I understood correctly the conclusion from the composition analysis is that the distribution of the different di- and tetrasaccharide motifs are similar to control, while total HS is significantly increased in C3orf58; therefore, the claim of 'defective HS maturation' is not clearly supported by this data.

Currently, the consequence of these differences in HS-mediated accumulation of a-syn PFFs is not examined. Below I suggest that it might be important to consider/account for the potential effects of gene KOs on cell viability, and possibly even proliferation effects that may not be accompanied by cell size changes. Equally, the effect of a-syn PFF exposure, and relative differences of a-syn PFF binding/accumulation on cell viability would seem relevant to contextualize the findings, and establish whether impairing HS-mediated a-syn PFF interactions would be predicted to have protective effects.

Below are more detailed comments relating to specific experiments and figures:

1. CRISPR/Cas9 screen and assessment of a-syn PFF binding/accumulation. Several of the identified genes were connected to cell growth/cell cycle. The authors discuss how altering cell cycle such that proliferation rate will be increased will result in smaller cells, while an increase will result in larger cells, impacting the cellular capacity to accumulate a-syn PFFs. Further, they discuss how the CRISPR/Cas9 system itself introduces DSBs that are known to impact cell cycle rates, and how the detection of non-targeting sgRNAs (such as LacZ) in the low a-syn PFF group (Fig. 1) is likely due to the fact that they are proliferating faster (as they are not incorporating DSB-associated DNA damage), resulting in smaller cells with reduced capacity to accumulate a-syn PFFs. They go on to perform experiments in SFig. 2 that convincingly demonstrate how cells targeted with LacZ sgRNA/Cas9 are smaller than cells transduced with the genome wide CRISPR library, which leads to lower absolute amounts of a-syn PFFs, but relatively similar amounts, when a cell size correction is implemented. The authors subsequently added sgRNA targeting the safe-harbour control AAVS1 to account for side-effects of the CRISPR/Cas9 approach, and as seen in Fig. 2 the majority of KOs that significantly impact area are cell growth/cycle regulators.

However, in subsequent figures where average a-syn PFF intensity is expressed relative to the AAVS1 (e.g. Fig. 3 and SFig. 4) or WT controls (e.g. Fig. 5 and 6) it is potentially important to determine whether the specific KO of genes, particularly C3orf58 and SLC39A9, impacts cell viability (e.g. via FACs based apoptosis/necrosis assay using Annexin V/PI labeling or similar), and/or proliferation rates (irrespective of cell size/area). As outlined, increased proliferation rate may reduce cell size and capacity to accumulate a-syn PFFs; however, it will also result in a relative reduction in the amount of a-syn PFFs available per cell. Equally, reduced viability may decrease cell numbers and the total amount of a-syn PFFs detected per well, which could actually be relatively the same per living cell. The authors describe using high content microscopy and image analysis for quantification, but it is not entirely clear whether total cell numbers per well are quantified in all cells, and whether accounting for this has any impact on the a-syn PFF intensity measurements.

Minor point. Figure 2B. In the figure legend it states that the color-code scaling is "at the bottom left". It appears to be in the top right of the B figure. Further, while I assume the scale to the left is for Cell area, and the one on the right is for PFF signal/cell area, this could be more clearly labelled. Also, it is somewhat confusing that the same color-code is used to represent 50-150%, and 0-200% (i.e. % of control sgRNAs) for the two different datasets.

2. Preparation and characterization of the a-syn PFFs. The design of the screening study relies heavily on the a-syn PFFs as a selection tool for identifying genes contributing to an increased or decreased capacity for a-syn PFF accumulation. However, while an effort has been made to characterize the PFFs its unclear to what extent monomeric or other non-fibrillar aggregation states have been excluded, or could be contributing to the cell associated "a-syn PFF" signals throughout the study. The a-syn PFFs are derived from recombinantly expressed a-syn, and the authors provide a TEM image and associated PFF length distribution analysis (SFig. 1). However, the fibrillation protocol does not seem to contain any form of centrifugation steps that could ensure partitioning of soluble oligomers/aggregates from insoluble fibrils, and there is no analysis of secondary structure using beta-sheet binding dyes (e.g. Thioflavin S/T) or similar (e.g. CD). The authors state that an SDS-PAGE separation and Coomassie staining of the monomeric a-syn prep confirmed a homogenous sample, but a similar analysis was not performed on the PFFs, and would be helpful in demonstrating the outcome of the aggregation protocol. Native PAGE under non-reducing/non-denaturing conditions, would better help to characterize these a-syn PFFs, and ideally should be carried out after the 633/488 labeling so that it is possible to assess how extensive/efficient this labeling is of the entire a-syn PFF sample. Additionally, all a-syn PFF internalization is tracked via the conjugated fluorophores; however, its unclear to what extent the internalized a-syn PFFs actually persist as "PFFs"; become disaggregated; and/or degraded, and in doing so assess whether the fluorophore is reliably reporting on a-syn distribution post internalization/accumulation. For example, the authors could separate cell lysates and immunoblot with a-syn antibodies, or look at the distribution with an a-syn antibody (ICC) in combination with conjugate imaging, and assess signal overlap.

3. Preparation of tau oligomers. The tau oligomers are prepared with 5 μ M of recombinant tau incubated with 5 μ M of heparin, which makes the related data difficult to interpret. Heparin is a highly-sulfated form of HS, which promotes aggregation of amyloid proteins and is typically considered to act as a scaffold to which monomers bind and form fibrils. Heparin addition in in vitro experiments is often used to compete with cell-surface HS interactions as a means of investigating the importance of cell surface HS in observed binding, internalization and signaling. While the average heparin polysaccharide chain length is not stated here, it is in molar equilibrium with the monomeric tau concentration, suggesting that every tau monomer has the potential to be heparin-bound/interacting. This heparin-interaction would be expected to interfere with cell-surface HS interactions, and so it is unclear to what extent the results in Fig. 3D, SFig. 4 (and related tau data) are likely influenced by this. As the a-syn PFFs are not prepared in a similar manner, and it is unclear to what extent they present similar structures (e.g. beta-sheets) as the tau-oligomers, it is also difficult to assess the relevance of any comparisons. Arguably all the experiments involving tau would need to be conducted with oligomers (or similar) that are prepared in a heparin-free manner.

As discussed above for a-syn PFFs. There is no structural characterization of these tau oligomer preparations. The oligomer term is often used to describe aggregates of amyloid proteins that are typically soluble, and generally lack a well-defined beta-sheet secondary structure, as determined by thioflavin T/S staining (or similar), native PAGE, or circular dichroism analysis. Further, amyloid proteins aggregated in the presence of heparin are often reported to more readily form fibrils (<https://www.sciencedirect.com/science/article/pii/S0022283623002930>). As suggested above for a-syn PFFs, the authors should provide data to define the aggregation state of the tau-oligomers (i.e. with and without heparin (if the authors intend to retain the tau/heparin oligomer data)), and make a side-by-side comparison with the a-syn PFF preparations. The authors attribute differences in a-syn PFFs and tau oligomer (which outcompeted a-syn PFFs) cell interactions to the smaller size of the tau oligomers, but there is no evidence supporting this claim in the manuscript. Further, as for a-syn PFFs, it would be relevant to establish to what extent the tau oligomers are labelled by the Cy3 labels, and whether subpopulations of aggregates/fibrils are equally represented. These comparisons are relevant given the differences in labeling approaches i.e. tau monomers were labelled prior to the heparin-mediated aggregation protocol, while for a-syn the sonicated PFFs were labelled.

4. Cargo specificity experiments. In these exps. the authors study the accumulation of different fluorescently tagged cargo molecules after 24 h of incubation with cells which are KO for one of the 8 genes that were identified as impacting a-syn PFF accumulation. From the data presented in Fig. 3 they conclude that only a-syn and tau are significantly affected by the different KOs.

Notably, in the accompanying SFig. 4 it appears that there are less cells in the tau and a-syn treated conditions than in the dextran, EGF, or transferrin-treated cultures. As discussed above, in Fig 3. the authors express values relative to the AAVS1 control, to account for random DSB effects of the sgRNA/Cas9 approach; however, this does not necessarily account for the possibility that the KOs may differentially impact viability or proliferation (see comment above). As this could change the number of cells available to interact with a-SYN PFFs (and tau oligomers) at the time of exposure, this could affect the interpretation of the analysis of increased/decreased a-syn PFF accumulation. The authors could consider including a marker to quantify total cell numbers at the end of the accumulation assay.

Comments on other cargo molecule selection: The authors incubate fluorescently tagged EGF and transferrin with the different KOs for 24 h, and then assess accumulation. This seems like a particularly long incubation period for proteins that, at least relative to the fibrillar a-syn and oligomeric tau preparations, would be expected to be more readily degraded and/or cleared, and may have receptors that are autoinhibited to downregulate signaling and associated endocytosis following such prolonged exposure. Consequently, analysis of these cargos at earlier timepoints may also reveal gene-specific differences in accumulation. Given this possibility it is not clear that the data in Fig. 3 convincingly supports the claims that the gene KOs specifically modulate the accumulation of proteinaceous aggregates.

Related/minor points:

- Can the authors offer an explanation why the COPB1 KO appears to have contradictory effects on the accumulation of tau (increases) and a-syn (decreases).
- The authors specify the MW of the dextran used, but it could be worth noting that this is (at least based on product number) an anionic preparation, and as such would not be expected to interact with HS chains.

5. 10E4 antibody. The 10E4 HS antibody is used throughout the study as a means of identifying altered availability of cell surface HS. The authors are careful in their presentation of data and related discussions to note the limitations of 10E4's utility. However, as discussed the relevance of the 10E4 results are at times unclear, and the authors should consider alternative means of assessing the availability of cell surface HS (e.g. using fluorescently tagged HS-binding proteins). Relating to Fig. 4:

- The authors could consider replacing "HS" in figure 4 with "10E4+" (or similar). As they discuss, 10E4 may not offer a complete detection of HS in general on these cells, and/or following these specific interventions.
- In panels C and D detailing the high-resolution distribution of 10E4 and a-syn PFF puncta, it is unclear exactly what the % of surface puncta refers to, there is no error/deviation and the number of experiments, cells or puncta analyzed does not seem to be provided

6. Trypsinization to distinguish attached and internalized. To distinguish between cell surface bound and internalized a-syn PFFs (and other cargo) the authors employ a trypsin wash (30 s) prior to fixation, and in SFig. 11 there are images of trypsin vs. PBS washed and fixed cells, comparing the degree of cell-associated a-syn PFFs following 2 h of incubation. However, there is no quantitative data supporting that the approach is in fact effective at making the proposed distinction, and for example in the WT trypsin and PBS washed samples the amount and intensity of a-syn PFFs/nuclei appears very similar following 2 h of a-syn PFF incubation. This could mean that the trypsinization is unnecessary as the majority of detectable a-syn PFFs are already internalized or that the trypsinization does not effectively cleave cell surface attached a-syn PFFs-HS chains on HSPG core proteins have been proposed to potentially sterically inhibit the access of trypsin to its cleavage sites, with potentially different cleavage efficiencies determined by the specific types of HSPGs (e.g. GPI-anchored glypicans vs. syndecans); the position of the HS chain on the core protein; and the properties of the chain itself. The effectiveness of trypsinization for digesting cell surface HS is somewhat supported in the HS compositional analysis exps. The authors carried out an overnight Trypsin/Lys C digestion, which their subsequent analysis demonstrates yielded HS-decorated core protein fragments in the supernatant fraction. However, this is a more extensive digestion protocol than that used for the imaging exps., and does not seem to discount the possibility that a population of HS decorated PGs would remain attached to the membrane. Further, if the trypsin protocol is efficiently cleaving cell surface proteins it would be reasonable to assume that it is also cleaving adhesion proteins required for cell attachment and spreading. However, cell morphology appears unaffected between the trypsin and PBS treated WT cells; perhaps 30 s of incubation is insufficient to observe these types of morphological changes, but it equally calls into question whether its sufficient to achieve the proposed objective of clearing surface attached a-syn PFFs.

I appreciate that it is challenging to establish reasonable criteria to make this on vs. distinction, especially when processing a large number of cells in multiple conditions at relatively low resolution. The rationale behind the approach taken by the authors is not unreasonable and may provide a good approximation of internalized a-syn PFFs. However, for the reasons described above, without some validation that the trypsin incubation does in fact reveal only internalized a-syn PFFs, this assumption is open to criticism. The authors should consider providing additional analysis to validate that this trypsin protocol actually separates cell surface from internalized a-syn PFFs. A suggestion would be to perform higher resolution confocal z-stacks in cells fixed and then co-stained with e.g. a plasma membrane marker/ endomembrane marker/cytosolic stain (to define intracellular volume) and to quantitatively assess a-syn PFF signal distribution in cells w/w trypsin incubation.

Associated point:

Image analysis details: The specifics of the image analysis used is not entirely clear. The cutoff for inclusion as a a-syn PFF positive cell does not seem to be defined

7. Effects of C3orf58 and SLC9A9 on HS and HSPGs, and a-syn PFF binding. In Fig. 2 C3orf58 sgRNA targeted cells are reported as having a significantly lower a-syn PFF signal/cell area, and further confirmed in Fig. 3. In Fig. 5 C3orf58 ^{-/-} cells revealed impaired a-syn PFF intensity after 24 h of incubation, and in C3orf58 ^{-/-} Fig. 6 revealed reduced surface a-syn PFF binding, with no significant effect on 10E4 immunoreactivity. Mass spec analysis determined a significant increase in total HS in C3orf58 ^{-/-} vs. WT, with additional significant increases in HSPG (SDC3, GPC1, GPC6) transcript expression, and associated HS biosynthesis enzymes. Therefore, C3orf58 ^{-/-} cells: 1. Increases the expression of HSPG transcripts, 2. Increases the expression of HS biosynthesis enzymes, 3. Increases total amount of HS, but does not alter HS composition, and 4. Present with similar levels of cell surface HS as WT based on 10E4 immunoreactivity.

Collectively these data raise a number of questions that do not seem to be clearly addressed. For example, it is unclear how the authors consider that the KO of the Golgi-associated genes is impacting mRNA expression of the detected HSPGs (or HS biosynthesis enzymes). Based on detection of increased CSPG4 mRNA expression in C3orf58 ^{-/-} cells, the authors suggest that there is possibly a more general 'dysregulation of glycosaminoglycan metabolism', but again it is unclear how mRNA transcript expression for these genes would be regulated by a deficiency in C3orf58 and SLC9A9. As outlined above in paragraph 3, the claim that HS maturation is defective is not clearly supported by the HS mass spec analysis. Further, the C3orf58^{-/-} cells have increased total HS, while SLC9A9^{-/-} cells do not, despite this both conditions reduced a-syn PFF accumulation. This disparity should be more clearly discussed, and as suggested above the authors should consider exps. potentially using alternative HS-binding proteins, to demonstrate altered availability of cell surface HS following the gene KOs.

Supplementary Figure 7. The Western blot in SFig. C should include immunoblotting for endogenous C3orf58. As far as I can tell, these blots are with antibodies against the HA-tag only (plus Actin loading controls). Without blotting with an antibody that detects the endogenous WT C3orf58 it is impossible to assess to what extent the ^{-/-} levels are actually different to the WT condition. Equally, the degree to which the rescue expression of C3orf58 is similar to WT cannot be determined. This seems particularly relevant given what the authors describe as the 'surprising' effects of C3orf58 ^{-/-} and rescue on a-syn PFF internalization (Fig 5D), 10E4 staining (SFig 6 and Fig. 6B) and a-syn PFF surface binding (Fig. 6B and C). Similarly, the blots should include immunolabelling for endogenous SLC39A9.

Additional minor points.

- Regarding the NDST1 immunoblots in Fig. 8a. The authors should consider providing the three original blots, add a MW marker, and clarify if the error bars in A represent SEM or SD?
- The authors comment that the lower MW for NDST1 in the SLC39A9^{-/-} sample may be due to reduced N-glycosylation. However, the actin loading control bands also appear to reveal different degrees of migration, so these differences may be simply technical issues relating to the electrophoresis.
- While the impaired trafficking of overexpressed HA-tagged NDST1 in the Golgi is an interesting finding it is also somewhat difficult to assess the relevance of this without comparing the overexpressed levels with endogenous (e.g. by performing HA-immunoblotting in addition to the NDST1 immunoblotting, similar to Fig. 8A)
- The version of the supplementary tables I could access only contained 1 of the tables, so it was not possible to assess Table S1 with regard to e.g. how well SLC35B2 was scored in the screen.

Reviewer #3

(Remarks to the Author)

The role of Heparan sulfate proteoglycans (HSPG) in a-syn internalization has attracted attention these last few years. Several authors have shown how extracellular matrix components (notably HSPC) play an important role in aggregate internalization, providing extensive data depending on cell type (PMID: 28827536) and in *Elegans* (PMID: 35790300), showing a mediator role of these molecules in PFF internalization, notably by neurons. In this work, authors use FACS-based genome-wide CRISPR/Cas9 knockout screening in RPE-1 cells to identify genes related to heparan sulfate proteoglycans that regulate the entry and accumulation of α -syn preformed fibrils (PFFs) in RPE-1. Authors identify several genes affecting PFF internalization, two of them being highly specific to α -syn PFF. Authors describe, identify and characterize these alterations. More specifically, authors use LC-MS/MS to provide information on HS chain structure and modifications.

Overall the in vitro work is well-structured and technically complete, although the model is somewhat reductionist. I believe the author's experiments are sufficiently self-explanatory for their conclusions, with a good quantity of controls and

supplementary data supporting them. While the important role of HSPC and its modifications in PFF internalisation has been shown already in the previous bibliography, the authors provide mechanistic insights and make here an interesting effort to characterize HSPG modifications specific to a-syn-PFF.

Both C3orf58 and SLC39A9 are important for the regulation of heparan sulfate proteoglycans (HSPGs) and, by extension, for PFF internalization. However, the way they affect HS modifications is not the same. In C3orf58 *-/-* cells, there's an overall increase in HS species with only minor compositional changes, whereas in SLC39A9 *-/-* cells, the HS composition is more severely altered—with a decrease in sulfated disaccharides and a relative increase in unmodified ones.

Authors make a first prediction in the discussion: “pattern of HS composition with an increased ratio of unmodified disaccharides coupled to a decrease in N-sulfated species was highly similar to that observed in C3orf58 resc.”

However, as the authors also state in the discussion, and admit as a limitation, the study fails to find the specific HSPC structure controlling PFF binding.

I believe that RPE-1 cells present a valuable opportunity to explore this open question further. It would be interesting to systematically investigate how variations in the length or proportion of unmodified disaccharides and sulfated species influence cell binding and internalization capacity

To further investigate the relationship between HS composition and PFF internalization, additional analyses are recommended. Size-exclusion chromatography or ion-mobility mass spectrometry could help determine the full-length distribution of HS chains, providing insights into whether and how chain length and modifications influences protein fibril binding and uptake. Even more interesting, authors could quantify the binding affinities of different HS species to PFFs by techniques such as surface plasmon resonance (SPR) or isothermal titration calorimetry (ITC), clarifying how specific sulfation patterns impact binding/internalization.

Beyond this, I believe some minor aspects would benefit from further clarification either in the introduction or discussion:

1. RPE-1 cells

Heparan sulphate proteoglycans (HSPGs) are abundant on the surface of RPE-1 cells, making them a good model for studying interactions between α -syn PFFs and HS receptors, however they differ significantly from neuronal cells. For example, work by Ihse and colleagues (PMID: 28827536). showed how HSPG role in PFF internalization depends on cell type, being relevant in neurons and oligodendrocytes, but secondary in microglia and astrocytes. The potential particularities of RPE-1 cells on results should be further discussed

2. HSPG as Receptors:

Several times in the text authors refer to HSPG as “receptors”. In my opinion this is a (lexical) inaccuracy. It is true previous bibliography refer to HSPG as “receptors” (PMID: 24145152) However, HSPGs are not classical receptors like G-protein-coupled receptors (GPCRs) or tyrosine kinase receptors. Instead, I believe it is important to state clearly HSPG are key multi-function components of the extracellular matrix. Among its different and varied roles (structural, communication...) they also hold an important role in reception. This should be specified in the introduction.

3. Next steps

Study of HSPG modifications is a promising avenue. Next steps should go towards testing the influence of PFF uptake and HSPG modifications in more complex models, such as organotypic or acute mouse models. Authors could state in the discussion potential difficulties and make recommendations in studying and characterizing these HSPG modifications in more complex systems in order to guide future research.

Version 1:

Reviewer comments:

Reviewer #2

(Remarks to the Author)

In their revised manuscript and rebuttal letter, Vanderperre et al. have made comprehensive efforts to address the majority of the questions and concerns raised in my initial report. This includes a substantial amount of new experimental work, as well as detailed and considered responses. I appreciate the time and effort that was invested to undertake this, and agree with the authors that their revised manuscript contains convincing evidence that further strengthens the interesting findings presented in their original draft, detailing the discovery of these novel regulators of HSPGs.

In particular, the authors have:

1. Added additional data to strengthen the characterization of the a-syn PFFs used in the study.
2. Used GFP+ to pursue an alternative means of assessing cell surface HS availability in the different C3orf58 *-/-* and SLC39A *-/-* conditions.
3. Conducted experiments in iPSC-derived dopaminergic neurons to assess the potential influence of cell-type specific differences in HS on the mechanism of a-syn PFF uptake.
4. Demonstrated that their trypsin treatment is effective at removing surface bound a-syn PFFs.

I remain concerned about the use of the heparin-seeded Cy3-labelled tau oligomers for investigating a potential role of cell surface HS. That said, I agree with the authors that the conclusions drawn from the tau data are reasonable, and as such they represent a minor part of the study as a whole. Further, I acknowledge that the authors have removed previous data in which the tau oligomers were co-incubated with a-syn PFFs, in which heparin could potentially have been a confounding factor. I do not consider that any further action is required with regard to this or any other aspects of the study, but I have taken this opportunity to clarify my concern below.

The data from EXT1 and NDST1 knockdown cells is convincing evidence that cell surface HS plays a role in tau association/internalization, and is in agreement with cited literature. However, I do not entirely agree with the authors claim that 'In our uptake experiment where only tau oligomers are analyzed (Fig 3D) we argue that the presence of heparin is not an issue, at least in our experimental conditions'. In their rebuttal they discuss MS data from Usenovic et al., demonstrating that the heparin in the tau prep is non-covalently associated with tau monomers, dimers, trimers and tetramers i.e. the majority of heparin is not free. However, as tau is already bound to heparin, it is reasonable to assume that any potential HS-binding sites on the tau protein are initially occupied by heparin. As noted, the heparin-tau interactions are non-covalent, and tau added to the cell medium would likely exist in some equilibrium between heparin-bound or cell surface HS-bound states; therefore, heparin may limit, but not prevent cell surface HS interactions. Potentially, this is the condition represented by the 100% control level of tau binding reported in Fig. 3D. The authors argue that if free heparin were competing with cell surface HS for tau, then EXT1 and NDST1 knockdown would not have further decreased tau oligomer association with cells. This is a reasonable assumption if heparin had blocked all cell surface HS interactions, but if it instead is limiting binding, then the additional reduction in cell surface HS by knockdown could be expected to further decrease cell association of tau-oligomers. It would have been interesting to determine if the non-significant reduction in cell associated tau in the C3orf58 knockdown (Fig. 3D) would have been more pronounced using tau oligomers produced without heparin, and perhaps the authors could consider including such a preparation if they pursue related work in the future.

Reviewer #3

(Remarks to the Author)

Overall, authors have made a remarkable effort to complete and perfect their work. Notably the deeper characterisation of PFF structure and the addition of PFF-uptake experiments in iPSC-derived dopaminergic neurons.

The cell-type specificity of PFF intake represents an interesting results by itself. While this calls for further analysis of cell-type chain specifics, the amount of work and analysis needed calls for a specific manuscript. The amount of data already present in the manuscript makes it fit for publication in my opinion.

Dear reviewers,

Thank you for your evaluation and constructive criticism of our manuscript. We have addressed your comments and questions carefully, as detailed in the point-by-point response below. This results in a strengthened study that we hope will be suitable for publication in *Communications Biology*. Changes to the main text are referenced to by lines numbers.

The most notable changes include:

- addition of **Supplementary Fig 1C-E**, strengthening our characterization of α -syn PFF,
- removal of **Supplementary Fig 4B** (tau oligomers/ α -syn PFF co-treatment),
- transfer of data from **Figure 4C** to **Supplementary Figure 13C,D**.
- addition of **Supplementary Figure 11E** showing efficiency of trypsin treatment to remove surface-bound PFF,
- addition of a GFP+ binding assay (HS-binding protein) in **Figure 6D,E** demonstrating decreased binding of another HS-binding protein to the cell surface of *C3orf58* $-/-$ and *SLC39A9* $-/-$ RPE-1 cells,
- addition of PFF uptake in *C3orf58* $-/-$ and *SLC39A9* $-/-$ iPSC-derived dopaminergic neurons in **Figure 9D,E**, showing that *C3orf58* deficiency impairs PFF uptake as in iPSC-microglia,
- addition of PFF binding and uptake assays in WT iDNs subjected to HS-altering treatments, demonstrating for the first time that HS are important for PFF binding and uptake in human dopaminergic neurons (incorporated in **Figure 4C-H**),
- modification of **Supplementary Figure 7**, to add detection of endogenous *C3orf58* and *SLC39A9* in our RPE-1 cell lines,
- addition of **Supplementary Figure 20**, showing Western blot analysis of inducible NDST1-HA expression levels,
- addition of **Supplementary Figure 21**, showing all full blots from the study,
- cosmetic changes to other Figures (overall harmonized graphs presentation including matching of color codes where applicable; individual data points shown).
- addition of a Data Availability Statement, Author contribution statement, and a “Statistics and Reproducibility” sub-section in the *Methods*.

Please also note that proteomics and RNAseq data have been deposited in public repositories and can be accessed during the review process.

- Surface proteomics data is available on the PRIDE repository:
Log in to the PRIDE website using the following details:
Project accession: PXD065224
Token: mAVP8e0PFqzC
Alternatively, log in to the PRIDE website using the following account details:
Username: reviewer_pxd065224@ebi.ac.uk
Password: AGMKOhIsN3Y7
- RNAseq data is available to the Gene Expression Omnibus
To review GEO accession GSE299483:
Go to <https://www.ncbi.nlm.nih.gov/geo/query/acc.cgi?acc=GSE299483>
Enter token gdqpuqucpzqfdut into the box

Point-by-point response to reviewers:

Reviewer #1 (Remarks to the Author):

Summary: Vanderperre et al. used a CRISPR/Cas9 KO screen and identified a number of genes that increase or decrease the accumulation of a-syn PFFs in RPE1 cells. They focus their study on a sub-selection of genes that regulate HSPGs, and two unexpected hits SLC39A9 (Golgi Zn²⁺ exporter) and C3orf58 (Golgi localized kinase). Enzymatic cleavage of cell surface HS and chlorate-treatment (intended to reduce HS sulfation) reduced cell associated a-syn PFFs. Further, in the abstract the authors claim that mass spec. analysis of SLC39A9 ^{-/-} and C3orf58 ^{-/-} cells indicated defective HS maturation, and was attributed to the impaired cell surface binding of a-syn PFFs. In C3orf58 KO RPE1 cells, an expressed HA-tagged NDST1 accumulated in the Golgi, which is suggested as a potential explanation for the defective HS maturation. Finally, C3orf58 KO in iPSC-derived microglia is found to impede a-syn PFF uptake. The data provided supports a role for cell surface HS as a point of cell surface contact for a-syn PFFs, and indicates that SLC39A9 and C3orf58 are required for normal Golgi function, which if perturbed impacts HS (and HSPG) homeostasis. The study adds additional support for a general role of cell surface HS in interactions with amyloid proteins. There are a number of interesting findings in the study relevant to those working in the HS, amyloid and PD fields. The authors have also taken what seems to be a careful and comprehensive approach to validate the genes identified in their screen, and address cell size as an assay confounder, which was interesting to read.

However, the study is lacking a comprehensive characterization of the a-syn PFFs, and given their importance in the screen for identifying the genes of interest the authors need to provide a more detailed description of what the a-syn PFF preparation actually contains.

> A more detailed characterization of the PFFs has been added to **Supplementary Figure 1** (see below and **L.110-111**, and *Methods* **L.940-941, L.955-981**), including the use of ThT labeling and differential centrifugation of soluble/insoluble material. We clearly show that our PFFs preparations are enriched in insoluble material, compared to monomers and oligomers, and show a dose- and time-dependent increase in ThT fluorescence. Our method of PFFs production and quality control (including electron microscopy, dynamic light scattering, and immunofluorescence, and comparison with monomers) has been described previously (Del Cid Pellitero et al., 2019; Maneca et al., 2019) and our PFFs preparations have been used recently to study PFFs uptake processes (Bayati et al., 2022). These references have been added to the corresponding *Methods* section. Overall, we are confident that our PFFs preparation mostly contain fibrillar, amyloid forms of a-syn, rather than soluble monomeric and oligomeric forms.

Supplementary Figure 1 – Recombinant α -syn-PFFs characterization

Relatedly, the fact that heparin is used in the protocol for forming tau oligomers is highly problematic (detailed below).

> A detailed answer to this concern is provided further down in this document.

The relevance of the RPE1 cell line for studying a-syn PFF uptake is also not clearly justified. HSPG expression patterns and HS composition is typically cell-type specific, and to what extent the observations would prove relevant for e.g. neurons in which a-syn inclusions (derived endogenously or transferred) are typically observed, is not clear. This potential for such cell-type specific effects is observed in Fig. 9, where the iPSC derived microglial reveal impaired a-syn PFF uptake in *C3orf58*^{-/-}, but not *SLC39A9*^{-/-} cells. The study is already of considerable size with the vast majority of the mechanistic findings derived from RPE1 cells, but the authors should provide evidence that the *C3orf58*^{-/-} impairment of a-syn PFFs in the microglia is in fact due to altered cell surface HS.

> These are legitimate concerns from the reviewer. RPE-1 cells were primarily used as they are of neuro-epithelial origin, and are pseudo-diploid, making them highly suitable for CRISPR/Cas9 genetic screening. To increase the relevance of our findings, we have now included additional PFFs uptake experiments in human iPSC-derived dopaminergic neurons (iDNs, **Figure 9 D,E**, and addition of iDNs in the pipeline in **panel A**; see below) where, similar to microglia, an approximately 60% decrease in uptake was observed in the *C3orf58*^{-/-} line. Related modifications to the main text can be found in **L.427-428, L.437-440, L.547-554**, and experimental details are described in the *Methods* section: **L.805-809**.

While no effect might be observed in other cell types, this additional finding brings further interest for future evaluation of the therapeutic potential of *C3orf58* and the pathway in which it acts to regulate PFF uptake. This will be the focus of future studies. This finding also suggests that the level of HS organization that is modulated by *C3orf58* is likely shared between RPE-1 and key PD cell types such as dopaminergic neurons and microglia.

Figure 9. *C3orf58* is important for PFFs uptake in human microglia and dopaminergic neurons

Regarding SLC39A9, microglia and dopaminergic neurons derived from *SLC39A9* *-/-* iPSCs had normal PFF uptake capacities, indeed indicating a level of cell-type specificity. This could be due to variability in the role of SLC39A9 in HS levels/composition between cell types or functional redundancy *via* the expression of paralog genes encoding other Golgi-localized Zn²⁺ transporters, as illustrated below for dopaminergic neurons (unpublished RNAseq data from a collaborator).

Expression of the 3 Golgi-localized *SLC39A* paralogs in DA neurons – redundancy ?

Evaluation of whether and how cell surface HS is affected by loss of C3orf58 in microglia and dopaminergic neurons is an important endeavor. Our LC-MS/MS absolute quantification of surface disaccharides in RPE1 (**Fig 7**) demonstrates how an IF strategy of HS surface analysis (**Fig 6A,C**) provides limited knowledge for analyzing C3orf58-dependent HS perturbations. Thus, in our opinion, HS surface analysis in microglia and dopaminergic neurons should be done by LC-MS/MS until a more specific perturbation is found (HS chains numbers and length, as well as domains size and organization will be evaluated in follow-up work). However, LC-MS/MS analysis of surface HS in iPSC-derived cells is limited by the confounding contribution of inadequately differentiated cells to the samples and would require significant additional funds that we cannot spare for now (cost of differentiation for large amounts of cells). We believe that our findings that C3orf58 plays a role in α -syn PFFs uptake in PD-relevant microglia and dopaminergic neurons are of sufficient impact for the scope of this paper, and hope to uncover the specific perturbations caused by C3orf58 loss in the future using more specific assays.

However, taking the reviewer’s comment into consideration, we added experiments demonstrating that HS are important players in PFF binding and uptake in iDNs, using heparinase and sodium chlorate treatments (novel **Figure 4C-H**). We have not found evidence elsewhere in the literature that HS have been proven as playing a role in this process in human dopaminergic neurons. These findings, combined with the reduction in PFF uptake in C3orf58 KO iDNs, highlight the biological relevance of better understanding how HS drive α -syn aggregates propagation. The following text was added in the *Results* section:

L.199-211: “To the best of our knowledge, it is unknown if HS are also involved in α -syn PFFs uptake in human dopaminergic neurons. To test this, we generated iPSC-derived dopaminergic neurons (iDNs) and treated them with heparinases or sodium chlorate before PFF binding and uptake assays. Both treatments significantly decreased PFFs binding to the surface of dopaminergic neurons (**Figure 4C,D**), although the low dynamic range of HS signal likely prevented observation of any change in HS intensity by HS-altering treatments (**Figure 4E**). These treatments were however confirmed to work properly, as a significant reduction in HS in presence of heparinases or sodium chlorate was observed in a 3 hours PFFs uptake assay (**Figure 4F,H**). PFF uptake in iDNs was significantly decreased upon exposure to sodium chlorate (app. 40% of control). This was not observed for heparinases (**Figure 4G**), which could be due to gradual surface presentation of initially intracellular pools of HSPGs during the 3h uptake assay, enzymatic inactivation with time, or the action of other PFFs receptors. Together, our data confirm that HS are major receptors for cell surface binding of α -syn PFFs, including in RPE-1 cells and dopaminergic neurons, even if we cannot exclude that other receptors might compensate for PFF uptake in iDNs when HS are altered.”

Figure 4. – HSPGs are major receptors for α -syn PFFs binding on the cell surface

The following text was added in the *Methods* section (L.795-804):

“For PFFs binding assay in iDNs following HS-altering treatments, iDNs were differentiated for 2 weeks in 96 well plates, and treated with either Heparinase I+III (3U/ml each, 1 hour) or 60 mM sodium chlorate (48 hours) before transfer on ice and incubation with 180 nM PFFs-A488 and 1/500 anti-HS antibody (10e4 epitope) for 20 min. iDNs were then fixed, and processed for TH immunostaining, 10e4 signal detection with a fluorescence secondary antibody, and nuclear staining with Hoechst 33342. Images were acquired using an Opera Phoenix high-content imaging platform before image analysis using Harmony Columbus software. PFFs spot intensity was quantified in TH-positive area and normalized to the PFFs-only condition. HS intensity in TH-positive area was also measured. PFFs uptake in WT iDNs following heparinases or sodium chlorate treatment was quantified in a similar manner, except that iDNs were incubated with PFFs-A488 and anti-HS antibody for 3 hours at 37°C before fixation.”

A major finding in the study is that *C3orf58*^{-/-} and *SLC39A9*^{-/-} cells accumulate/bind less a-syn PFFs than controls. Digestion of cell surface HS (with heparinase), or reducing the degree of HS sulfation (chlorate treatment), also reduced a-syn PFF binding. However, mass spec revealed that *C3orf58*^{-/-} cells had higher total levels of HS (cell surface, based on the digestion protocol, correct?), more HSPG transcripts, and more HS biosynthetic enzymes. While at the same time there was no apparent difference in 10E4 binding, indicating similar availability of cell surface HS (albeit for this 10E4-positive subpopulation of HS). *SLC39A9*^{-/-} cells also had significantly less a-syn PFF signal intensity, and significantly less 10E4-positivity, but mass spec reveals no significant differences in total HS (relative to WT). The authors discuss how the di/tetrasaccharide HS compositional analysis may not provide sufficient resolution to determine differences in HS chain domains that might be important for a-syn PFF binding. It is particularly difficult to reconcile how suppressing these two genes, proposed as potential “general regulators of glycans/glycosaminoglycans biosynthesis and/or HS maturation in the Golgi”, differently effect total HS levels, while both similarly impact a-syn PFF binding. The discussion around these differences needs greater clarity.

> For *SLC39A9*^{-/-}, while the decrease in total surface HS levels does not reach significance in our triplicate analysis (p=0.198, **Fig 7E**), the 3 most abundant disaccharides in RPE1 cells are significantly decreased albeit by varying magnitudes (**Fig 7F,G,H**). We think this is compatible with our results from 10E4-positivity by IF. *C3orf58*^{-/-} surface HS levels were indeed higher by LC-MS/MS but this did not translate in 10E4-positivity, which is not unexpected since observing a very specific (poorly characterized) epitope brings only limited information compared to mass spectrometry.

The current overarching hypothesis, that would reconcile the fact that both genes “**differently effect total HS levels, while both similarly impact a-syn PFF binding**” is that there is a general increase (*C3orf58*^{-/-}) or decrease (*SLC39A9*^{-/-}) in several disaccharides that accompanies the appearance of HS chains that are structurally distinct from WT. Whether this is due to changes in length or numbers of HS chains, or in length, numbers and distribution of high-/low-sulfation domains along the chains remains to be determined. This has been clarified in the *Discussion* by the addition of two sentences:

L.508-512: “Despite KO of *C3orf58* and *SLC39A9* having opposite effects on global HS levels in our LC-MS/MS analysis, they had similar effects on binding of HS ligands (PFF and GFP+) to the cell surface, strongly indicating that both genes regulate HS homeostasis, although the specific perturbations of HS chains require additional analyses especially for *C3orf58*.”

L.517-518: “The contribution of more complex parameters (length or numbers of HS chains, or length, numbers and distribution of high-/low-sulfation domains along the chains) to PFF binding remains to be determined”

Additional experimental proof that the cell surface HS in *C3orf58*^{-/-} and *SLC39A9*^{-/-} cells has altered binding potential could perhaps be explored using other established HS-binding proteins (e.g. fluorescently labelled FGF), and performing a similar approach as taken with the 10E4 labelling of live cells. As mentioned above, the abstract claims that the mass spec data indicates defective HS maturation in *C3orf58* and *SLC39A9*. However, if I understood correctly the conclusion from the composition analysis is that the distribution of the different di- and tetrasaccharide motifs are similar to control, while total HS is significantly increased in *C3orf58*; therefore, the claim of ‘defective HS maturation’ is not clearly supported by this data.

> We added experimental proof that cell surface HS is altered in both KO lines, using positively charged GFP as a cargo (GFP+, new **Fig 6D,E**). This engineered GFP exposes positive charges at its surface, allowing interaction with negatively charged sulfate groups on HS chains (Zhang et al., 2020). Its binding was significantly decreased in *C3orf58*^{-/-} and *SLC39A9*^{-/-} lines. The following text was added in the *Results* section:

L.341-343: “Interestingly, the cell surface binding of GFP+, a recombinant version of GFP exposing positively charged residues to mediate its HS-dependent cell surface binding and endocytosis¹⁶, was significantly decreased as well in *C3orf58* *-/-* and *SLC39A9* *-/-* cells (**Figure 6D,E**).”

We added this text to the *Methods*:

L.784-785: “GFP+ binding assay was performed as for PFFs binding assay, except cells were treated with 50 nM GFP+ for 20 min without co-treatment with anti-HS antibody.”

Figure 6. PFFs binding to the cell surface is decreased in *C3orf58* *-/-* and *SLC39A9* *-/-* cells

Currently, the consequence of these differences in HS-mediated accumulation of a-syn PFFs is not examined. Below I suggest that it might be important to consider/account for the potential effects of gene KOs on cell viability, and possibly even proliferation effects that may not be accompanied by cell size changes. Equally, the effect of a-syn PFF exposure, and relative differences of a-syn PFF binding/accumulation on cell viability would seem relevant to contextualize the findings, and establish whether impairing HS-mediated a-syn PFF interactions would be predicted to have protective effects.

> Our analyses already take into account potential effects of KOs on cell viability when quantifying PFF uptake.

1. In the screen itself, it is the relative enrichment/depletion of sgRNAs in the Low PFF vs High PFF population that allows to identify genes that positively/negatively influence PFF accumulation. Thus, if a specific gene KO reduced/increased viability relative to other cells in the perturbed population of the genome-wide screen, this will influence the number of reads for the specific sgRNAs targeting that gene relative to the total pool of sgRNAs present in the sample, but not the relative enrichment in the Low/High PFF populations sorted by FACS.

2. The legend of **Figure 2B** (screen validation) specifies that the normalization of PFF uptake to cell area was done on a per-cell basis. In other words, we calculated, for each cell analyzed, the mean PFF/cell area value, then calculated the mean value for all analyzed cells, before normalizing to the AAVS1 sgRNA control mean value. Hence, if variations in the number of cells occur upon a specific gene KO, it has no consequence in our analysis. The analysis conducted is outlined in the “*Validation of screen hits by high-content microscopy*” subsection of the *Methods* section.

3. For all subsequent PFF uptake experiments, the total PFF intensity is measured for each cell analyzed, then the mean total PFF intensity was obtained for each condition, and compared to the mean total PFF intensity of the control. This value is independent of the number of cells analyzed. As a precaution, we always counted cells before seeding and plated the same number of cells for each genotype. More details have been added to the “*PFFs uptake and binding assays for PFFs and GFP+, and live staining of surface heparan sulfate*” subsection of the *Methods* section (L.789-794).

> In RPE1 cells, we did not observe any change in cell viability in WT upon PFF treatment (see below, Cell Titer Blue or Nuclei count for viability assessment with increasing concentration of PFFs). Thus, a protective effect of the KOs cannot be tested in this *in vitro* setup.

PFF uptake itself is not particularly toxic to cultured cells except for neurons exposed for at least 14 days. Testing a protective effect of *C3orf58* KO in our iPSC-derived dopaminergic neurons model would require 14 days of differentiation, followed by 14 days of PFF treatment. Given technical difficulties encountered to maintain our neurons healthy beyond 14 days after the start of the final differentiation (regardless of genotype), we unfortunately are not in a position to test this in a timely manner.

Below are more detailed comments relating to specific experiments and figures:

1. CRISPR/Cas9 screen and assessment of a-syn PFF binding/accumulation. Several of the identified genes were connected to cell growth/cell cycle. The authors discuss how altering cell

cycle such that proliferation rate will be increased will result in smaller cells, while an increase will result in larger cells, impacting the cellular capacity to accumulate a-syn PFFs. Further, they discuss how the CRISPR/Cas9 system itself introduces DSBs that are known to impact cell cycle rates, and how the detection of non-targeting sgRNAs (such as LacZ) in the low a-syn PFF group (Fig. 1) is likely due to the fact that they are proliferating faster (as they are not incorporating DSB-associated DNA damage), resulting in smaller cells with reduced capacity to accumulate a-syn PFFs. They go on to perform experiments in SFig. 2 that convincingly demonstrate how cells targeted with LacZ sgRNA/Cas9 are smaller than cells transduced with the genome wide CRISPR library, which leads to lower absolute amounts of a-syn PFFs, but relatively similar amounts, when a cell size correction is implemented. The authors subsequently added sgRNA targeting the safe-harbour control AAVS1 to account for side-effects of the CRISPR/Cas9 approach, and as seen in Fig. 2 the majority of KOs that significantly impact area are cell growth/cycle regulators.

However, in subsequent figures where average a-syn PFF intensity is expressed relative to the AAVS1 (e.g. Fig. 3 and SFig. 4) or WT controls (e.g. Fig. 5 and 6) it is potentially important to determine whether the specific KO of genes, particularly *C3orf58* and *SLC39A9*, impacts cell viability (e.g. via FACs based apoptosis/necrosis assay using Annexin V/PI labeling or similar), and/or proliferation rates (irrespective of cell size/area). As outlined, increased proliferation rate may reduce cell size and capacity to accumulate a-syn PFFs; however, it will also result in a relative reduction in the amount of a-syn PFFs available per cell.

> Differences in proliferation could indeed be a concern. However, in previous genome-wide CRISPR/Cas9 screens for fitness genes by Hart et al., it was shown that *C3orf58* and *SLC39A9* are not fitness genes in RPE-1 cells (Table S2 therein, showing Bayes factors of -4,533 and -56,859 respectively, where negative values are associated with genes that do not affect fitness). This is in agreement with our **Figure 2**, that shows that polyclonal *C3orf58* or *SLC39A9* KO cells display no significant difference in cell area. In addition, we performed cell counting of polyclonal KO cells from **Fig 2** (see graph on the right), and no significant difference in cell number was observed for *C3orf58*, while a decrease was observed for *SLC39A9*, but which could be the result of more efficient double strand

breaks induction than controls by the sgRNAs targeting this gene, not necessarily to the gene affecting fitness itself. In a nutshell, we observed a decrease in PFFs fluorescence per cell despite no change (*C3orf58* KO) or a decrease (*SLC39A9* KO) in cell number, and a similar cell size (EGFP mask area) compared to control. Thus we are confident that the chain of events proposed by the reviewer (underlined sentence above) does not occur in the present context, and that other factors than cell fitness explain the role of *C3orf58* and *SLC39A9* in PFF uptake.

Equally, reduced viability may decrease cell numbers and the total amount of a-syn PFFs detected per well, which could actually be relatively the same per living cell. The authors describe using high content microscopy and image analysis for quantification, but it is not entirely clear whether total cell numbers per well are quantified in all cells/genotypes, and whether accounting for this has any impact on the a-syn PFF intensity measurements.

> As explained further up in this document, our method of quantification was cell-based rather than well-based, so any variation in cell number was inherently taken into account.

Minor point. Figure 2B. In the figure legend it states that the color-code scaling is “at the bottom left”. It appears to be in the top right of the B figure. Further, while I assume the scale to the left is for Cell area, and the one on the right is for PFF signal/cell area, this could be

more clearly labelled. Also, it is somewhat confusing that the same color-code is used to represent 50-150%, and 0-200% (i.e. % of control sgRNAs) for the two different datasets.
 > Labeling issues have been corrected (L.1359). We think the color-code provides a clear overview of the data, even if the dynamic range of variations is different between the two parameters.

Figure 2. – Screen validation by high-content microscopy

2. Preparation and characterization of the a-syn PFFs. The design of the screening study relies heavily on the a-syn PFFs as a selection tool for identifying genes contributing to an increased or decreased capacity for a-syn PFF accumulation. However, while an effort has been made to characterize the PFFs its unclear to what extent monomeric or other non-fibrillar aggregation states have been excluded, or could be contributing to the cell associated “a-syn PFF” signals throughout the study. The a-syn PFFs are derived from recombinantly expressed a-syn, and the authors provide a TEM image and associated PFF length distribution analysis (SFig. 1). However, the fibrillation protocol does not seem to contain any form of centrifugation steps that could ensure partitioning of soluble oligomers/aggregates from insoluble fibrils, and there is no analysis of secondary structure using beta-sheet binding dyes (e.g. Thioflavin S/T) or similar (e.g. CD). The authors state that an SDS-PAGE separation and Coomassie staining of the monomeric a-syn prep confirmed a homogenous sample, but a similar analysis was not performed on the PFFs, and would be helpful in demonstrating the outcome of the aggregation protocol.

> As explained further up in the document, additional characterization data has been added to **Supplementary Figure 1**, including the use of ThT to confirm the formation of beta-sheet containing amyloid fibrils (**Supplementary Figure 1C**) and of a sedimentation assay (**Supplementary Figure 1D,E**) confirming the large enrichment in insoluble material in the PFF samples compared to monomers and oligomers. The corresponding methods have been added.

Native PAGE under non-reducing/non-denaturing conditions, would better help to characterize these a-syn PFFs, and ideally should be carried out after the 633/488 labeling so that it is possible to assess how extensive/efficient this labeling is of the entire a-syn PFF sample. Additionally, all a-syn PFF internalization is tracked via the conjugated fluorophores; however, its unclear to what extent the internalized a-syn PFFs actually persist as “PFFs”; become disaggregated; and/or degraded, and in doing so assess whether the fluorophore is reliably reporting on a-syn distribution post internalization/accumulation. For example, the authors could separate cell lysates and immunoblot with a-syn antibodies, or look at the distribution with an a-syn antibody (ICC) in combination with conjugate imaging, and assess signal overlap.

> Regarding the concerns with labeling efficiency of PFF samples, it should first be noted that any dye that has not been conjugated to the PFF was dialyzed out of the samples (see *Methods*), so any observed fluorescent signal does not come from free dye in the PFF preparations. It is true that we do not have a guarantee that 100% of individual fibrils have been labeled with the Alexa dyes. Yet, since we are only reporting on the fluorescent signal, we do not consider unlabeled PFF to be an issue. As suggested by the reviewer, we performed IF staining using an anti a-syn antibody (BD Biosciences) in cells treated for 18 h with PFF-A488. The vast majority of the PFF signal appeared colocalized with the a-syn signal, which was confirmed by colocalization analysis with ImageJ Coloc2 plugin, at a resolution of 4.83 px/um: Manders' M1=0.742 (autothresholded pixels of PFF signal being above autothresholded pixels of a-syn signal). Thus, in agreement with the widely accepted use of conjugated PFF in the field (with Alexa dyes or other), our PFF fluorescent signal accurately reports on exogenous a-syn distribution post internalization.

In addition, C3orf58 and SLC39A9 deficiency had clear effects on PFF binding (**Figure 6**) and early uptake (15 min, **Supplementary Figure 11**), where degradation/disaggregation are much less likely to impact the ability of the fluorescent probe to report on PFF load.

3. Preparation of tau oligomers. The tau oligomers are prepared with 5 μM of recombinant tau incubated with 5 μM of heparin, which makes the related data difficult to interpret. Heparin is a highly-sulfated form of HS, which promotes aggregation of amyloid proteins and is typically

considered to act as a scaffold to which monomers bind and form fibrils. Heparin addition in *in vitro* experiments is often used to compete with cell-surface HS interactions as a means of investigating the importance of cell surface HS in observed binding, internalization and signaling. While the average heparin polysaccharide chain length is not stated here, it is in molar equilibrium with the monomeric tau concentration, suggesting that every tau monomer has the potential to be heparin-bound/interacting. This heparin-interaction would be expected to interfere with cell-surface HS interactions, and so it is unclear to what extent the results in Fig. 3D, SFig. 4 (and related tau data) are likely influenced by this. As the a-syn PFFs are not prepared in a similar manner, and it is unclear to what extent they present similar structures (e.g. beta-sheets) as the tau-oligomers, it is also difficult to assess the relevance of any comparisons. Arguably all the experiments involving tau would need to be conducted with oligomers (or similar) that are prepared in a heparin-free manner.

As discussed above for a-syn PFFs. There is no structural characterization of these tau oligomer preparations. The oligomer term is often used to describe aggregates of amyloid proteins that are typically soluble, and generally lack a well-defined beta-sheet secondary structure, as determined by thioflavin T/S staining (or similar), native PAGE, or circular dichroism analysis. Further, amyloid proteins aggregated in the presence of heparin are often reported to more readily form fibrils (<https://www.sciencedirect.com/science/article/pii/S0022283623002930>). As suggested above for a-syn PFFs, the authors should provide data to define the aggregation state of the tau-oligomers (i.e. with and without heparin (if the authors intend to retain the tau/heparin oligomer data)), and make a side-by-side comparison with the a-syn PFF preparations. The authors attribute differences in a-syn PFFs and tau oligomer (which outcompeted a-syn PFFs) cell interactions to the smaller size of the tau oligomers, but there is no evidence supporting this claim in the manuscript.

> We thank the reviewer for the care taken in evaluating the biochemical procedures used to generate protein aggregates studied in our paper. More information regarding tau oligomers is available in a publication from Merck (Usenovic et al., 2015, now referenced in the *Methods* **L.990**), who gifted us the Cy3-labeled tau oligomers. Briefly, these oligomers (formed 4 h after heparin addition) are on a path to the formation of beta-sheet rich, Thioflavin positive tau fibrils (after 5-15 days of incubation), but the oligomers are Thioflavin negative, so they are structurally different from our a-syn PFF. Given that our purpose in **Fig3** was to evaluate the effect of the KO of our hits on the uptake of another HS-dependent kind of protein aggregates, we argue that their differing structure is not an issue and does not change the point of the experiment.

Besides, as explained by the reviewer, the presence of heparin is used to compete and block cell-surface interaction of proteinaceous aggregates that depend on HS for binding. In Usenovic et al., Figure 1C shows the following (figure on the right): “High-mass MALDI-TOF MS comparing cross-link-stabilized (red) and untreated (blue) tau oligomers demonstrates that our preparations consist of a heterogeneous, multimeric population of tau monomers, dimers, trimers, and tetramers that include noncovalent binding with heparin.”

In our uptake experiment where only tau oligomers are analyzed (**Fig 3D**), we argue that the presence of heparin is not an issue, at least in our experimental conditions. If free heparin was significantly competing with tau oligomers uptake, then no additional inhibition by KO of *EXT1* or *NDST1* would be observable, but we do see a significant decrease when these two

genes are targeted with CRISPR. Yet, we agree that the presence of free heparin in the sample could be a confounding factor in **Supplementary Figure 4B**, where cells are co-treated with tau oligomers and a-syn PFFs. Competition of free heparin with a-syn PFFs offers a reasonable alternative explanation to our proposition of competition by smaller size tau oligomers. In absence of a clear explanation of the observed competition between preparations, we removed **Supplementary Figure 4B** from the manuscript.

Further, as for a-syn PFFs, it would be relevant to establish to what extent the tau oligomers are labelled by the Cy3 labels, and whether subpopulations of aggregates/fibrils are equally represented. These comparisons are relevant given the differences in labeling approaches i.e. tau monomers were labelled prior to the heparin-mediated aggregation protocol, while for a-syn the sonicated PFFs were labelled.

> Usenovic et al., 2015, shows that Tau monomers are conjugated to Cy3 with final degree of labeling 3.2 (3.2 moles of dye/mole of protein), and that no apparent populations of higher order aggregates are present based on atomic force microscopy and ThT assay.

4. Cargo specificity experiments. In these exps. the authors study the accumulation of different fluorescently tagged cargo molecules after 24 h of incubation with cells which are KO for one of the 8 genes that were identified as impacting a-syn PFF accumulation. From the data presented in Fig. 3 they conclude that only a-syn and tau are significantly affected by the different KOs.

Notably, in the accompanying SFig. 4 it appears that there are less cells in the tau and a-syn treated conditions than in the dextran, EGF, or transferrin-treated cultures. As discussed above, in Fig 3. the authors express values relative to the AAVS1 control, to account for random DSB effects of the sgRNA/Cas9 approach; however, this does not necessarily account for the possibility that the KOs may differentially impact viability or proliferation (see comment above). As this could change the number of cells available to interact with a-SYN PFFs (and tau oligomers) at the time of exposure, this could affect the interpretation of the analysis of increased/decreased a-syn PFF accumulation. The authors could consider including a marker to quantify total cell numbers at the end of the accumulation assay.

> As for evaluation of PFFs uptake, the accumulation of other cargoes was measured at the single cell level before calculation of the mean accumulation per cell in each condition, and subsequent normalization to the AAVS1 control. Cell numbers thus have no impact on the data presented. In addition, it should be noted that the images shown in **Supplementary Figure 4** are for illustration purpose only and do not represent the actual cell density across a given well and across replicates analyzed.

Comments on other cargo molecule selection: The authors incubate fluorescently tagged EGF and transferrin with the different KOs for 24 h, and then assess accumulation. This seems like a particularly long incubation period for proteins that, at least relative to the fibrillar a-syn and oligomeric tau preparations, would be expected to be more readily degraded and/or cleared, and may have receptors that are autoinhibited to downregulate signaling and associated endocytosis following such prolonged exposure. Consequently, analysis of these cargoes at earlier timepoints may also reveal gene-specific differences in accumulation. Given this possibility it is not clear that the data in Fig. 3 convincingly supports the claims that the gene KOs specifically modulate the accumulation of proteinaceous aggregates.

> Since accumulation, rather than uptake, was evaluated here, we purposely chose the same 24 h time point as done for PFF in the screen (which has been used previously for EGF). This allows, indeed, to take into account several steps that contribute to the accumulation: binding, endocytosis, trafficking, recycling, feedback on receptors, degradation rate, etc. Choosing any

other timepoint would thus prevent the possible action of some of these mechanisms and impair our ability to compare accumulation of the other cargoes with that of PFFs. While we cannot exclude that some intermediate steps might be altered by some gene KOs for some of the control cargoes, we think our conclusion that the accumulation of these other cargoes is vastly unchanged stands true.

As a precaution, we added the hereafter underlined statement in the corresponding *Results* section (L.173-174): “Among the seven facilitators of α -syn PFFs accumulation, none of the gene KOs decreased dextran, EGF, or transferrin accumulation, nor did the silencing of *VPS35* increase the accumulation of these cargoes (**Figure 3A-C**), at least at the 24 h time point tested.”

Related/minor points:

- Can the authors offer an explanation why the COPB1 KO appears to have contradictory effects on the accumulation of tau (increases) and a-syn (decreases).

> In COPB1 polyclonal KO cells, we have observed high variability in the amount of protein aggregates per cell, as can be appreciated in **Supplementary Figure 4**. For instance, some cells have virtually accumulated no tau oligomers, while others have large amounts of them compared to the AAVS1 control. The same can be seen for a-syn treatment, although the tendency was more towards a decreased accumulation. This is reminiscent of what we have seen in cells expressing variable levels of NDST1-HA, with a drop in PFF accumulation when NDST1-HA is most abundant (see **Fig 8B**, WT + dox condition, and graph below).

Overall, we attribute the contradictory effects of COPB1 on variability of the mutations present in the polyclonal KO cell populations, and their downstream effects on processes governing HS-dependent proteinaceous aggregate uptake. Such downstream effects could be on the expression levels or proper sorting of HS enzymes and HSPGs.

- The authors specify the MW of the dextran used, but it could be worth noting that this is (at least based on product number) an anionic preparation, and as such would not be expected to interact with HS chains.

> This is indeed the case, allowing us to test bulk endocytosis independent from HS-dependency.

5. 10E4 antibody. The 10E4 HS antibody is used throughout the study as a means of identifying altered availability of cell surface HS. The authors are careful in their presentation of data and related discussions to note the limitations of 10E4's utility. However, as discussed the relevance of the 10E4 results are at times unclear, and the authors should consider alternative means of assessing the availability of cell surface HS (e.g. using fluorescently tagged HS-binding proteins).

> The surface binding of GFP+ has been added to **Figure 6** see higher up in the document). We think this helps clarify the effects of C3orf58 and SLC39A9 on the binding capacity of HS chains exposed at the cell surface, at least for the analyzed cargoes.

Relating to Fig. 4:

- The authors could consider replacing “HS” in figure 4 with “10E4+” (or similar). As they discuss, 10E4 may not offer a complete detection of HS in general on these cells, and/or following these specific interventions.

> We have changed “HS” to “HS (10e4+)” in **Figure 4A**, **Figure 6A**, **Supplementary Figure 6A**, and **Supplementary Figure 13**.

- In panels C and D detailing the high-resolution distribution of 10E4 and a-syn PFF puncta, it is unclear exactly what the % of surface puncta refers to, there is no error/deviation and the number of experiments, cells or puncta analyzed does not seem to be provided

> We have clarified in the figure legend what this quantification in our previous **Figure 4C** means (now moved to **Supplementary Figure 13C**), presented that quantification in a graph format, and moved it to a new panel (**Supplementary Figure 13D**). Briefly, n=698-2248 PFF puncta per replicate (N=3) per condition were classified as colocalizing with a surface HS puncta (10e4 epitope) or not. The percentage of colocalized puncta for cells treated with sodium chlorate or not was plotted, revealing that perturbation of HS sulfation with sodium chlorate significantly reduced colocalization between PFF and HS puncta at the cell surface.

Supplementary Figure 13. C3orf58 is necessary for optimal colocalization of surface-bound PFFs with surface HS

6. Trypsinization to distinguish attached and internalized. To distinguish between cell surface bound and internalized a-syn PFFs (and other cargo) the authors employ a trypsin wash (30 s) prior to fixation, and in SFig. 11 there are images of trypsin vs. PBS washed and fixed cells, comparing the degree of cell-associated a-syn PFFs following 2 h of incubation. However, there is no quantitative data supporting that the approach is in fact effective at making the proposed distinction, and for example in the WT trypsin and PBS washed samples the amount and intensity of a-syn PFFs/nuclei appears very similar following 2 h of a-syn PFF incubation. This could mean that the trypsinization is unnecessary as the majority of detectable a-syn PFFs are already internalized or that the trypsinization does not effectively cleave cell surface attached a-syn PFFs-HS chains on HSPG core proteins have been proposed to potentially sterically inhibit the access of trypsin to its cleavage sites, with potentially different cleavage efficiencies determined by the specific types of HSPGs (e.g. GPI-anchored glypicans vs. syndecans); the position of the HS chain on the core protein; and the properties of the chain itself. The effectiveness of trypsinization for digesting cell surface HS is somewhat supported in the HS compositional analysis exps. The authors carried out an overnight Trypsin/Lys C digestion, which their subsequent analysis demonstrates yielded HS-decorated core protein fragments in the supernatant fraction. However, this is a more extensive digestion protocol than that used for the imaging exps., and does not seem to discount the possibility that a population of HS decorated PGs would remain attached to the membrane. Further, if the trypsin protocol is efficiently cleaving cell surface proteins it would be reasonable to assume that it is also cleaving adhesion proteins required for cell attachment and spreading. However, cell morphology appears unaffected between the trypsin and PBS treated WT cells; perhaps 30 s of incubation is insufficient to observe these types of morphological changes, but it equally calls into question whether its sufficient to achieve the proposed objective of clearing surface attached a-syn PFFs.

I appreciate that it is challenging to establish reasonable criteria to make this on vs. in distinction, especially when processing a large number of cells in multiple conditions at relatively low resolution. The rationale behind the approach taken by the authors is not unreasonable and may provide a good approximation of internalized a-syn PFFs. However, for the reasons described above, without some validation that the trypsin incubation does in fact reveal only internalized a-syn PFFs, this assumption is open to criticism. The authors should consider providing additional analysis to validate that this trypsin protocol actually separates cell surface from internalized a-syn PFFs. A suggestion would be to perform higher resolution confocal z-stacks in cells fixed and then co-stained with e.g. a plasma membrane marker/endomembrane marker/cytosolic stain (to define intracellular volume) and to quantitatively assess a-syn PFF signal distribution in cells w/wo trypsin incubation.

Associated point:

Image analysis details: The specifics of the image analysis used is not entirely clear. The cutoff for inclusion as a a-syn PFF positive cell does not seem to be defined

> The use of a 90 sec trypsin wash to eliminate non-internalized PFF after on-ice binding to HeLa cells was published previously (Bayati et al., 2022, Fig S1G therein, shown below).

Nevertheless, to show that our 30s treatment (adjusted to prevent cell detachment) performed as expected in our specific assay and in our RPE1 cells, we quantified the PFF intensity in cells incubated for 20 min on ice with 180 nM PFF-A633, and then washed or not with trypsin before fixation. Cells were imaged with an epifluorescence microscope at 20x. Image quantification was performed using ImageJ. Briefly, for each cell, an ROI was drawn manually using the Hoechst and WGA stainings as guides. The PFF signal was thresholded (pixel intensities min=1087, max=1580) and the Mean PFF fluorescence intensity per cell was measured. PFF positive and negative cells were identified manually on thresholded images.

Our results clearly show that our trypsin wash method robustly degrades surface-bound PFF. This data was added as **Supplementary Figure 11E**.

Details of the image analysis have been added to the *Methods* section (**L.815-818**), and precisions have been made in the legend of **Supplementary Figure 11**.

7. Effects of C3orf58 and SLC9A9 on HS and HSPGs, and a-syn PFF binding. In Fig. 2 C3orf58 sgRNA targeted cells are reported as having a significantly lower a-syn PFF signal/cell area, and further confirmed in Fig. 3. In Fig. 5 C3orf58 $-/-$ cells revealed impaired a-syn PFF intensity after 24 h of incubation, and in C3orf58 $-/-$ Fig. 6 revealed reduced surface a-syn PFF binding, with no significant effect on 10E4 immunoreactivity. Mass spec analysis determined a significant increase in total HS in C3orf58 $-/-$ vs. WT, with additional significant increases in HSPG (SDC3, GPC1, GPC6) transcript expression, and associated HS biosynthesis enzymes. Therefore, C3orf58 $-/-$ cells: 1. Increases the expression of HSPG transcripts, 2. Increases the expression of HS biosynthesis enzymes, 3. Increases total amount of HS, but does not alter HS composition, and 4. Present with similar levels of cell surface HS as WT based on 10E4 immunoreactivity.

>Specifically for point 4., we

Collectively these data raise a number of questions that do not seem to be clearly addressed. For example, it is unclear how the authors consider that the KO of the Golgi-associated genes is impacting mRNA expression of the detected HSPGs (or HS biosynthesis enzymes). Based on detection of increased CSPG4 mRNA expression in C3orf58 $-/-$ cells, the authors suggest that there is possibly a more general 'dysregulation of glycosaminoglycan metabolism', but again it is unclear how mRNA transcript expression for these genes would be regulated by a deficiency in C3orf58 and SLC9A9.

> Since both proteins are localized to the Golgi apparatus, we think that transcriptomic changes observed are better explained as secondary effects, i.e. compensatory mechanisms that KO

cells might develop to counteract defects in specific cellular pathways (here, HS and possibly glycosaminoglycan homeostasis). We added this possible explanation of these changes at the mRNA level in the discussion (underlined):

L.490: “Fifth, both *C3orf58* and *SLC39A9* KO lines displayed changes in HSPG-related genes (**Supplementary Figure 14**, that may arise in response to HS dyshomeostasis) and ...”.

Given that the changes observed at the transcriptomic level are highly similar between the two KO lines, we strongly think that these changes are not a common underlying cause of defective PFF/HS binding, otherwise they would lead to similar changes in HS levels/composition, which our LC-MS/MS analysis infirms. Thus, we feel like investigating the chain of events leading to these transcriptomic changes is out of the scope of this manuscript.

As outlined above in paragraph 3, the claim that HS maturation is defective is not clearly supported by the HS mass spec analysis. Further, the *C3orf58*^{-/-} cells have increased total HS, while *SLC9A9*^{-/-} cells do not, despite this both conditions reduced a-syn PFF accumulation. This disparity should be more clearly discussed, and as suggested above the authors should consider exps. potentially using alternative HS-binding proteins, to demonstrate altered availability of cell surface HS following the gene KOs.

> We agree that a maturation defect of HS in *C3orf58*-deficient cells is not directly detected in our MS analysis. However, if the only change in HS was their surface levels in these cells, then the PFF binding should be increased, which is not the case. In addition, GFP⁺ also had defective binding in *C3orf58* KO cells (new **Figure 6 D,E**), and entry of other HS-dependent cargoes are impaired by *C3orf58* deficiency (*Discussion*, **L.492-495**, Zika and Chikungunya viruses, and recent addition of cell-surface RNAs to the list)

We feel like we have taken sufficient precautions at times in our narrative (e.g. in the *Discussion*, **L.524-525**: “*C3orf58* and *SLC39A9* may act as general regulators of glycans/glycosaminoglycans biosynthesis and/or maturation in the Golgi.”). But we agree that this should be the case throughout the manuscript regarding this indirect interpretation of defective maturation. Thus, where applicable (e.g. *Abstract* and other parts of the *Discussion*), we have changed the term “maturation” to “homeostasis” (**L.41, 97, 490**).

We also added a clearer discussion of the disparity mentioned by the reviewer in the *Discussion* section: **L.508-512**: “Despite KO of *C3orf58* and *SLC39A9* having opposite effects on global HS levels in our LC-MS/MS analysis, they had similar effects on binding of HS ligands (PFF and GFP⁺) to the cell surface, strongly indicating that both genes regulate HS homeostasis, although the specific perturbations of HS chains require additional analyses especially for *C3orf58*.”.

Supplementary Figure 7. The Western blot in SFig. C should include immunoblotting for endogenous *C3orf58*. As far as I can tell, these blots are with antibodies against the HA-tag only (plus Actin loading controls). Without blotting with an antibody that detects the endogenous WT *C3orf58* it is impossible to assess to what extent the ^{-/-} levels are actually different to the WT condition. Equally, the degree to which the rescue expression of *C3orf58* is similar to WT cannot be determined. This seems particularly relevant given what the authors describe as the ‘surprising’ effects of *C3orf58*^{-/-} and rescue on a-syn PFF internalization (Fig 5D), 10E4 staining (SFig 6 and Fig. 6B) and a-syn PFF surface binding (Fig. 6B and C). Similarly, the blots should include immunolabelling for endogenous *SLC39A9*.

> We have added blots against the endogenous proteins (see **Supplementary Figure 7C,D** and full blots in **Supplementary Figure 21**). We confirm loss of *SLC39A9* in KO cells, and expression of WT and H185R mutant rescue constructs at levels comparable to the endogenous protein (the D159A was expressed at lower levels). For *C3orf58*, as we have attempted previously with several commercial and home-made antibodies, we did not manage to detect

the endogenous protein (no band that would specifically be lost in the KO line was detected), and estimation of overexpression was also hindered by the detection of a non-specific band around the expected MW of endogenous C3orf58. An example with a commercial antibody from Proteintech (cat. 27145-1-AP) is shown in **Supplementary Figure 7C**.

Supplementary Figure 7 – Localization of SLC39A9 and C3orf58 in RPE-1 cells

Yet, in the past, a home-made C3orf58-specific antibody (see example below) robustly detected the HA-tagged version in the *C3orf58*^{-/-} resc. WT line, indicating an evident overexpression relative to endogenous levels. Unfortunately, this antibody is no longer available for us to probe all samples shown in **Supplementary Figure 7C**. The blot below has been incorporated as **Supplementary Figure 21E** (mentioned in **L. 329**).

1. RPE-1 WT
2. RPE-1 C3orf58 KO + rescue C3orf58-HA (WT)
3. RPE-1 C3orf58 KO

Additional minor points.

- Regarding the NDST1 immunoblots in Fig. 8a. The authors should consider providing the three original blots, add a MW marker, and clarify if the error bars in A represent SEM or SD?
- > We thank the reviewer for noticing these oversights. We added the following text to the legend (**L.1555**): “Bar graphs represent mean \pm SD.” The three original blots are provided in **Supplementary Figure 21**, and the MW marker was added.

Figure 8. NDST1 is mistrafficked in C3orf58 $-/-$ cells, and NDST1 overexpression inhibits PFFs uptake

- The authors comment that the lower MW for NDST1 in the SLC39A9 $-/-$ sample may be due to reduced N-glycosylation. However, the actin loading control bands also appear to reveal different degrees of migration, so these differences may be simply technical issues relating to the electrophoresis.

> The actin blot had been cropped before rotation of the image to ensure horizontality of the migration. This has been corrected (see above), and from the original blots that were added in **Supplementary Figure 21**, the NDST1 shift in the SLC39A9 $-/-$ line is now more evident.

- While the impaired trafficking of overexpressed HA-tagged NDST1 in the Golgi is an interesting finding it is also somewhat difficult to assess the relevance of this without comparing the overexpressed levels with endogenous (e.g. by performing HA-immunoblotting in addition to the NDST1 immunoblotting, similar to Fig. 8A)

> We have added blots in **Supplementary Figure 20** to compare the expression levels of endogenous NDST1 vs NDST1-HA in cell lines containing the NDST1-HA cassette (+/- dox), and the control parental lines. Even with a clear induction of NDST1-HA in presence of dox

(see HA blot), there was no perceptible increase in total NDST1 signal when the membranes were probed with an antibody against endogenous NDST1. A brief description of this was added to the *Results* section (see underlined text below), and we thank the reviewer as it improves the relevance of our findings (L.412-413): “Upon doxycycline induction, NDST1-HA levels increased similarly independent from genotype (Figure 8B,C), yet showing low levels of expression relative to endogenous NDST1 (see absence of increase in total NDST1 in Supplementary Figure 20).”

Supplementary Figure 20 – Expression of dox-inducible NDST1-HA by Western blot

• The version of the supplementary tables I could access only contained 1 of the tables, so it was not possible to assess Table S1 with regard to e.g. how well SLC35B2 was scored in the screen.

> We suggest the reviewer contact the journal to gain access to the correct file, or download it directly from our *bioRxiv* version of the manuscript at the following link: <https://www.biorxiv.org/content/10.1101/2023.09.29.560170v2.supplementary-material>. For the reviewers' convenience, here is a screenshot of the SLC35B2-related data in Table S1, with NDST1, COPG1 and EXT1 as comparisons:

	A	B	C	D	E	F	G	H	I	J	K	L	M
1	Table S1: MAGeCK analysis of NGS data from genome-wide CRISPR screen - enrichment in cells with 15% lower vs 15% higher PFF signal												
2													
3	id	num	neg.score	neg.p-value	neg.fdr	neg.rank	neg.goodsgrn	pos.score	pos.p-value	pos.fdr	pos.rank	pos.goodsgrna	
4	NDST1	4	1	1	1	18055	0	2,56E-09	2,74E-07	0,00135	2	4	
5	COPG1	4	1	1	1	18054	0	2,26E-08	8,22E-07	0,00135	3	4	
6	EXT1	4	0,99997	0,99996	1	18044	0	4,43E-08	8,22E-07	0,00135	4	4	
7	SLC35B2	4	0,99849	0,99854	1	17997	0	0,00047081	0,0015213	0,460883	59	3	
8													
9	column identifier	definition											
10	id	Gene symbol											
11	num	The number of targeting sgRNAs for each gene in the TKOv3 library											
12	neg.score	The RRA (robust rank aggregation) lo value of this gene in negative selection, i.e. depletion in cells with 15% lower PFF signal											
13	neg.p-value	The raw p-value (using permutation) of this gene in negative selection, i.e. depletion in cells with 15% lower PFF signal											
14	neg.fdr	The false discovery rate of this gene in negative selection, i.e. depletion in cells with 15% lower PFF signal											
15	neg.rank	The ranking of this gene in negative selection, i.e. depletion in cells with 15% lower PFF signal											
16	neg.goodsgrna	The number of "good" sgRNAs, i.e., sgRNAs whose ranking is below the alpha cutoff (determined by the --gene-test-fdr-threshold option), in negative selection.											
17	pos.score	The RRA (robust rank aggregation) lo value of this gene in positive selection, i.e. enrichment in cells with 15% lower PFF signal											
18	pos.p-value	The raw p-value (using permutation) of this gene in positive selection, i.e. enrichment in cells with 15% lower PFF signal											
19	pos.fdr	The false discovery rate of this gene in positive selection, i.e. enrichment in cells with 15% lower PFF signal											
20	pos.rank	The ranking of this gene in positive selection, i.e. enrichment in cells with 15% lower PFF signal											
21	pos.goodsgrna	The number of "good" sgRNAs, i.e., sgRNAs whose ranking is below the alpha cutoff (determined by the --gene-test-fdr-threshold option), in positive selection.											

Reviewer #2 (Remarks to the Author):

The role of Heparan sulfate proteoglycans (HSPG) in α -syn internalization has attracted attention these last few years. Several authors have shown how extracellular matrix components (notably HSPC) play an important role in aggregate internalization, providing extensive data depending on cell type (PMID: 28827536) and in *Elegans* (PMID: 35790300), showing a mediator role of these molecules in PFF internalization, notably by neurons.

In this work, authors use FACS-based genome-wide CRISPR/Cas9 knockout screening in RPE-1 cells to identify genes related to heparan sulfate proteoglycans that regulate the entry and accumulation of α -syn preformed fibrils (PFFs) in RPE-1. Authors identify several genes affecting PFF internalization, two of them being highly specific to α -syn PFF. Authors describe, identify and characterize these alterations. More specifically, authors use LC-MS/MS to provide information on HS chain structure and modifications.

Overall the *in vitro* work is well-structured and technically complete, although the model is somehow reductionist. I believe the author's experiments are sufficiently self-explanatory for their conclusions, with a good quantity of controls and supplementary data supporting them. While the important role of HSPC and its modifications in PFF internalisation has been shown already in the previous bibliography, the authors provide mechanistic insights and make here an interesting effort to characterize HSPG modifications specific to α -syn-PFF.

Both *C3orf58* and *SLC39A9* are important for the regulation of heparan sulfate proteoglycans (HSPGs) and, by extension, for PFF internalization. However, the way they affect HS modifications is not the same. In *C3orf58* $-/-$ cells, there's an overall increase in HS species with only minor compositional changes, whereas in *SLC39A9* $-/-$ cells, the HS composition is more severely altered—with a decrease in sulfated disaccharides and a relative increase in unmodified ones.

Authors make a first prediction in the discussion: “pattern of HS composition with an increased ratio of unmodified disaccharides coupled to a decrease in N-sulfated species was highly similar to that observed in *C3orf58* resc.”

However, as the authors also state in the discussion, and admit as a limitation, the study fails to find the specific HSPC structure controlling PFF binding.

I believe that RPE-1 cells present a valuable opportunity to explore this open question further. It would be interesting to systematically investigate how variations in the length or proportion of unmodified disaccharides and sulfated species influence cell binding and internalization capacity

To further investigate the relationship between HS composition and PFF internalization, additional analyses are recommended. Size-exclusion chromatography or ion-mobility mass spectrometry could help determine the full-length distribution of HS chains, providing insights into whether and how chain length and modifications influences protein fibril binding and uptake.

> We thank the reviewer for noticing our efforts to highlight our study's limitations. We agree that our model offers further opportunities to explore the relationship between HS composition and PFF internalization. Our future plans involve analyzing HS chains (number and length) and distribution of sulfated/non-sulfated domains along the chain. Given the large amount of data already present in the present manuscript, and the fact that we need additional funding for this (which we are actively seeking), we hope the reviewer will understand that these additional experiments are not realistically achievable in the context of this submission. These considerations have nevertheless been added to the *Discussion* (2 sentences added):

L.508-512: “Despite KO of *C3orf58* and *SLC39A9* having opposite effects on global HS levels in our LC-MS/MS analysis, they had similar effects on binding of HS ligands (PFF and GFP+)

to the cell surface, strongly indicating that both genes regulate HS homeostasis, although the specific perturbations of HS chains require additional analyses especially for *C3orf58*.”

L.517-518: “The contribution of more complex parameters (length or numbers of HS chains, or length, numbers and distribution of high-/low-sulfation domains along the chains) to PFF binding remains to be determined”

Even more interesting, authors could quantify the binding affinities of different HS species to PFFs by techniques such as surface plasmon resonance (SPR) or isothermal titration calorimetry (ITC), clarifying how specific sulfation patterns impact binding/internalization.

> Such analyses, testing the impact of various sulfation patterns on binding of α -syn fibrils (and other proteinaceous aggregates), was already done by Stopchinski et al. using a microarray of modified heparin polysaccharides. They tested the role of N-sulfation, O-sulfation, and overall sulfation (desulfated vs oversulfated) in the binding of these aggregates. While length and specific sulfate moieties were important for tau aggregates binding, this was not the case for α -syn, which did not depend on single sulfate moieties, but on overall sulfation status. These results were confirmed by Ihse et al.. This is consistent with our findings, as only EXT1 or NDST1 (not other sulfation enzymes) were hit in our screen, with NDST1 being considered as impacting overall sulfation (Marques et al.).

These elements are discussed in the *Discussion* section:

“In contrast to the binding of tau aggregate to HS which relies on specific N- and 6-O sulfation, it appears that no specific sulfate moiety, but rather global sulfation levels, are necessary for α -syn PFF binding and uptake^{26,36}. We confirm these findings since we did not find specific changes in the di-/tetra-saccharide unit at the cell surface of *C3orf58* *-/-* and *SLC39A9* *-/-* cells that correlated with changes in α -syn PFF binding/uptake (**Figure 7, Supplementary Figure 15**). The observation that KO of *NDST1* greatly decreases PFF uptake (**Figure 2B, Figure 3E**) is insufficient to conclude that N-sulfation is the sole necessary modification for HS/PFF interaction. Indeed, NDST1 activity adds an N-sulfate group that favors higher overall sulfation by facilitating the action of most other HS modification enzymes⁶⁹.”

Overall, the accumulated knowledge from the literature, and the fact that *C3orf58* KO reduces PFF binding while increasing HS levels without modifying di-/tetra-saccharide composition, point towards higher order HS chains organization as key for PFF binding, which we will address in the future.

Beyond this, I believe some minor aspects would benefit from further clarification either in the introduction or discussion:

1. RPE-1 cells

Heparan sulphate proteoglycans (HSPGs) are abundant on the surface of RPE-1 cells, making them a good model for studying interactions between α -syn PFFs and HS receptors, however they differ significantly from neuronal cells. For example, work by Ihse and colleagues (PMID: 28827536). showed how HSPG role in PFF internalization depends on cell type, being relevant in neurons and oligodendrocytes, but secondary in microglia and astrocytes. The potential particularities of RPE-1 cells on results should be further discussed

> The quoted text below was added to the *Discussion*, highlighting possible specificities but also points of convergence between RPE-1 cells and other cell types of interest.

L.547-554: “A limit of our study is the use of RPE-1 cells as the primary model, which are likely not recapitulating the HS-related biology of more PD-relevant cell types (e.g. expression levels, modifications, trafficking). This is illustrated by unperturbed PFF uptake in *SLC39A9*-

deficient human iPSC-derived microglia and dopaminergic neurons, which could be due to expression of redundant Golgi-localized Zn²⁺ transporters (e.g. SLC39A7, SLC39A13). Nevertheless, our initial focus on RPE-1 led us to identify C3orf58 as a key factor for PFF uptake in human dopaminergic neurons and microglia, suggesting that the RPE-1 model remains useful for studying how C3orf58 regulates HS homeostasis in the future, in the context of α -syn spreading but also regarding fundamental aspects of HS biology.”

2. HSPG as Receptors:

Several times in the text authors refer to HSPG as “receptors”. In my opinion this is a (lexical) inaccuracy. It is true previous bibliography refer to HSPG as “receptors” (PMID: 24145152) However, HSPGs are not classical receptors like G-protein-coupled receptors (GPCRs) or tyrosine kinase receptors. Instead, I believe it is important to state clearly HSPG are key multi-function components of the extracellular matrix. Among its different and varied roles (structural, communication...) they also hold an important role in reception. This should be specified in the introduction.

> We agree that the most common view of HSPG regards them as organizers and multi-functional components of the extracellular matrix, as well as coreceptors. However, as the reviewer points out, their role as receptors has also been observed, although it is less clearly understood (several examples are provided in a review by Sarrazin et al.). When they act as receptors, they and their cargo (that piggyback on HSPG) are directed to lysosomes, in a lipid-raft dependent manner, which is compatible with macropinocytosis (**Supplementary Fig 19** and as previously reported in Bayati et al. & Holmes et al.). We thus think that referring to HSPG as receptors is accurate. Nevertheless, we have now added a clear mention in the *Introduction* that HSPG display other key – non-receptor – roles as membrane/ECM components:

L.71-73: “HSPGs are key multifunctional components of the cell surface and extracellular matrix (ECM) that play crucial structural and communication roles and can act as receptors/coreceptors of a wide variety of ligands²⁷.”

3. Next steps

Study of HSPG modifications is a promising avenue. Next steps should go towards testing the influence of PFF uptake and HSPG modifications in more complex models, such as organotypic or acute mouse models. Authors could state in the discussion potential difficulties and make recommendations in studying and characterizing these HSPG modifications in more complex systems in order to guide future research.

> These important considerations are now mentioned at the end of the *Discussion*:

L.564-570: “Validation of *C3orf58* as a gene regulating the spread of α -syn aggregation *in vivo* or using human organotypic *in vitro* models will be an important milestone for further demonstrating its therapeutic potential. More generally, the use of such complex systems to study if and how targeting HS is a promising avenue for stopping the progression of synucleinopathies. Because of the broad importance of HS in many biological processes, such studies will likely require precise spatio-temporal control over experimenter-induced HS perturbations, and a deeper understanding of mechanisms governing HS-PFF interactions and organization of HS chains in general.”